# *RAS* mutations drive proliferative chronic myelomonocytic leukemia via a KMT2A-PLK1 axis

Ryan M. Carr [1,2,13], Denis Vorobyev[3,13], Terra Lasho[1], David L. Marks[2], Ezequiel J. Tolosa [2], Alexis Vedder[4], Luciana L. Almada[2], Andrey Yurcheko[3], Ismael Padioleau [3], Bonnie Alver[5], Giacomo Coltro[1], Moritz Binder [1], Stephanie L. Safgren [2], Isaac Horn[2], Xiaona You [6], Eric Solary [7], Maria E. Balasis[4], Kurt Berger[8], James Hiebert[1], Thomas Witzig [1], Ajinkya Buradkar[1], Temeida Graf[9], Peter Valent[9,10], Abhishek A. Mangaonkar [1], Keith D. Robertson[5], Matthew T. Howard[11], Scott H. Kaufmann [1], Christopher Pin [8], Martin E. Fernandez-Zapico[2], Klaus Geissler[12], Nathalie Droin [7], Eric Padron[4], Jing Zhang [6], Sergey Nikolaev [3✉] & Mrinal M. Patnaik [1✉]

Proliferative chronic myelomonocytic leukemia (pCMML), an aggressive CMML subtype, is associated with dismal outcomes. RAS pathway mutations, mainly *NRAS^{G12D}*, define the pCMML phenotype as demonstrated by our exome sequencing, progenitor colony assays and a *Vav-Cre-Nras^{G12D}* mouse model. Further, these mutations promote CMML transformation to acute myeloid leukemia. Using a multiomics platform and biochemical and molecular studies we show that in pCMML RAS pathway mutations are associated with a unique gene expression profile enriched in mitotic kinases such as polo-like kinase 1 (*PLK1*). *PLK1* transcript levels are shown to be regulated by an unmutated lysine methyl-transferase (KMT2A) resulting in increased promoter monomethylation of lysine 4 of histone 3. Pharmacologic inhibition of PLK1 in *RAS* mutant patient-derived xenografts, demonstrates the utility of personalized biomarker-driven therapeutics in pCMML.

[1] Division of Hematology, Department of Internal Medicine, Mayo Clinic, MN, USA. [2] Schulze Center for Novel Therapeutics, Division of Oncology Research, Mayo Clinic, MN, USA. [3] INSERM U981, Gustave Roussy Cancer Center, Villejuif, France. [4] Chemical Biology and Molecular Medicine Program, Moffitt Cancer Center, Tampa, FL, USA. [5] Molecular Pharmacology and Experimental Therapeutics, Mayo Clinic, MN, USA. [6] McArdle Laboratory for Cancer Research, University of Wisconsin-Madison, Madison, WI, USA. [7] INSERM U1170 and Department of Hematology, Gustave Roussy Cancer Center, Villejuif, France. [8] London Regional Transgenic and Gene Targeting Facility, Lawson Health Research Institute University of Western Ontario, London, ON, Canada. [9] 5TH Department of Internal Medicine I, Division of Hematology and Hemostaseology, Medical University of Vienna, Vienna, Austria. [10] Ludwig Boltzmann Institute for Hematology and Hemostaseology, Medical University of Vienna, Vienna, Austria. [11] Department of Laboratory Medicine and Pathology, Mayo Clinic, MN, USA. [12] Sigmund Freud University Vienna, Vienna, Austria. [13] These authors contributed equally: Ryan M. Carr, Denis Vorobyev. ✉email: sergey.nikolaev@gustaveroussy.fr; patnaik.mrinal@mayo.edu

Chronic myelomonocytic leukemia (CMML) is an aggressive hematological malignancy characterized by sustained peripheral blood (PB) monocytosis (absolute monocyte count/AMC ≥1×109/L, with monocytes ≥10% of the white blood cell count differential), bone marrow (BM) dysplasia and an inherent risk for leukemic transformation (LT) to secondary acute myeloid leukemia (sAML)[1,2]. The only curative modality, allogeneic hematopoietic cell transplantation (HCT), is rarely feasible because of age (median age at diagnosis 73 years) and comorbidities[2–4]. Patients not eligible for HCT receive symptom-guided therapies, including hypomethylating agents (HMA)[2,5]. While HMA epigenetically restore hematopoiesis in a subset of CMML patients, they fail to significantly alter the disease course or impact mutational allele burdens[5,6]. Finally, relative to patients with de novo AML, survival in sAML is much shorter (<4 months), even with the use of AML-directed therapies[7]. Thus, there is an urgent and unmet need for rationally developed therapies for patients with CMML.

The 2016 iteration of the World Health Organization (WHO) classification of myeloid malignancies has called for the recognition of proliferative CMML (pCMML) and dysplastic (dCMML) subtypes based on a diagnostic white blood cell count (WBC) of ≥13×109/L for the former[1]. This classification has often been criticized as being somewhat arbitrary and biological differences remain to be explored. The molecular fingerprint for CMML combines recurrent mutations in approximately 40 genes, some of which are associated with poor outcomes (ASXL1, NRAS, RUNX1, and SETBP1), and are incorporated into CMML-specific prognostic scoring systems[8–10]. Prior studies have demonstrated a higher frequency of RAS mutations (NRAS/KRAS) in pCMML; hypothesized to be secondary events and associated with poor outcomes[11]. Unresolved questions in CMML include whether there are true biological differences between pCMML and dCMML, how genetic and epigenetic events contribute to CMML progression and LT, and whether these molecular events render CMML vulnerable to specific therapeutic interventions.

In the present study, by using a combination of DNA sequencing, RNA-, and chromatin immunoprecipitation and sequencing (ChIP-seq), we demonstrate that pCMML and dCMML are independent biological entities, with genetic, transcriptomic, and epigenetic differences; with pCMML being defined by the presence of clonal RAS pathway mutations. Using whole and targeted exome sequencing, in vitro patient-derived progenitor colony forming assays and a Vav-Cre-Nras[G12D] mouse model, we demonstrate conclusively that clonal NRAS mutations can drive the pCMML phenotype. In clonal RAS pathway mutant pCMML, RAS pathway mutations result in increased expression of the mitotic checkpoint kinase PLK1 through the lysine methyltransferase KMT2A (MLL1). KMT2A mediates expression of PLK1 via monomethylation of histone 3 (H3) lysine 4 (K4) at the PLK1 promoter. While monomethylation of H3K4 is usually placed by KMT2C/2D, in clonal RAS pathway mutant pCMML, this histone mark is specifically regulated by an unmutated KMT2A. Further, pharmacological inhibition of PLK1 decreases monocytosis and hepatosplenomegaly and improves hematopoiesis in patient-derived, RAS-mutant pCMML xenografts. These results support a role for clonal RAS pathway mutations in pCMML biology, CMML progression to AML; demonstrating oncogenic function of an unmutated KMT2A, laying a framework for rationally defined personalized therapies in CMML.

## Results

### Clonal RAS pathway mutations correlate with proliferative CMML. Survival differences were analyzed between WHO-defined dCMML and pCMML subtypes. Univariate analysis of

a cohort of 1183 Mayo Clinic-GFM-Austrian CMML patients (Supplementary Table 1; dCMML, 607 and pCMML, 576), median age 72 years (range, 18–95 years), 66% male, with a median follow-up of 50 months (range, 41–54.5 months) validated an inferior overall-survival (OS) of pCMML patients relative to dCMML (23 vs 39.6 months; p < 0.0001; Fig. 1A). This stratification also predicted a shorter AML-free survival in patients with pCMML versus dCMML (Fig. 1B, 18 vs 32 months; p < 0.0001).

Genetic differences were assessed between the subtypes in 973 molecularly annotated Mayo Clinic-GFM-Austrian CMML patients (477 with pCMML and 496 with dCMML) using a targeted Targeted next-generation sequencing (NGS) assay designed to detect mutations in oncogenic RAS pathway genes; NRAS, KRAS, CBL, and PTPN11. Of note, while NF1 is an important component of the RAS pathway[1], it is the least frequently mutated RAS pathway gene in CMML and was not included in the NGS panels (Supplementary Fig. 1A)[12]. A higher frequency of NRAS (30% versus 8%; p < 0.001) and oncogenic RAS pathway mutations (46% versus 25%, p < 0.001) were identified in pCMML versus dCMML (Fig. 1C and Supplementary Table 2). Additional differences between pCMML and dCMML included a higher frequency of ASXL1 (60% vs 42%; p < 0.0001), JAK2[V617F] (12% vs 5%; p = 0.00011) mutations and trisomy 8 (11% vs 4%; p = 0.02) in pCMML, while TET2 (43% vs 54%; p = 0.002) mutations were more common in dCMML (Supplementary Table 2). Phenotypically, pCMML patients had higher white blood cell (WBC) counts, absolute monocyte counts (AMC), peripheral blood (PB) blasts, and circulating immature myeloid cells (IMC), relative to dCMML patients (Supplementary Fig. 1B).

Patients stratified by the presence or absence of NRAS mutations demonstrated an inferior OS for NRAS mutant CMML versus NRAS wild type (24 vs 33 months; p = 0.0055; Fig. 1D). This impact was also retained for cumulative oncogenic RAS pathway mutations (Supplementary Fig. 1C). pCMML with NRAS mutations was the most aggressive CMML subtype, with significantly shorter OS (22 months; p = 0.0007) relative to all other CMML subtypes combined (33 months) (Fig. 1E). Notably, OS was not significantly different when comparing each CMML subtype stratified by NRAS mutational status (Supplementary Fig. 1D). Similarly, NRAS mutant pCMML had the shortest AML-FS (16 months; p = 0.0001) versus other subtypes combined (AML-FS 29 months) and individually (Fig. 1F and Supplementary Fig. 1E). In addition, based on variant allele frequency (VAF) analysis, in 977 CMML patients that underwent NGS, there was a higher frequency of clonal NRAS, KRAS and CBL mutations in pCMML relative to dCMML (Fig. 1G and Supplementary Fig. 1F).

We then estimated the odds ratios of driver mutations including NRAS selected by a L1-regularized logistic regression model to assess their impact on the pCMML/dCMML phenotype, LT risk, OS, and AML-FS (Fig. 1H, I). Mutations negatively impacting OS and AML-FS included NRAS, ASXL1, RUNX1, DNMT3A, and EZH2 (Fig. 1H, I). We then applied a multivariate cox model for prediction of OS using mutations identified above and then combined the mutational data with clinical variables known to be prognostic in CMML (Supplementary Fig. 1G–J)[8–10]. Analysis was first carried out for individual RAS pathway mutations and then for oncogenic (combined) RAS pathway mutations. Importantly, mutant NRAS alone did not reach statistical significance as an independent factor impacting OS or AML-FS. The combined oncogenic RAS pathway category was statistically significant in a model that only included genetic factors (HR = 1.2, CI: 1.08–1.33), while there was a trend towards inferior outcomes when assessed in a model that included genetic and clinical parameters (HR = 1.11, CI: 0.99–1.24).

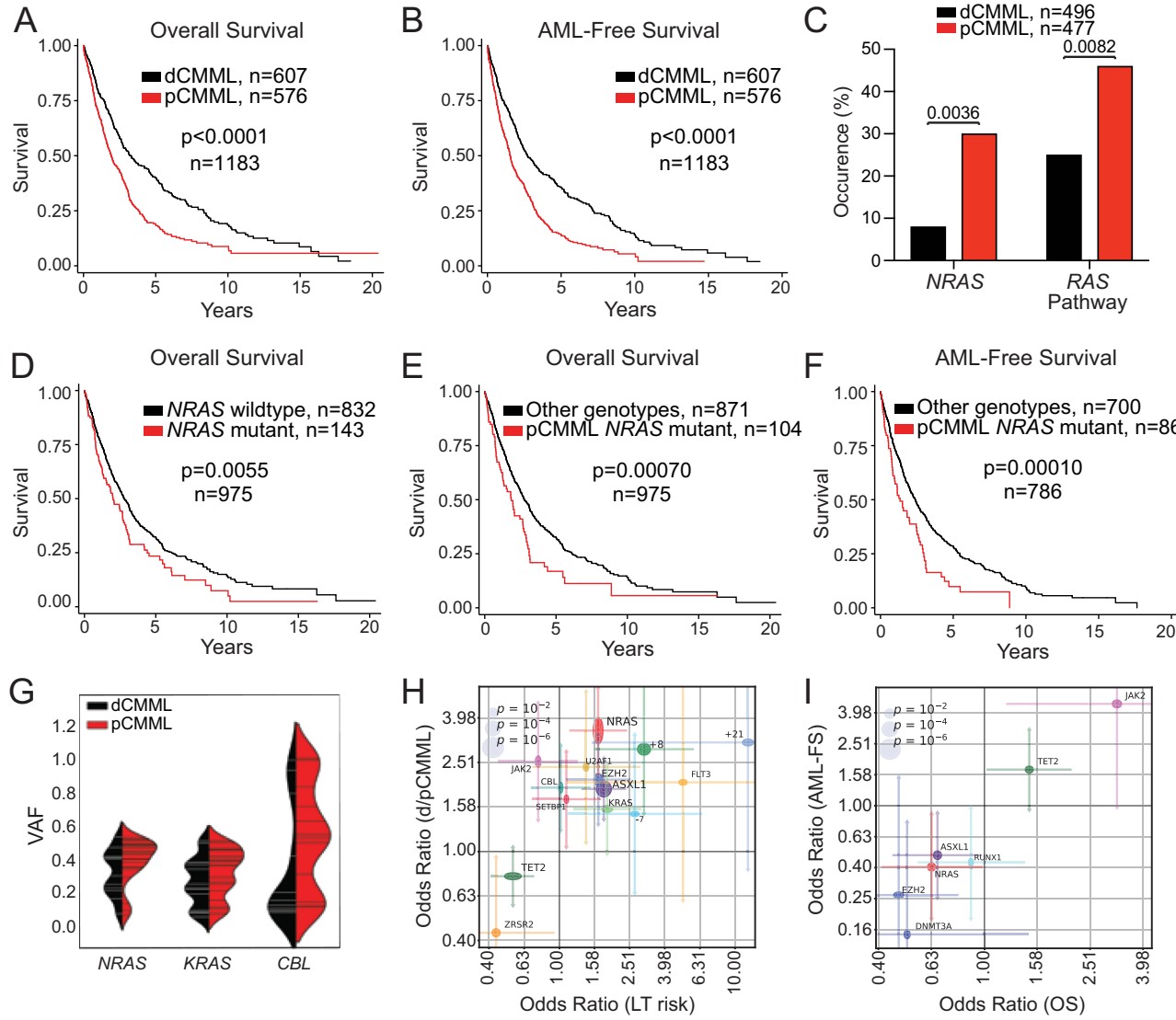

**Fig. 1 RAS pathway mutations correlate with WHO-defined proliferative chronic myelomonocytic leukemia (pCMML). A**. Kaplan–Meier curve depicting overall survival (OS) of CMML patients from the Mayo-GFM-Austrian cohort stratified by dCMML and pCMML subtypes. **B** Kaplan–Meier curve depicting AML-free survival. **C** Prevalence of *NRAS* and RAS pathway mutations in dCMML and pCMML by next generation sequencing. **D** Kaplan–Meier curve depicting OS of CMML patients stratified by *NRAS* wild type and *NRAS* mutant cases. **E** Kaplan–Meier curve depicting OS in pCMML patients with *NRAS* mutations relative to other CMML genotypes. **F** Kaplan–Meier curve depicting AML-free survival in pCMML patients with *NRAS* mutations relative to other CMML genotypes. **G** Violin plots representing variant allele frequencies (VAFs) of the most frequent RAS pathway mutations in dCMML and pCMML. VAF is depicted on the y-axis. Width of horizontal hatches correlates to number of samples with the indicated VAF. **H, I** Odds ratios of genetic factors selected by L1-regularized logistic regression model that have the greatest impact on clinical parameters. **H** Impact on dysplastic/proliferative CMML categorization and leukemic transformation (LT) risk. The x-axis represents odds ratios of LT risk. The y-axis represents categorization of low or high white blood cell (WBC) count, or pCMML vs dCMML classification. **I** Impact on AML-free survival (LFS) and OS. The x- and y-axes represent odds ratios of binarized OS and LFS values, respectively (by the median). The horizontal and vertical size of ellipses reflects corresponding p-values. Confidence intervals ($a = 0.05$) of odds ratio values are presented. p-values are indicated with each comparison.

Paired whole-exome sequencing (WES) performed on BM or PB mononuclear cells (MNC) collected at both chronic phase CMML and sAML stages in a cohort of 48 patients were analyzed (Supplementary Table 3). Median OS was 25 months after diagnosis and 6.4 months after LT. At CMML diagnosis, or first referral to study institution, mutations in driver genes (detected in every patient; mean, 3.6 per patient) and somatic copy number alterations (SCNA; Fig. 2A, B and Supplementary Fig. 2A) were comparable to previous reports[6]. Some mutations, such as in *TET2*, *SRSF2*, and *ASXL1*, were predominantly clonal while others, including the oncogenic RAS pathway, such as *NRAS*, *KRAS*, and *CBL*, showed a bimodal distribution of VAF (Fig. 2C).

Compared to chronic phase CMML, additional driver mutations (median = 2) were detected in 44% of LT samples (Supplementary Fig. 2B–E). The most striking changes included an increase in mutations of the oncogenic RAS pathway (from 52 to 67%), and in SCNA (from 52 to 75%; Fig. 2A, B and Supplementary Fig. 2).

Tracing clonal evolution from CMML onset to LT, we categorized driver mutations (Table S4 for definition of driver mutations) into those detected in CMML and at LT (primary drivers or 1d), only in LT (secondary drivers, 2d) and those in CMML and lost in LT, suggesting sub clonal secondary drivers (3d; Supplementary Fig. 2D). Secondary drivers (2d and 3d) were

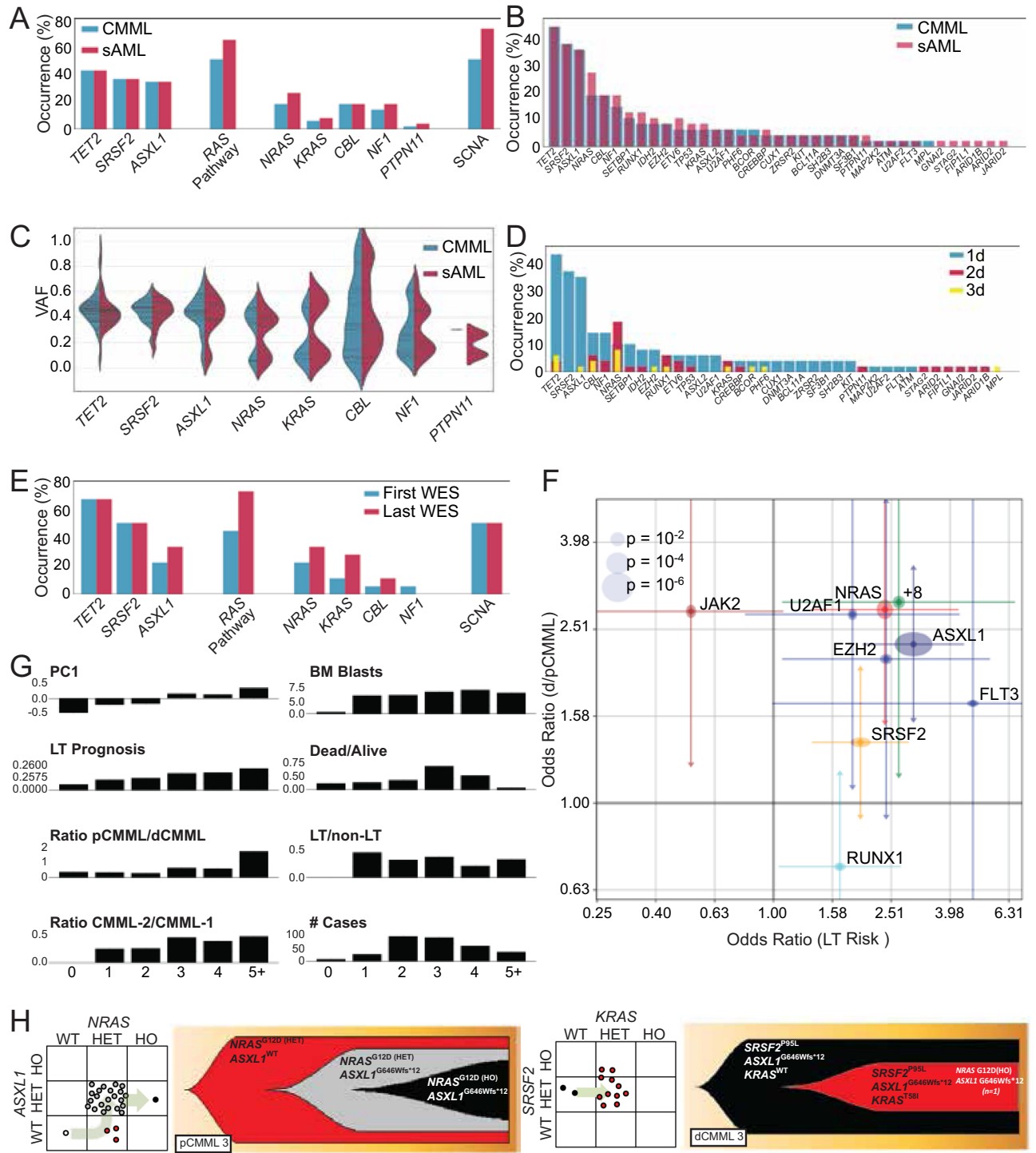

**Fig. 2 Alterations in driver mutation profiles delineate stages of CMML progression. Whole exome analysis of 48 paired CMML and secondary AML (sAML) samples. A** Prevalence of driver mutations and somatic copy number alterations (SCNAs) in CMML and sAML. RAS pathway represents all mutations in MAPK pathway. **B** Frequency of driver mutations and SCNAs in CMML and sAML. **C** Violin plots depicting variant allele frequencies (VAFs) for the most frequent driver mutations in CMML and sAML. **D** Frequency of driver mutations and SCNAs in CMML and sAML and their categorization as 1d, 2d, or 3d. **E** Genetic profiling of CMML evolution in eighteen patients at two time points by whole exome sequencing. **F** Odds ratios of genetic factors selected by the L1-regularized logistic regressions that have the greatest impact on dysplastic/proliferative CMML categorization and leukemic transformation (LT) risk. The x-axis represents odds ratios of LT risk (high or low). The y-axis represents categorization of low or high white blood cell (WBC) count, or pCMML vs dCMML classification. The horizontal and vertical size of ellipses reflects corresponding p-values calculated by two-tailed Fisher's exact test. Additionally, confidence intervals ($a = 0.05$) of odds ratio values are presented. The mean is the measure of center. **G** Stratification of parameters associated with aggressive disease phenotype by number of driver mutations. **H** Fish plots derived from patient-derived single colony assays of representative cases of pCMML (left) and dCMML (right). Scatter plot to the left of each fish plot represents the genotype of individual colonies as determined by Sanger sequencing of the indicated genes. RAS mutation status is depicted on the x-axis. The arrow indicates inferred evolutionary trajectory from which the fish plot was derived.

significantly different from primary drivers (1d), i.e. average number of mutations per category (1d: 3.3; 2d: 0.8; 3d: 0.4) and the spectrum of mutated genes were distinct. *TET2, ASXL1, SRSF2* mutations as well as monosomy 7 were almost exclusively 1d. *NRAS* was the main gene whose mutation rate increased from chronic phase to LT (Fig. 2D). The fraction of *NRAS* mutations detected in LT was significantly higher than that detected in CMML at diagnosis ($p = 0.0002$; Fig. 2D and Supplementary Fig. 2C). In 27% of cases, driver mutations were lost between chronic phase and LT. Eliminated sub clones harbored fewer driver mutations than progressive sub clones (19 vs 38; binomial test, $p = 0.01$) (Supplementary Fig. 2E).

In the WES cohort, CMML patients without RAS pathway mutations had a better OS, in comparison to those with RAS pathway mutations ($p = 0.045$; Supplementary Fig. 2F). WES of serial samples collected along disease evolution from 18 CMML patients (data set previously published[6]) showed time-dependent accumulation of driver mutations affecting RAS pathway genes, including *NRAS, KRAS,* and *CBL* (Fig. 2E). In addition, in the combined WES cohort, *NRAS* mutations were most significantly associated with aggressive disease features, such as leukocytosis (OR = 3.0, CI: 1.7–5.1) and LT risk (OR = 2.7, CI: 1.4–5.3; Fig. 2F), with the number of driver mutations correlating with disease severity in CMML (Fig. 2G).

Of note, targeted sequencing of patient-derived progenitor colony forming assays indicated that mutations can accumulate in multiple orders, e.g. mutations in *ASXL1* or *SRSF2* can appear after those in *NRAS* or *KRAS*, which was the case in pCMML (Fig. 2H). Altogether, these analyses in independent patient cohorts indicate that enrichment in oncogenic RAS pathway mutations is typically associated with the pCMML phenotype, aggressive disease features and with transformation to sAML.

**Clonal *NRAS/KRAS* activation drives a proliferative CMML phenotype.** To explore the impact of RAS pathway mutations in CMML, we used patient-derived cells to perform progenitor colony forming assays. Transduction of *NRAS* wild-type dCMML MNC with a *NRAS*-expressing adenovirus resulted in significant increases in colony formation relative to empty vector controls (Fig. 3A). Electroporation of primary CMML samples was highly efficient based on green fluorescent protein controls (Supplementary Fig. 3A). *NRAS*^G12D transfection in *NRAS* wild-type CMML MNC increased ex vivo proliferation whereas knockdown of *NRAS* in *NRAS*^G12D MNC demonstrated the opposite effect (Fig. 3B). Knockdown of *NRAS* was highly effective as confirmed by Western blot (Supplementary Fig. 3B). Similarly, *KRAS*^G12D transfection in *KRAS* wild-type CMML MNC promoted cell growth and *KRAS*^G12D knockdown in *KRAS*^G12D mutant CMML MNC (Supplementary Fig. 3C, D) lowered proliferation, indicating that the growth effects extended to other RAS pathway mutations. Finally, knockdown of *JAK2* in *JAK2* mutant, *NRAS* wild-type pCMML MNC resulted in significant reduction in cell proliferation, confirming its role in the proliferative phenotype in a subset of patients (Supplementary Fig. 3E).

Since RAS pathway mutations can also be seen in dCMML, we first demonstrated a clear correlation between *NRAS* mutant pCMML with higher risk phenotypic features and an inferior AML-FS, in comparison to *NRAS* mutant dCMML (Supplementary Fig. 1B, E). We then demonstrated that knockdown of *NRAS*^G12D in *NRAS* mutant dCMML MNC did not have an impact on ex vivo proliferation (Supplementary Fig. 3F). Next, to better define clonal versus sublonal RAS pathway mutation thresholds, we assessed mutant *NRAS* and mutant oncogenic RAS pathway VAF data and correlated RAS VAF as a continuous measure of subclonality in relationship to clinical parameters. A

receiver operating characteristic curve (ROC) analysis demonstrated that a *NRAS* VAF ≥ 0.3 or an oncogenic RAS VAF of ≥0.19 best correlated with a pCMML phenotype, in comparison to binary RAS mutation status (Fig. 3C–F). Finally, we showed that the *NRAS* VAF threshold of ≥0.3 correlated with lower platelet counts (Supplementary Fig. 3G), while the RAS VAF threshold of ≥0.19 correlated with lower platelet counts and higher PB blasts (Supplementary Fig. 3H, I); with thrombocytopenia and PB blasts being markers of higher risk disease[10].

To determine the in vivo effect of *Nras*^G12D on hematopoiesis, a genetically engineered *Vav-Cre-Nras*^G12D mouse model on a C57Bl/6 background was developed[13]. This was developed over the existing *Mx1-Cre-Nras*^G12D model given the mice develop a myeloproliferative disorder, but eventually die of a spectrum of hematological malignancies[14]. The *Mx1-Cre* transgene is active in the liver and hematopoietic cells after induction. Due to activity in the liver, mice develop hepatic histiocytic infiltrates and other phenotypes not specific to CMML. With *Cre* recombinase under control of the *Vav* promoter, *Nras*^G12D is more specifically induced in the hematopoietic system and phenocopies pCMML. At 6 weeks, these mice demonstrated a significant increase in PB and BM monocyte counts, without significant changes in absolute neutrophil counts (Fig. 3G). Analysis of ERK phosphorylation in BM progenitor cells (Lin−,c-kit+) suggested increased sensitivity to GM-CSF relative to controls, previously described in human CMML (Fig. 3H)[15]. Histopathologic analyses showed BM hypercellularity with megakaryocytic atypia/dysplasia and splenic red pulp hyperplasia, architectural distortion, and splenic extramedullary hematopoiesis (Fig. 3I and Supplementary Fig. 3J). BM MNC obtained from *Vav-Cre-Nras*^G12D animals formed increased colony numbers versus control mice (Fig. 3J). Finally, *Vav-Cre-Nras*^G12D mice had significantly shortened survival relative to control mice (median 400 days vs not reached; Fig. 3K). These data demonstrate that *Nras*^G12D alone expressed in *Vav* + hematopoietic cells is sufficient to initiate CMML-like disease.

**RAS pathway mutations drive the enrichment in the mitotic checkpoint kinase *PLK1* in pCMML.** To define the mechanism underlying the biological effects of RAS pathway mutations in CMML, transcriptome sequencing (RNA-seq) was performed. We attempted CD34 + selection from PB MNC prior to RNA-seq, however, there was significant attrition in cell numbers resulting in sample inadequacy to carry out large scale experiments. A pilot verification study analyzed RNA-seq data from 5 CMML patients using unsorted PB MNC, 5 using CD14 + sorted PB cells (monocytes) and 5 using CD34 + selected PB progenitor cells (no overlap between groups). This demonstrated a linear correlation in pair-wise gene expression (Supplementary Fig. 4A). Furthermore, comparing expression of housekeeping genes and select genes relevant to myeloid biology in CD34 + sorted and unsorted MNCs revealed no appreciable systematic differences between the two populations for these genes (Supplementary Fig. 4B). Finally, differential gene expression between sorted and unsorted samples identified very few significant differences in the transcriptome (Supplementary Fig. 4C). Taken together, this supports the use of unsorted MNC for transcriptome analysis in our study.

Thus, RNA-seq was performed on PB MNC from 25 CMML patients; 12 pCMML with RAS pathway mutations; and 13 dCMML without RAS pathway mutations. RAS pathway-mutated samples demonstrated a unique gene expression profile with 3729 upregulated and 2658 downregulated genes versus RAS pathway wild-type samples (Fig. 4A, B and Supplementary Data 1). Pathway analysis (ingenuity) identified cascades preferentially

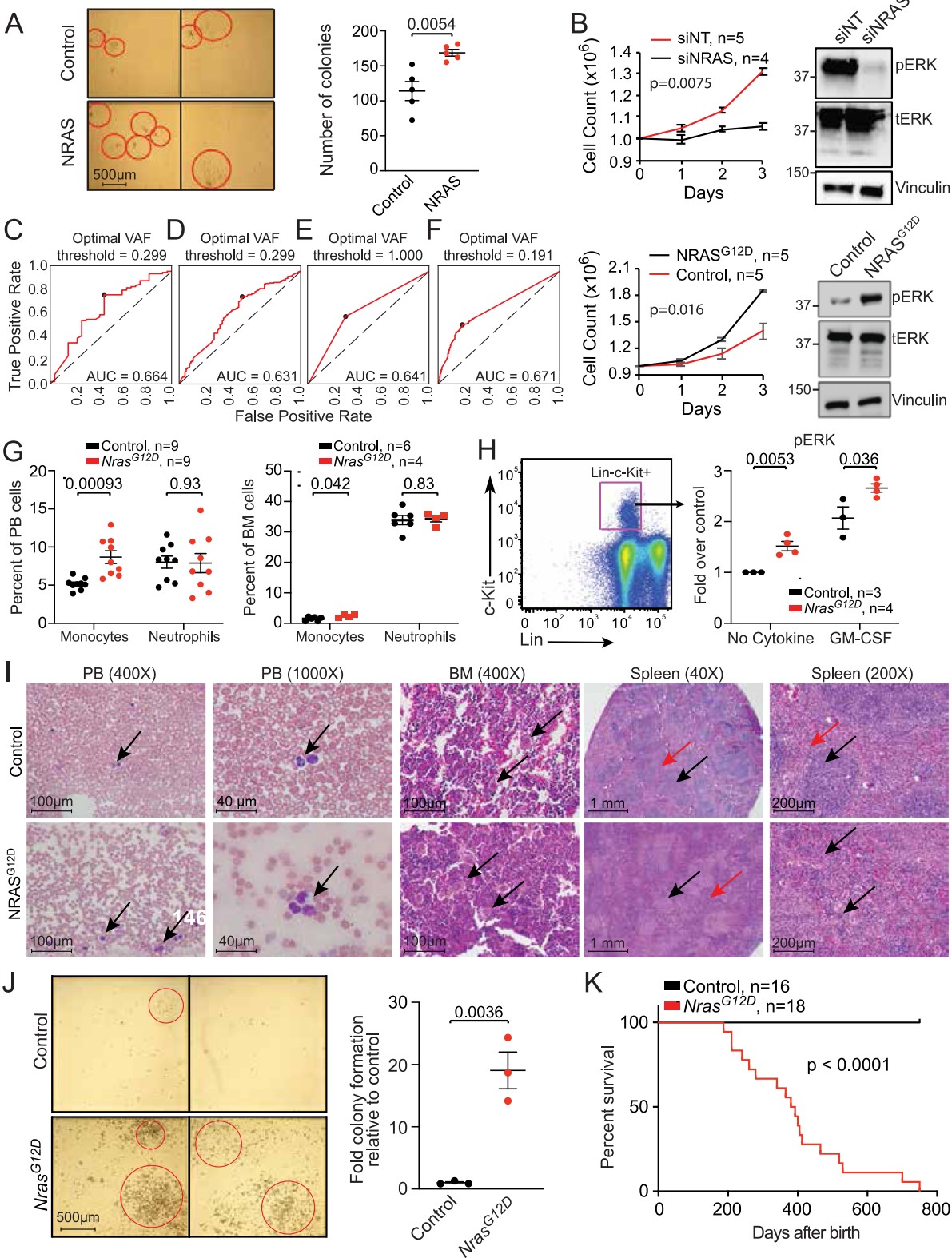

expressed in RAS pathway-mutated pCMML as (1) mitotic cell cycle process (e.g. *WEE1, PLK1, PLK2*), (2) cell cycle G2/M checkpoint regulation (e.g. *WEE1, AURKA, CHEK1*), (3) chromosomal segregation (e.g. *CDC45, RPA3*), and (4) mitotic cell cycle phase transition (e.g. *PLK1*; Supplementary Fig. 4D). Focusing on protein coding, targetable genes downstream of the oncogenic RAS pathway, we selected *PLK1*, a conserved kinase

involved in the regulation of cell cycle progression for further mechanistic studies[16,17]. Although *PLK1* was not the top upregulated gene in our data set (Fig. 4C), the rationale for further studying *PLK1* included, (a) prior genome-wide RNAi screen that had demonstrated that RAS mutant cells were hypersensitive to PLK1 inhibition[18], (b) data demonstrating that RAF1/CRAF localize to the mitotic spindle of proliferating tumor

**Fig. 3** *NRAS*[G12D] **mutation drives a proliferative CMML phenotype. A**. Representative hematopoietic progenitor colony formation assay using *NRAS*-wild-type CMML patient-derived mononuclear cells (MNCs) after transduction with either a null adenoviral construct (control) or NRAS-expressing vector (NRAS). Red circles indicate individual colonies. Scatter plot depicts colony counts at day 12 after inoculation in five individual cases. **B** Daily cell counts of CMML patient-derived MNCs after siRNA depletion of *NRAS* in *NRAS*[G12D] cells (above) and transfection of *NRAS*[G12D] in *NRAS* wild-type cells (below). Representative western blots of three experiments are depicted for validation by assessing phosphorylated ERK (pERK) relative to total ERK (tERK) levels with Vinculin as a loading control. Knockdown of *NRAS* by qPCR was ≥85%. Indicated *p*-value in panels **A** and **B** by two-tailed Student's *t* test. **C–F** Receiver operating characteristic curve (ROC) analyses illustrating the diagnostic ability of variant allele frequency (VAF) of *NRAS* mutations (**C**) or oncogenic RAS pathway mutations (*NRAS, KRAS, CBL, PTPN11*) (**D**) at discrimination between pCMML and dCMML phenotypes. If >1 mutation was present in the sample, maximal VAF was used. Impact of $VAF_{RAS}$ as a predictor of pCMML vs dCMML phenotype, using binary RAS mutation status (**E**), or a continuous $VAF_{RAS}$ value (**F**). Characterization of the *Vav-Cre-Nras*[G12D] mouse model of CMML is depicted at 6 weeks of age (panels **G–I**) and when moribund (panels **J** and **K**). **G** Differences in peripheral blood (PB, left) and bone marrow (BM, right) monocytes and neutrophils in *Nras*[G12D] mice relative to wild-type controls. **H** BM cells were serum- and cytokine-starved for 2 h at 37 °C. Cells were then stimulated with or without 2 ng/ml of mGM-CSF for 10 min at 37 °C. Levels of phosphorylated ERK1/2 (pERK) were measured using phosphor-flow cytometry. Myeloid progenitors are enriched in Lin[-/low] c-Kit[+] cells. Indicated *p*-value in panels **G** and **H** by two-tailed Student's *t* test. **I** Histopathologic comparisons between wild-type control (top) and *Nras*[G12D] (bottom) mouse PB, BM, and spleen. PB smears depict monocytes (arrows). BM shows normal hematopoiesis (arrows, above) and megakaryocytic atypia and hyperplasia (arrows, below). Lower power spleen reveals normal white and red pulp (black and red arrows respectively, above) with effacement of this architecture (black and red arrows, below). High power spleen reveals normal white and red pulp architecture (black and red arrows respectively, above) and dysplastic megakaryocytes (arrows, below). **J** Representative hematopoietic progenitor colony formation assay using MNCs from wild-type control (top) and *Nras*[G12D] (bottom) mice. Bar graph depicts colony counts at day 12 after inoculation. Indicated *p*-value by two-tailed Student's *t* test. **K** Kaplan–Meier curve demonstrating overall survival of *Nras*[G12D] mice relative to wild-type controls. Data in panels **A**, **B**, **G**, **H** and **J** are presented as mean ± SEM. The indicated n represents the number of biologic replicates. *P*-values are indicated with each comparison. Source data are provided as a Source Data file.

cells, with RAF1 specifically associating with PLK1 at the centrosomes and spindle poles during G2/Mitosis, (c) the fact that Inhibition of RAF1 phosphorylation impaired PLK1 activation[19], and (d) the PLK1 inhibitor volasertib had already completed extensive pre-clinical and clinical trial testing, with the phase 3 trial in AML providing extensive safety and efficacy data.

Using qPCR on several CMML patient samples, we validated a statistically significant increase in expression of *PLK1* in *NRAS* mutant pCMML ($n = 21$), in comparison to *NRAS* wild-type dCMML ($n = 14$), *NRAS* mutant dCMML ($n = 4$), and *JAK2* mutant pCMML ($n = 9$; Fig. 4D); indicating that overexpression of *PLK1* was specific to RAS mutant pCMML. Overexpression of *PLK1* was found in CD14 + sorted PB MNC in RAS mutant pCMML relative to RAS wild-type dCMML and normal controls (Supplementary Fig. 4E). Further, we demonstrated a significant positive correlation between *PLK1* expression and *NRAS*[G12D] VAF, in *NRAS* mutant pCMML samples (Fig. 4E). Knockdown of *NRAS* in *NRAS*[G12D] mutant CMML MNC resulted in reduced *PLK1* expression relative to controls (Fig. 4F). Conversely, transfection of *NRAS*[G12D] into RAS wild-type CMML MNC induced upregulation of *PLK1* (Fig. 4G). Similar results were obtained with mutant *KRAS*[G12D] (Fig. 4H, I), indicating that this regulation extended to other RAS pathway mutations. These findings were confirmed at the protein level (Supplementary Fig. 4F). We then demonstrated that the association between RAS and *PLK1* was clonal RAS mutation specific, by showing that knockdown down of *JAK2* in *JAK2*[V617F] mutant/RAS wild-type pCMML MNC, and *NRAS* in *NRAS*[G12D] mutant dCMML MNC each, had no impact on *PLK1* expression (Supplementary Fig. 4G, H)[20]. Notably, the mean *NRAS* VAF in pCMML samples used for these experiments was 38% versus 15% for dCMML samples, suggesting that the differences in mitotic kinase expression with *NRAS* knockdown may be related to the clonal versus subclonal nature of RAS mutations. Together, these results suggest that acquisition of clonal oncogenic RAS pathway mutations upregulate *PLK1*; a phenomenon specific to clonal RAS mutant pCMML.

**Mutant RAS regulates *PLK1* expression through the lysine methyltransferase KMT2A (MLL1).** Interrogating molecular events involved in RAS-mediated increase in *PLK1* expression, ChIP-seq was performed on BM MNC from 40 CMML patients [RAS pathway-mutated pCMML, $n = 18$; RAS pathway wild-type

dCMML, $n = 12$) and healthy, age-matched controls ($n = 10$). From this data pool we selected 6 samples [3 RAS pathway mutant pCMML (#1 *TET2/DNMT3A/NRAS*, #2 *ASXL1/CBL/IDH2/SRSF2* and #3 *ASXL1/TET2/EZH2/NRAS*) and 3 RAS-pathway wild-type dCMML (#1 *TET2/SRSF2*, #2 *IDH2/SRSF2/RUNX1* and #3 *TET2/SRSF2*)] that had adequate number of reads and read depth and that met ENCODE criteria (https://www.encodeproject.org/chip-seq/histone/) for assessment of the three aforementioned histone marks. A peak was defined as either present (MACS2 FDR < 0.05 for a given peak) or not present (MACS2 $p \geq 0.05$ for a given peak). When we compared the differences between the mean number of peaks between RAS mutant pCMML and RAS wild-type dCMML for all three histone marks, there were no statistically significant differences in these six samples (Fig. 5A). However, this analysis does not leverage the confidence of the presence of a given peak across replicates (MACS2 FDR). Therefore, a differential binding analysis (Diff-Bind) was performed, demonstrating numerically more H3K4me1 peaks in pCMML vs dCMML (Fig. 5B). Metagene analysis similarly demonstrated an increase in the number of differentially enriched H3K4me1 peaks in RAS mutant pCMML relative to RAS wild-type dCMML (Supplementary Fig 5A). Comparisons of overexpressed genes in RAS mutant pCMML with gene promoter enrichment for H3K4me1 peaks demonstrated significant overlap (Fig. 5C and Supplementary Data 2), with *PLK1* identified in this group.

Sequence-specific analysis on these six samples also found enrichment of H3K4me1 at the promoter (conservatively defined by a 2Kb region from the transcription start site) of *PLK1* (Fig. 5D), with a statistically significant increase in measured H3K4me1 peak height in RAS mutant pCMML vs RAS wild-type dCMML.

We then confirmed these findings with *NRAS* and *KRAS* knockdowns that decreased H3K4me1 occupancy at the *PLK1* promoter site in *NRAS/KRAS* mutant pCMML MNC, relative to controls (Fig. 5E, F). Conversely, *NRAS*[G12D]/*KRAS*[G12D] transfection of *NRAS/KRAS* wild-type CMML MNC resulted in an increased occupancy of H3K4me1 at *PLK1* the promoter (Fig. 5G, H). In contrast, *JAK2*[V617F] knockdown in *JAK2* mutant/RAS wild-type pCMML and *NRAS*[G12D] knockdown in *NRAS* mutant dCMML MNC had no effect on H3K4me1 occupancy (Fig. 5I, J); underscoring the specificity of this axis to clonal RAS pathway

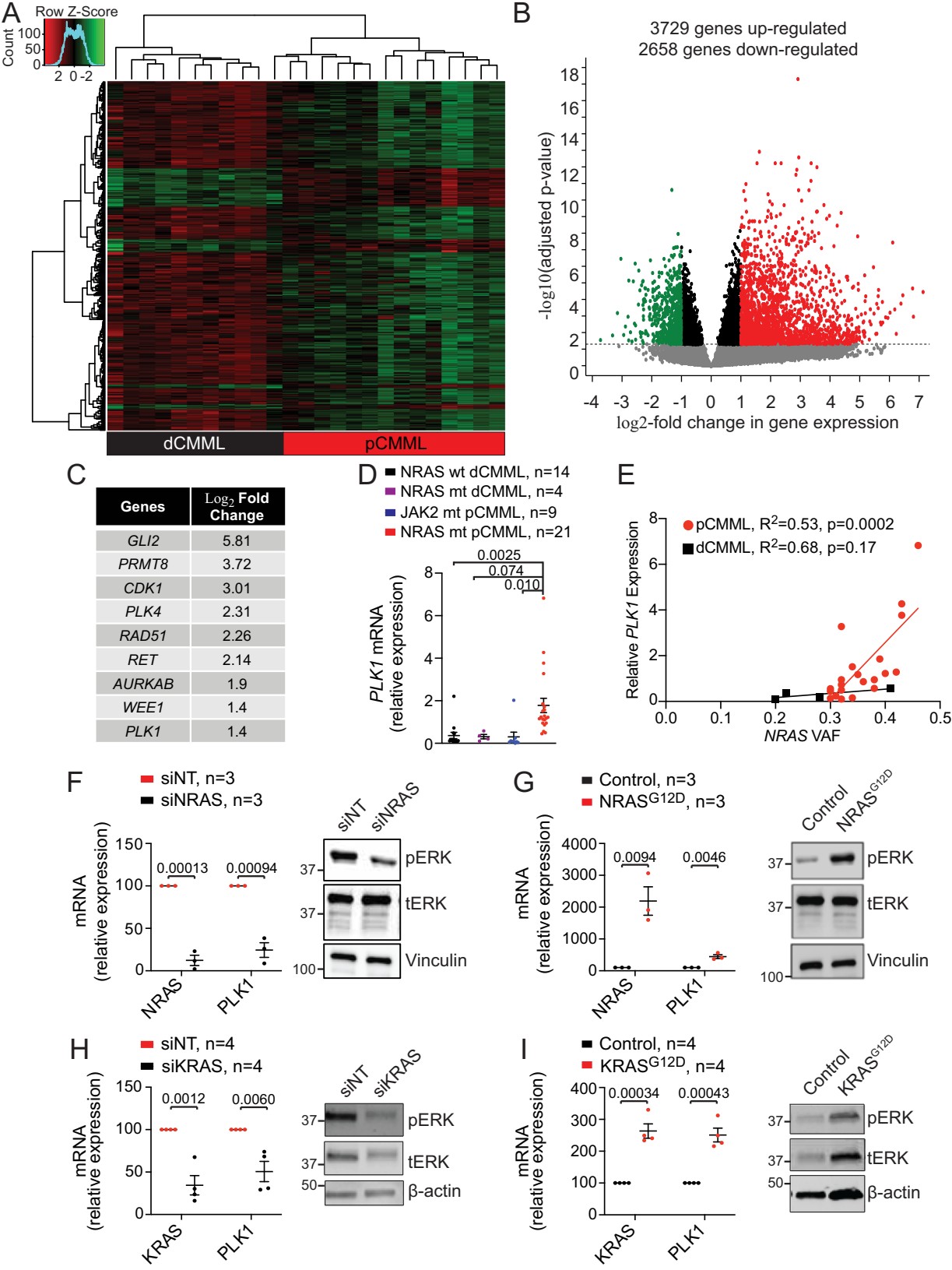

mutations despite *JAK2* representing an alternative driver of the pCMML phenotype. Thus, oncogenic RAS signaling enhances expression of *PLK1* in association with enrichment of H3K4me1 at its promotor.

Since H3K4me1 is carried out by the lysine methyltransferases KMT2A-D, SET1A and SET1B[21], we focused on these entities.

RNA-seq analysis detected overexpression of *KMT2A* (*MLL1*) and its coactivator *MEN1* in RAS mutant pCMML samples versus controls (Fig. 6A), with no statistical difference between RAS mutant pCMML and RAS wild-type dCMML. Unlike in AML, where *KMT2A* fusions and partial tandem duplications (PTD) are frequent, FISH (fluorescence in situ hybridization) and WES

**Fig. 4 RAS pathway mutations drive expression of mitotic checkpoint kinase *PLK1*. A** Unsupervised hierarchical clustering of RNA-seq performed on peripheral blood samples from 35 CMML patients. Cluster 1 (black) with RAS wild-type dCMML cases and Cluster 2 (red) with RAS mutant pCMML cases. **B** Volcano plot demonstrating differentially upregulated (red) and downregulated (green) genes in RAS mutant pCMML (vs RAS wild-type dCMML) expressed as log2-fold change. Indicated *p*-values on the *y*-axis is calculated by the Wald test and false discovery rate-corrected. **C** Table of most upregulated therapeutically actionable genes in pCMML relative to dCMML. **D** Quantitative PCR (qPCR) validation of *PLK1* expression in patient-derived MNCs from *NRAS* wild-type (wt) dCMML, *NRAS* mutant (mt) dCMML, *JAK2* mt pCMML, and *NRAS* mt pCMML cases. **E** Scatter plot comparing relative *PLK1* expression to variant allele frequency (VAF) of *NRAS* mutation. **F** qPCR for *NRAS* and *PLK1* in *NRAS* mutant pCMML MNCs after transfection with non-targeting siRNA (siNT), or siRNA against *NRAS* (siNRAS). **G** qPCR for *NRAS* and *PLK1* in *NRAS* wild-type dCMML patient-derived MNCs after transfection with an empty vector (Control) or *NRAS*^G12D. **H** qPCR for *KRAS* and *PLK1* in *KRAS* mutant pCMML MNCs after transfection with siNT or siKRAS. **I** qPCR for *KRAS* and *PLK1* in *KRAS* wild-type dCMML MNCs after transfection with an empty vector (Control), or *KRAS*^G12D. Western blots in panels **F**–**I** are representative of three experiments. Data in panels **D** and **F**–**I** are presented as mean ± SEM. Indicated *p*-value in panels **D**–**I** by two-tailed Student's *t* test. The indicated *n* represents the number of biologic replicates. Source data are provided as a Source Data file.

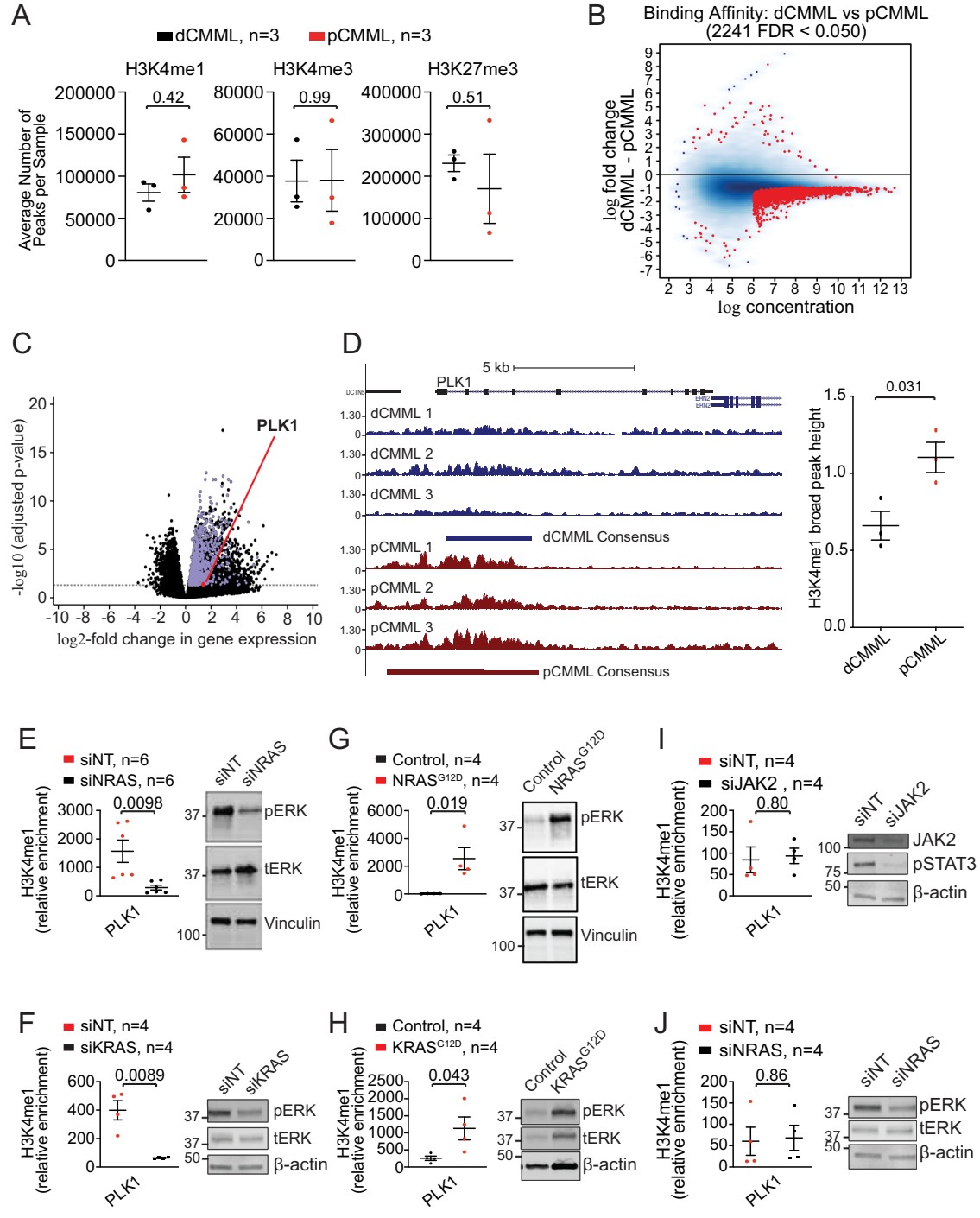

**Fig. 5 Genome-wide and sequence-specific enrichment of the H3K4me1 histone mark in RAS mutant pCMML. A** ChIP-seq on BM MNC from CMML patients assessing relative global enrichment of histone 3 lysine 4 monomethylation (H3K4me1), trimetylation (H3K4me3), and histone 3 lysine 27 trimethylation (H3K27me3). Indicated *p*-value by two-tailed Student's *t* test. **B** MA plot of differential binding analysis of ChIP-seq H3K4me1 peak data. Each point represents a binding site with each red data point representing sites identified as differentially enriched. **C** Volcano plot integrating differential enrichment of H3K4me1 at gene promoters in RAS mutant pCMML (vs RAS wild-type dCMML) with differentially upregulated gene expression (purple) in RAS mutant pCMML expressed as log2-fold change. *PLK1* is indicated as a red data point. Indicated *p*-values on the *y*-axis is calculated by the Wald test and false discovery rate-corrected. **D** Enrichment and consensus peak calling for H3K4me1 occupancy at the *PLK1* locus. Three individual traces from RAS wild-type dCMML (blue, above) and RAS mutant pCMML (red, below) patients are depicted. Relative localization along the *PLK1* gene body is depicted. The broad consensus peaks are indicated below each set of patient samples. The accompanying box plot indicates differences in H3K4me1 consensus peak height between pCMML and dCMML samples. Indicated *p*-value by two-tailed Student's *t* test. **E–J** ChIP-PCR of CMML MNCs assessing *PLK1* promoter occupancy of H3K4me1. This was done in *NRAS* mutant (**E**) or *KRAS* mutant (**F**) MNCs after transfection with siNT or siNRAS/siKRAS. Western blots are representative validation of RAS knockdown. It was also performed in *NRAS* (**G**) or *KRAS* (**H**) wild-type MNCs after transfection with a control vector or *NRAS*[G12D]/*KRAS*[G12D]. Western blots are representative validation of RAS knockdown. **I** ChIP-PCR was performed in *JAK2* mutant MNCs after transfection with siNT or siJAK2. Validation of JAK2 knockdown is done using phosphorylated STAT3 (pSTAT3) levels. **J** ChIP-PCR was also performed in *NRAS* mutant dCMML MNCs after transfection with siNT or siNRAS. Western blots in panels **E–J** are representative of three experiments. Indicated *p*-value in panels **E–J** by two-tailed Student's *t* test. Data in panels **A**, **D**, and **E–J** are presented as mean ± SEM. The indicated *n* represents the number of biologic replicates. Source data are provided as a Source Data file.

analyses in 500 CMML samples identified only a single patient with a *MLLT3-KMT2A* fusion (Fig. 6B), without any *KMT2A* mutations or PTD[22,23]. These data suggest that wild-type KMT2A could drive oncogenic activity in RAS mutant pCMML. ChIP-PCR studies in *NRAS* mutant pCMML MNC detected occupancy of *KMT2A* at the *PLK1* promoter (Fig. 6C). Knockdown of *NRAS*[G12D]/*KRAS*[G12D] in *NRAS/KRAS* mutant pCMML MNC decreased KMT2A occupancy at the *PLK1* promotor relative to controls (Fig. 6D, E). Interestingly, knockdown of *NRAS*[G12D] in *NRAS*[G12D] mutant dCMML and *JAK2*[V61F] in *JAK2*[V617F] mutant pCMML MNC did not result in significant changes in KMT2A occupancy at the *PLK1* promotor (Supplementary Fig. 6A, B).

Conversely, transfection of *NRAS/KRAS* wild-type dCMML MNC with *NRAS*[G12D]/*KRAS*[G12D] enhanced KMT2A occupancy at the *PLK1* promoter (Fig. 6F, G). Knockdown of *KMT2A* in *NRAS* mutant pCMML MNC decreased H3K4me1 enrichment at the *PLK1* promotor, correlating with decreased *PLK1* expression (Fig. 6H, I). Further, *KMT2A* knockdown was also associated with a significant decrease in ex vivo cell proliferation in RAS mutant pCMML MNC in comparison to controls (Supplementary Fig. 6C). In addition, inhibition of the KMT2A-MEN1 interaction with Mi-503, an orally bioavailable small molecule inhibitor[24], similarly decreased enrichment of H3K4me1 at the *PLK1* promoter (Supplementary Fig. 6D). qPCR studies at day 4 after Mi-503 treatment confirmed decreased *PLK1* expression (Supplementary Fig. 6E). Since KMT2C and KMT2D are well-known mediators of H3K4me1, we knocked down *KMT2C* and *KMT2D* in *NRAS*[G12D] mutant pCMML MNC but found no changes in H3K4me1 enrichment at the *PLK1* promotor versus controls (Supplementary Fig. 6F). These data suggest that KMT2C and KMT2D do not play roles in modulating H3K4me1 in pCMML MNC. Together, these data support the hypothesis that *PLK1* expression in clonal *RAS* mutant CMML is dependent on an unmutated *KMT2A* as a mediator of oncogenic RAS signaling.

**Therapeutic efficacy of targeting PLK1 in pCMML.** Individual depletion of *PLK1* in patient-derived *NRAS* mutant pCMML cells (*n* = 5, mean VAF 35%) reduced cell growth relative to controls (Fig. 7A). PLK1 inhibition with Volasertib also efficiently reduced generation of progenitor colonies derived from *NRAS* mutant pCMML samples with greater efficacy compared to *NRAS* wild-type dCMML controls (Fig. 7B). IC$_{50}$ levels for volasertib were 29.9 nM in *NRAS* mutant pCMML and 0.14 M for *NRAS* wild-type dCMML.

Finally, *NRAS*[G12D] mutant pCMML-patient-derived xenografts (PDX) using NSG-S (NOD.Cg-*Prkdc*[scid] *Il2rg*[tm1Wjl]/SzJ-SGM3) mice[25] were treated with volasertib. Six-week-old NSG-S mice were sub-lethally radiated with 25 Gy of radiation and 18 h later were injected with 2 ×10$^6$ CMML patient-derived BM MNCs via their tail veins (Supplementary Fig. 7A). Four weeks later, after confirming engraftment by documenting hCD45 by flow cytometry, triplicate sets of mice were treated with volasertib 20 mg/kg/week administered intraperitoneally for 2 weeks versus a vehicle control. After 2 weeks of therapy, the volasertib and vehicle-treated mice were sacrificed, followed by flow cytometry and extensive histopathological analysis of blood, BM and spleen specimens (Fig. 7C–H). Immunohistochemistry indicated that neoplastic cells expressed hCD45, CD14, and CD163 while not expressing hCD3 (Fig. 7C, D). In volasertib-treated mice there was marked reduction in BM and splenic malignant monocytic/ histiocytic tumor infiltrates versus vehicle-treated controls (Fig. 7C, D). In addition, partial normalization of splenic follicular architecture (Fig. 7C) and partial restoration of normal hematopoietic islands in BM (Fig. 7D) were observed with volasertib versus vehicle. Findings were confirmed by carrying out flow cytometry with gating strategy depicted in Supplementary Fig. 7B. Analysis demonstrated complete clearance of human leukocytes (hCD45 vs mCD45.1) from the PB, BM, and spleens of volasertib-treated mice versus controls (Fig. 7E, F). Median spleen size and mass were also significantly lower in volasertib-treated mice compared to vehicle-treated controls (Fig. 7G, H). Consequently, these data support pharmacological inhibition of PLK1 as a potentially efficacious therapeutic strategy in RAS mutant pCMML.

**Discussion**

CMML is an aggressive hematological malignancy with dismal outcomes, with the pCMML subtype in particular having median OS of <2 years and high rates of AML transformation[10]. Given limited therapeutic options for affected patients, we carried out this study to define the genetic and epigenetic landscape of pCMML and identify therapies that could modify disease biology. Phenotypically related to pCMML, juvenile myelomonocytic leukemia (JMML) is a pediatric neoplasm often initiated by germline (*CBL, NF1, PTPN11*) or somatic (*NRAS, KRAS, CBL, RRAS*) RAS-activating mutations and is described as a bona fide RASopathy[26,27]. JMML outcome is associated with a dose-dependent effect for RAS pathway activation and accumulation of additional recurrent mutations in genes involved in the polycomb repressive complex 2 (PRC2-*ASXL1*) and gene transcription/

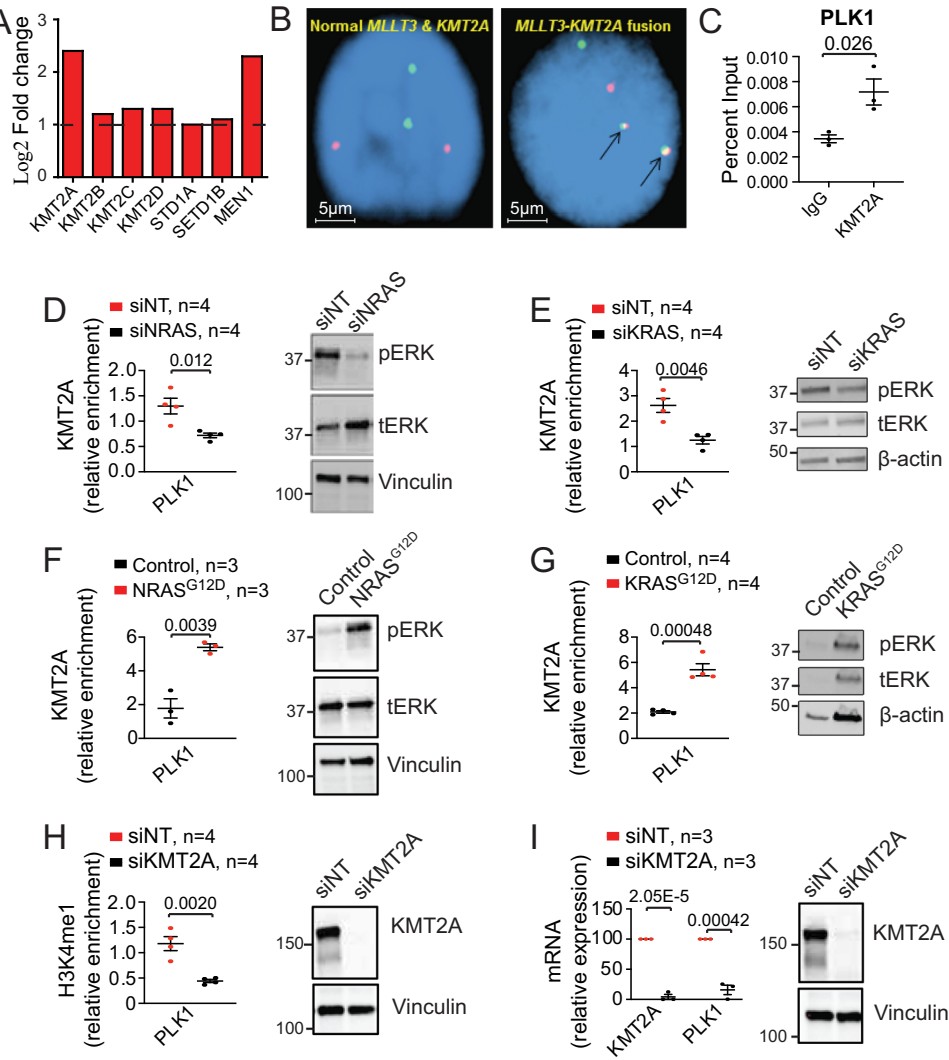

**Fig. 6 Mutant *RAS* regulates *PLK1* expression through the lysine methyltransferase KMT2A (MLL1). A** RNA-seq expression levels of enzymes potentially involved in monomethylation in pCMML (vs dCMML and healthy volunteers). **B** Representative fluorescence in situ hybridization (FISH) in two patients assessing for *MLLT3-KMT2A* fusions ($n = 500$ biological replicates). Arrows (right) indicate presence of fusion. **C** ChIP-PCR in *NRAS* mutant CMML patient-derived MNCs assessing KMT2A occupancy at the promoter of *PLK1* with immunoprecipitation of KMT2A relative to isotype control (IgG). **D, E** ChIP-PCR assessing KMT2A occupancy at the *PLK1* promoter in *NRAS* mutant (**D**) and *KRAS* mutant (**E**) CMML MNC after transfection with siNT or siNRAS/siKRAS. Western blots are representative validation of RAS knockdown from three experiments. **F, G** ChIP-PCR assessing KMT2A occupancy at the *PLK1* promoter in *NRAS* wild type (**F**), and *KRAS* wild type (**G**) CMML MNC, after transfection with control vector or *NRAS*^G12D/*KRAS*^G12D. Western blots are representative validation of RAS expression. **H, I** ChIP-PCR assessing H3K4me1 at the promoter of *PLK1* (**H**) and qPCR assessing levels of *KMT2A* and *PLK1* (**I**) in *NRAS* mutant CMML MNC after transfection with siNT or siKMT2A. Western blots are representative validation of KMT2A knockdown. Western blots in panels **D–I** are representative of three experiments. Data in panels **C–I** are presented as mean ± SEM and indicated *p*-values by two-tailed Student's *t* test. The indicated *n* represents the number of biologic replicates. Source data are provided as a Source Data file.

signaling (*SETBP1* and *JAK3*)[27–29]. Here, we show in pCMML that acquisition of RAS pathway mutations often occurs early (*NRAS*^G12D most frequent), commonly on a background of epigenetic (*TET2*) and splicing gene mutations (*SRSF2*) and correlates with WHO-defined prognostic factors including leukocytosis, BM blasts and LT to sAML; further implying that severe forms of this disease often seen in the elderly, are also RASopathies that, contrary to JMML, develop on a background of age-related clonal alterations (clonal hematopoiesis)[30]. Analysis of mouse models suggest that loss of *Tet2*, which induces hypermethylation of *Spry2* encoding a negative regulator of the RAS signaling pathway, results in a fully penetrant CMML phenotype when combined with *Nras*^G12D expression in mouse hematopoietic cells by synergistically enhancing MAPK signaling. These findings suggest that preexisting mutations in *TET2* may

favor expansion of cells that acquire a RAS pathway mutation[31]. Of note, hematopoietic progenitor colony hypersensitivity to GM-CSF, an established hallmark of JMML, has also been depicted in CMML, especially in RAS mutant pCMML[15,32] and was also seen in progenitor cells obtained from the *Vav-Cre-Nras*^G12D pCMML mouse model in this study.

WES studies have shown that, on average, CMML patients harbor between 10–15 somatic mutations per exon/coding region, similar to de novo AML and other myeloid malignancies[6,33]. Of these 10–15 somatic mutations, a mean number of three affect recurrently mutated genes. These genes (~40 have been identified) largely encode epigenetic regulators (*TET2*-60%, *ASXL1*-40%), proteins of pre-mRNA splicing complexes (*SRSF2*-50%) and members of signaling pathways (*NRAS* 30%; *JAK2* 10%)[10,34,35]. Prior clonal architecture studies at the single cell

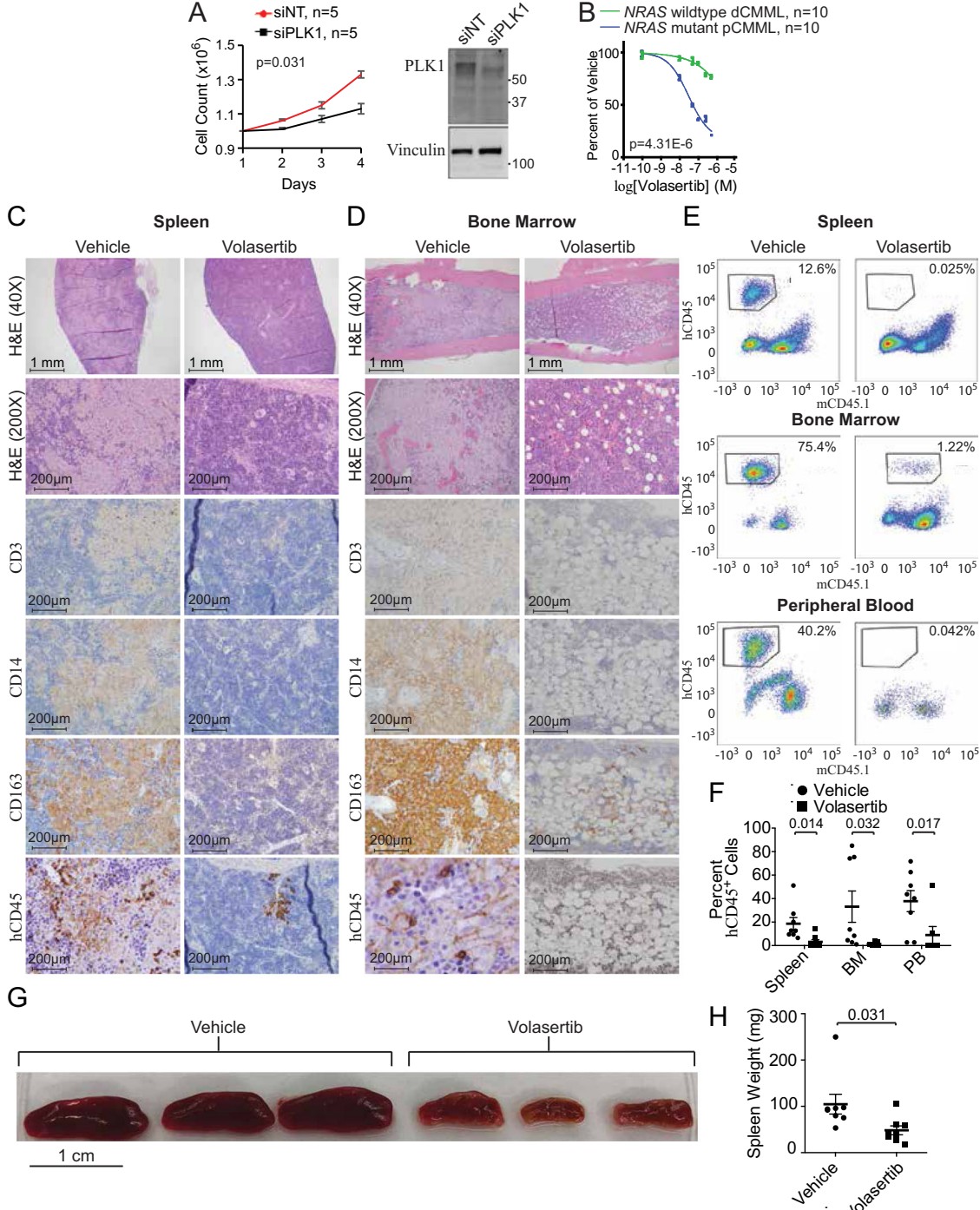

**Fig. 7 Therapeutic efficacy of targeting PLK1 in pCMML. A** Daily cell counts of CMML MNC after transfection with either siNT or siPLK1. Representative western blot depicts validation of PLK1 knockdown from three experiments. Knockdown of *PLK1* by qPCR was ≥90%. **B** Progenitor colony forming assay using CMML MNC with increasing doses of volasertib. Indicated *p*-value in panels **A** and **B** by two-tailed Student's *t* test. **C**, **D** Histopathologic analysis with H&E staining and immunohistochemistry (IHC) of spleen (**C**) and BM (**D**) of murine patient-derived xenografts (PDX) after treatment with vehicle control or volasertib. Magnification is ×200 unless otherwise indicated. hCD45 is human CD45. **E** Representative flow cytometry of spleen, BM and PB of PDXs after treatment with vehicle control (left) or volasertib (right). The *y*-axis indicates hCD45 expression status while *x*-axis indicates murine CD45 (mCD45). Percentages indicate proportion of hCD45+ and mCD45- cells in the respective tissues. **F** Flow cytometry of proportion of hCD45+ cells in spleen, BM, and PB of PDX mice. Data are presented as mean ± SEM from nine mice. **G**, **H** Effect of vehicle versus volasertib treatment on PDX spleen size (**G**) and weight (**H**). Data in panel H are presented as mean ± SEM from seven mice treated with vehicle and eight treated with volasertib. Indicated *p*-value in panels **F** and **H** by two-tailed Student's *t* test. Data in panels **A** and **B** are presented as mean ± SEM. The indicated *n* represents the number of biologic replicates. *p*-values are indicated with each comparison. Source data are provided as a Source Data file.

level have reported that in most adult onset hematological malignancies, including CMML, oncogenic RAS pathway mutations are usually secondary/subclonal events[6,36]. While that is true in dCMML, we show that in pCMML, RAS pathway mutations are often clonal and occur early in the disease course. With the help of a large molecularly annotated database we show that clonal RAS pathway mutations (VAF ≥ 0.19) were associated with high risk disease features and were associated with an inferior AML-FS (increased risk of LT). Interestingly, NRAS and PTPN11 are also among the recurrently mutated genes associated with LT in patients with myelodysplastic syndromes[37]. Of note, compared to chronic phase CMML, sAML demonstrated additional somatic mutations in only 44% of cases, indicating that LT can occur without additional driver mutations in the coding regions of DNA. Acquisition of RAS mutations correlated with WBC proliferation and blast cell accumulation, two main prognostic factors recognized by the WHO in CMML, suggesting that RAS mutations form the landscape for epigenetic and evolutionary events leading to AML transformation.

We show that acquisition of clonal RAS pathway mutations in CMML samples is associated with overexpression of PLK1. Polo-like kinases are involved in regulation of centrosome maturation, bipolar spindle formation, anaphase-promoting complex, and chromosome segregation[16]. Although this was not the top upregulated gene, given the specificity of PLK1 overexpression to clonal RAS mutant pCMML, along with prior work using a genome-wide RNAi screen that had shown sensitivity of RAS mutant cells to PLK1 inhibition and that RAF1/CRAF has shown to have direct interactions (MEK independent) with PLK1 at centrosomes and spindle poles during G2/mitosis, and the fact that phase III trials with PLK1 inhibitors (volasertib) had already been completed in AML, we chose to further assess the functional and therapeutic relevancy of this mitotic checkpoint kinase[18,19]. Our ChIP-seq/ChIP-PCR data demonstrated enrichment of H3K4me1, KMT2A, and MEN1 at the PLK1 promoter in RAS mutant pCMML. H3K4me1 is an activating histone configuration associated with increased gene transcription. While the entire KMT2 family (KMT2A-D, SET1A, and SET1B) can result in methylation of H3K4, only KMT2A and KMT2B do so using MEN1 as a coactivator[21]. KMT2C and KMT2D are reported to be enzymes usually responsible for H3K4 monomethylation, however, we detected no changes in H3K4me1 enrichment at the PLK1 promoter after knockdown of these lysine methyltransferases. Unexpectedly, while KMT2A can cause trimethylation of H3K4[21], using ChIP-seq, both globally and in a sequence-specific manner, we saw no difference in H3K4me3 enrichment between RAS pathway mutant and wild-type CMML samples, but instead note much higher global and local H3K4me1 at gene promoters in RAS pathway mutant samples. In addition, knockdown of KMT2A and treatment with Mi-503, a small molecule inhibitor of the KMT2A-MEN1 axis, resulted in decreased H3K4me1 promoter occupancy and decreased expression of PLK1. While translocations and PTD of KMT2A have long been implicated in AML pathogenesis, <1% of CMML patients demonstrate KMT2A translocations[22] and no patients had KMT2A PTD in our study. Thus, our findings demonstrate that an unmutated KMT2A can also have an oncogenic role[24] and that KMT2A plays an unusual role in pCMML by depositing H3K4me1 at promoters of activated genes.

TET2, ASXL1, and SRSF2 are the most commonly mutated genes in CMML, with truncating ASXL1 mutations being prognostically detrimental[2,10,34]. ASXL1 mutations are thought to decrease catalytic activity of PRC2, resulting in depletion of H3K27me3, a repressive mark[38–41]. A second school of thought attributes detrimental effects of ASXL1 mutations to enhanced activity of the ASXL1-BAP1 complex, resulting in global erasure

of H2AK119Ub, and H3K27me3[39]. By using ChIP-seq in CMML samples, we demonstrate that H3K27me3 occupancy and recruitment at the PLK1 promoter is not significantly different in RAS pathway-mutated pCMML relative to RAS pathway wild-type dCMML, regardless of ASXL1 mutational status (sequence-specific analysis restricted to PLK1).

PLK1 (volasertib and onvansertib) inhibitors have completed clinical trials for patients with myeloid neoplasms (MDS and AML) with reasonable safety data, prompting us to establish the therapeutic efficacy of these agents in pCMML models[22,42–44]. While a recent randomized phase III clinical trial testing the efficacy of volasertib with low-dose cytarabine (LDAC vs LDAC and placebo- NCT01721876) in AML patients ineligible for standard induction chemotherapy, did not meet its stated endpoints, this trial was not personalized to subsets of AML patients more likely to respond to therapy. In this study, the overall response rates for volasertib with LDAC vs LDAC with placebo were 25% and 16%, respectively ($p = 0.07$). We show that targeting PLK1 with volasertib, both in vitro and in vivo, results in effective killing of neoplastic cells, restoring normal hematopoiesis, with a differential sensitivity favoring RAS mutant pCMML cells. Given these drugs have an established safety record in myeloid malignancies; we aim to translate these findings into an early-phase clinical trial for pCMML.

In summary, we demonstrate acquisition of clonal oncogenic RAS pathway mutations defines a unique subgroup of CMML with poor outcomes. Detection of these mutations offers a therapeutic window of opportunity for pCMML by identifying the significance of KMT2A expression and downstream mitotic check point kinases such as PLK1, which can be targeted efficiently using existing small molecule inhibitors. These findings provide detailed insight into the biology of CMML and lay the foundation for development of personalized clinical therapeutics for this otherwise devastating disease.

## Methods

**Authorizations, patient cohorts, cell collection, and sorting**. This study was carried out at the Mayo Clinic in Rochester, Minnesota, after approval from the Mayo Clinic Institutional Review Board (IRB- 15-003786), at Gustave Roussy Cancer Center in Villejuif, France, following the approval of the ethical committee Ile-de-France 1 (DC-2014-2091), and at Sigmund Freud University in Vienna, Austria, following the approval of the ethics committee of the City of Vienna (15-059-VK). Animal experiments were carried out at the Mayo Clinic after approval from the Mayo Clinic IACUC (protocol A00003488-18), the Moffitt Cancer Center in Florida IACUC (protocol 6041) and the University of Wisconsin at Madison IACUC (protocol M005328). Several patient cohorts were analyzed. In all cases, diagnosis was according to 2016 iteration of the WHO classification of myeloid malignancies[1]. PB and BM samples were collected in EDTA tubes after informed consent. MNCs were collected on Ficoll-Hypaque whereas CD14- and CD3-positive cells were sorted using magnetic beads and the AutoMacs system (Miltenyi Biotech, Bergish Gladbach, Germany). In five cases, buccal mucosa cells were used as germline control cells. All international cohorts on which NGS was performed have been represented in Supplementary Table 1 including the Mayo Clinic, Austrian and French (GFM) cohorts. Of the 591 patients with CMML collected at Mayo Clinic, genetic abnormalities were screened in 397 using a 36-gene panel targeted NGS assay. The 592 patients with CMML recruited in France ($n = 417$) and in Austria ($n = 175$) had molecular information obtained using a 38-gene NGS panel. One patient cohort on which WES was performed is represented in Supplementary Table 3. This included 48 patients with CMML whose paired MNC samples were collected in chronic phase and in LT to perform WES. A second cohort of 18 patients with CMML was used to perform WES on samples serially collected along disease progression. This cohort had been described previously[6].

All patient samples were obtained either at diagnosis, or at first referral, which was usually within 3–6 months of diagnosis. Patients not meeting these criteria were not included in these cohorts. All patients were treatment naïve at the time of collection of their samples for sequencing. None of them were in a clinical trial at the time of sample collection. The disease stage is mentioned in the table under the World Health Organization-2016 diagnosis section. While 250 of the Mayo Clinic patients were eventually treated with hypomethylating agents, given the retrospective nature of this study and treatment at centers outside of Mayo Clinic, accurate response data and duration of treatment information is not available. The same applies to the European cohorts. All patients in this data set that underwent

allogeneic stem cell transplant (10%) were accurately accounted for and were censored in survival data calculations, as transplant is the only disease modifying treatment modality for CMML.

The following statistical methods were used on the clinical data sets: Distribution of continuous variables was statistically compared using Mann–Whitney or Kruskal–Wallis tests, while nominal or categorical variables were compared using the Chi-Square or Fischer's exact test. Time to event analyses used the method of Kaplan–Meier for univariate comparisons using the log-rank test. Overall survival was calculated from the date of diagnosis to date of death or last follow-up, while AML-free survival was calculated from date of diagnosis to date of AML transformation or death. All statistical calculations were carried out using JMP statistical software, version 16.0, SAS Institute Inc. Cary, NC, 1989-2019.

**Whole exome sequencing**. One microgram of genomic DNA was sheared with the Covaris S2 system (LGC Genomics). Exome capture and sequencing library preparation were performed using Agilent SureSelect Human All Exon V5 (Agilent Genomics, Santa Clara, CA), following standard protocols. The final libraries were indexed, pooled and paired-ends ($2 \times 100$ bp) sequenced on Illumina HiSeq platforms (San Diego, CA). Germ-line DNA was obtained from CD3-positive cells for 5 patients of the CMML LT dataset and 17 patients with serial WES analyses[6]. The mean coverage in the targeted regions was 100-fold. Sequencing data are deposited at the European Genome–Phenome Archive (EGA), hosted by the European Bioinformatics Institute (EBI). Paired-end reads in the form of FASTQ files were aligned to the GRCh37 version of the human genome with BWA or Novalign to generate BAM files. Variant calling analysis was performed using Agilent SureCall software. Genomic variant information including evidence of and somatic status in cancer, significant protein domains, and known pathogenicity was further annotated using public databases: The Catalogue of Somatic Mutations in Cancer -COSMIC (https://cancer.sanger.ac.uk/cosmic), ClinVar (https://www.ncbi.nlm.nih.gov/clinvar) and UniProt (https://www.uniprot.org). Allele frequency in the normal population was calculated using the Exome Aggregation Consortium ExAC (http://exac.broadinstitute.org). Alternative alleles with <5 reads and/or a frequency <5% were excluded. All variants passing the sequencing quality-control filters and within a gene coding region, regardless of database population frequency, underwent characterization. Since CD3-positive cells are known to be contaminated with tumor cells we performed MuTect2 in the 'tumor-only' mode[27]. For all identified mutations and indels variant allele frequencies (VAF) in the corresponding CD3-positive DNA was retrieved from BAM files. All the variants with $p$ values from Fisher exact test below $10^{-4}$ were considered as somatic. Regions of somatic copy number alterations (SCNA) were screened manually. A panel of normal samples (PON) was constructed from 50 germline samples and was used to reduce the rate of false positives. Additionally, only stop-gain, splice-site and probably damaging nonsynonymous mutations were considered to be putative drivers in tumor suppressor genes. SCNA analysis was performed with Facets statistical and computational analysis pipeline in 'tumor-normal' mode where possible, or in 'tumor only' mode. cnLOH was called based on the germline heterozygous SNPs VAFs in the tumor[45]. Tumor evolution was visualized with ClonEvol (https://ieeexplore.ieee.org/document/6650525). WES data that support the findings in Fig. 2 have been deposited in EGA under ID EGAD00001007026.

**Targeted next-generation sequencing**. Analysis of the Mayo cohort (#3) was done using a panel in which the regions of 36 genes was selected for custom target capture using Agilent SureSelect Target Enrichment Kit. Briefly, libraries derived from each DNA sample were prepared using NEB Ultra II (New England Biolabs, Ipswitch, MA) and individually bar coded by dual indexing. Sequencing was performed on a HiSeq 4000 (San Diego, CA) with 150-bp paired-end reads and consisted of 48 pooled libraries per lane. Median sequence read depth was ~400×. The custom panel of target regions covered all coding regions and consensus splice sites from the following 36 genes: *ASXL1, CALR, CBL, CEBPA, DNMT3A, EZH2, FLT3, IDH1, IDH2, IKZF1, JAK2, KRAS, MPL, NPM1, NRAS, PHF6, PTPN11, RUNX1, SETBP1, SF3B1, SH2B3, SRSF2, TET2, TP53, U2AF1,* and *ZRSR2.* Paired end reads (FASTQ files) were processed and analyzed using same method as WES and as previously published by our group[10,34,46]. The French part of cohort (# 4) was analyzed using an Ion AmpliSeq™ Custom Panel Primer Pools (10 ng of gDNA per primer pool) to perform multiplex PCR targeting all coding regions or selected regions of the following 36 genes (92 kb): *ASXL1* (exons 11-12), *BCOR* (2-14), *BRAF* (15), *CBL* (4-5, 7-9, 11-14), *CEBPA* (1), *CSF3R* (3-18), *DNMT3A* (2-23), *ETV6* (1-8), *EZH2* (2-20), *FLT3* (15-20), *GATA2* (4-6), *IDH1* (4, 6), *IDH2* (4), *JAK2* (12, 14), *KDM6A* (full), *KIT* (2, 8-11, 13-14, 16-17), *KRAS* (2-4), *MPL* (10), *MYD88* (5), *NRAS* (2-4), *OGT* (1-22), *PHF6* (1, 4, 6-7, 9), *PTPN11* (1-15), *RAD21* (2-14), *RIT1* (4-5), *RUNX1* (2-9), *SETBP1* (4-5), *SF3B1* (full), *SRSF2* (1–2), *STAG1* (2-34), *STAG2* (3-35), *TET2* (3-11), *TP53* (2-11), *U2AF1* (2, 6), *WT1* (1, 7), and *ZRSR2* (full). PCR amplicons after Fupa digestion were end-repaired, extended with an 'A' base on the 3'end, ligated with indexed paired-end adapters (NEXTflex, Bioo Scientific) using the Bravo Platform (Agilent), amplified by PCR for 6 cycles and purified with AMPure XP beads (Beckman Coulter). Sequencing ($2\times250$ bp reads) was performed on a MiSeq flow cell (Illumina, San Diego, CA) using the onboard cluster method.

Sequencing reads were mapped to the GRCh37 version of reference genome with the Burrows-Wheeler Aligner (BWA) (PMID: 19451168) and then processed

with the Genome Analysis Toolkit (GATK; version 4.1.2.0) (PMID: 20644199) following best practices for exomes and specific indications for smaller targeted regions. Sequencing quality and target enrichment were verified with Picard tools metrics. Germinal DNA was obtained from CD3-positive cells for 5 patients from the CMML/sAML dataset and for 17 patients from CMML with serial WES analyses. Since CD3-positive cells are known to be contaminated with tumor cells (PMID: 26457647; PMID: 26908133) we performed MuTect2 in the 'tumor-only' mode. For all identified mutations and indels Variant Allele Frequencies in the corresponding CD3 + DNA was retrieved from bam files. All the variants with p values from Fisher exact test below $10^{-4}$ were considered as somatic. Regions of Somatic copy Number Alterations SCNA were screened manually. A panel of normal samples (PON) was constructed from 50 germline samples ad was used to reduce the rate of false positives. We filtered out all polymorphisms from ExAC database (http://exac.broadinstitute.org) with allelic frequency above 0.001. Additionally, variants were considered putative drivers if reported in the COSMIC v89 database. Only stop-gain, frameshift indels, splice-site and probably damaging nonsynonymous mutations were considered to be putative drivers in tumor suppressor genes.

SCNA analysis was performed with Facets statistical and computational analysis pipeline in 'tumor-normal' mode where possible, or in 'tumor only' mode. cnLOH was called based on the germline heterozygous SNPs VAFs in the tumor (PMID: 27270079).

All the variant processing was conducted in python with pandas library. Fisher exact test was used for contingency tables. Mann-Whitney test was used for two groups of continuous variables. Pearson-Spearman test was used for pairwise correlations of continuous variables. Log-rank test was used for survival analyses. Multivariate analysis was executed using logistic regression model, LASSO with selected features (Regression Shrinkage and Selection via the Lasso. Robert Tibshirani) and applied to measure enrichment / depletion between two subgroups. Tumor evolution was visualized with ClonEvol software (https://ieeexplore.ieee.org/document/6650525).

**Hematopoietic progenitor colony forming assay**. PB and/or BM samples were first treated with ammonium chloride to deplete red blood cells, washed in RPMI, then plated at a final concentration of $5 \times 10^4$–$2 \times 10^5$ cells/mL into methylcellulose formulated with recombinant cytokines to support the optimal growth of erythroid progenitor cells (BFU-E and CFU-E), granulocyte-macrophage progenitor cells (CFU-GM, CFU-G and CFU-M), and multipotent granulocyte, erythroid, macrophage, and megakaryocyte progenitor cells (CFU-GEMM). Plates were incubated at 37 °C and colonies were enumerated on day 10–14 as described[47]. Individual colonies were isolated, placed into a 40 mM NaOH buffer, and heated to 95°C for 15 min. Up to 5 μL of product was submitted for polymerase chain reaction (PCR) using standard conditions. Cycling conditions consisted of: an initial denaturation at 94 °C for 2 min; 40 cycles of denaturation at 94 °C for 30 s, annealing at 56 °C for 30 s, and extension at 72 °C for 1 min, and final extension at 72 °C for 3 min. PCR products were visualized by 1.3% agarose gel, purified, and the final product was subjected to bidirectional sequence analysis (GeneWiz, South Plainfield, NJ).

**RNA sequencing**. Library preparation for the RNA samples was done using Illumina TruSeq Stranded Total RNA and sequencing was done using Illumina HiSeq 4000, 100 cycles × 2 paired-end reads. RNA-seq sequencing reads were processed through the MAPRSeq v2.0. bioinformatics workflow as described in Kalari et al.[48]. Briefly reads were mapped using TopHat version 2.06 against the reference GRCh37 without alternative haplotypes using the transcript models from UCSD (March 2012) available from Illumina iGenomes Project. Gene counts were calculated with featureCounts[49], differential expression was performed using edgeR and clustering was performed using heatmap.2[50]. RNA samples were prepared and RNA quality was determined using an Agilent Bioanalyzer RNA Nanochip or Caliper RNA assay and arrayed into a 96-well plate. The paired-end sequencing libraries were prepared following BC Cancer Genome Sciences Centre's strand-specific, plate-based library construction protocol on a Microlab NIMBUS robot (Hamilton Robotics, USA). Libraries were sequenced on an Illumina HiSeq2500 platform to a read depth of approximately 50 million reads per sample. Reads were aligned to the human genome (GRCh38/hg38) using the HISAT2 aligner[51]. Read counts were generated using featureCounts with reference to Homo_sapiens. GRCh38.90.gtf annotation file from Ensembl[49,52]. Patient's samples were grouped based on the previously identified histological classifications and differential expression analyses were performed using the R Bioconductor package DESeq2[53]. Pathway data was derived from ConsensusPathDB[54]. Heatmaps were derived using the R package pheatmap (pheatmap: CRAN.R-project.org/package=pheatmap) and principle component analyses were derived using the R package rgl (rgl: CRAN.R-project.org/package=rgl).

**RNA PCR**. For the RNAseq validation we analyzed a subset of target genes by Quantitative Reverse Transcription Polymerase Chain Reaction (RT-qPCR). Total left-over RNA from the RNAseq was used to synthetize cDNA with the High-Capacity cDNA Reverse Transcription Kit (Applied Biosystems, Carlsbad, CA). A dilution 1/3 of the total cDNA was amplified by real-time PCR. Samples were prepared with PerfeCTa SYBR Green FastMix (Quanta BioSciences Inc., Gaithersburg, MD) and the primer sets described in Supplementary Table 4.

**Chromatin immunoprecipitation and sequencing**. Approximately 100,000 cells from each CMML patient were used for native chromatin immunoprecipitation (N-ChIP). Cells were lysed on ice for 20 min in lysis buffer containing 0.1% Triton X-100, 0.1% deoxycholate, and protease inhibitors. Extracted chromatin was digested with 90 U of MNase enzyme (New England Biolabs) for 6 min at 25 °C. The reaction was quenched with 250 µM of EDTA post-digestion. A mix of 1% Triton X-100 and 1% deoxycholate was added to digested samples and incubated on ice for 20 min. Digested chromatin was pooled and pre-cleared in IP buffer (20 mM Tris-HCl [pH7.5], 2 mM EDTA, 150 mM NaCl, 0.1% Triton X-100, 0.1% deoxycholate) plus protease inhibitors with pre-washed Protein A/G Dynabeads (Thermo Fisher Scientific; Waltham, MA, USA) at 4 °C for 1.5 h. Supernatants were removed from the beads and transferred to a 96-well plate containing the antibody-bead complex. Following an overnight 4 °C incubation, samples were washed twice with Low Salt Buffer (20 mM Tris-HCl [pH 8.0], 0.1% SDS, 1% Triton X-100, 2 mM EDTA, 150 mM NaCl) and twice with High Salt Buffer (20 mM Tris-HCl [pH 8.0], 0.1% SDS, 1% Triton X-100, 2 mM EDTA, and 500 mM NaCl). DNA-antibody complexes were eluted in Elution Buffer (100 mM NaHCO$_3$, 1% SDS), incubated at 65 °C for 90 min. Protein digestion was performed on the eluted DNA samples at 50 °C for 30 min (Qiagen Protease mix). ChIP DNA was purified using Sera-Mag beads (Fisher Scientific) with 30% PEG before library construction. Library construction using enriched DNA from histone mark ChIP and Input was made according to a modified indexed paired-end library protocol (Illumina Inc., USA). Unique indexed-libraries were prepared for each sample and epigenetic mark. Samples were pooled at 1:1:2:2 ratios for H3K4Me1:H3K4me3: H3K27me3: Input, respectively and subjected to paired-end sequencing (100 cycles) on Illumina HiSeq2500. Sequencing to a depth of ~25 million (H3K4me1 and H3K4me3) or ~50 million reads (H3K27me3 and Input) per sample was performed. Reads were de-multiplexed and aligned to the human genome (GRCh38/hg38) using bowtie2[55]. Aligned reads were sorted and indexed using samtools[56] and areas in the genome enriched with aligned reads (also called peaks) were called using MACS2[57] callpeak v2.2.6 with the following parameters: --format BAMPE --gsize hs --keep-dup all --q-value 0.05 --broad --broad-cutoff 0.1 for H3K4me1broad mark. All peaks were called relative to their respective input controls using the paired end mode at a default q-value cut-off of 0.05 as indicated. Both H3K4me1 and H3K27me3 samples were called using default options for broad peaks whereas H3K4me3 samples were called using default options for narrow peaks according to ENCODE recommendation. With peak comparisons, for each group, overlapping peaks were identified by Homer mergePeaks v4.11.1 software with the following options: -gsize hs -d given. Regions of differential enrichment were derived using the R Bioconductor package DiffBind (citation: *DiffBind: differential binding analysis of ChIP-Seq peak data.* http://bioconductor.org/packages/release/bioc/vignettes/DiffBind/inst/doc/DiffBind.pdf) and areas of interest were annotated using Homer[58–62]. To identify differences in DNA binding, the Bioconductor package Diffbind v2.16.0 was used in R v4.0.3. Reads were counted in intervals using dba.count function and mapQCth = 1, minOverlap =3 parameters. Differential binding affinity analysis was performed using dba.analyze with method = DBA_DESEQ2, bCorPlot = F parameters. Differentially bound sites identified were then used to plot MA with dba.plotMA function. Bigwig files, normalized for sequencing depths, have been generated with deeptools. Individual tracks for dysplastic and proliferative samples were visualized using UCSC repository. ChIP-seq data that support the findings in Fig. 5 have been deposited in GEO under series GSE156377.

**Chromatin immunoprecipitation-qPCR**. ChIP was performed as described previously[60]. Briefly, PB MNC from CMML patients (20 × 106) were crosslinked with 1% formaldehyde directly into the medium for 10 min at room temperature. The cells were then washed with PBS, collected by centrifugation at 800×*g* for 5 min at 4 °C, re-suspended in Cell Lysis Buffer (20 mM Tris [pH 8];15 mM PIPES; 85 mM KCl; 0.5% NP-40), and incubated on ice for 15 min. Resulting pellets were then re-suspended in Nuclear Lysis Buffer (50 mM Tris [pH 8.1]; 10 mM EDTA; 1% SDS) and chromatin was sheared to an average fragment size of 200–600 bp using a water bath sonicator (Bioruptor, Diagenode, Denville, NJ). Input chromatin was retained from each individual sample prior to addition of antibody for qPCR analysis. Chromatin was incubated overnight at 4 °C with magnetic beads (Dynabeads Protein G, Invitrogen) and the following antibodies (4 µg each): KMT2A (Abcam); H3K4me1 (Abcam); normal rabbit IgG (EMD Millipore) and normal mouse IgG (EMD Millipore). Following immunoprecipitation, crosslinks were removed, and immunoprecipitated DNA was purified using spin columns and subsequently amplified by qPCR. Quantitative PCR was performed using a primer set to amplify a region located between -588bp and -482bp relative to the first base pair of exon one of human *PLK1*. Base pairs in the table below indicate the position of the primer set relative to the first base pair of exon one of *PLK1*. The primer sequences can be found in Supplementary Table 4. The primer set was designed for optimal performance while localizing the amplicon within approximately the center of the pCMML H3K4me1 consensus peak in the ChIP-seq data. Quantitative SYBR PCR was performed in triplicate for each sample or input using the C1000 Thermal Cycler (BioRad). The relative enrichment for ChIP qPCR was calcutated by starting with the raw Ct values for each antibody used (H3K4me1, KMT2A, IgG control). The ΔΔCt was then calculated by subtracting the IgG control Ct from the immunoprecipitation Ct. Fold enrichment was then calculated as $2^{-\Delta\Delta Ct}$.

**Electroporation technique for siRNA experiments**. Patient-derived PB and/or BM MNC were first treated with ammonium chloride to deplete mature, non-nucleated red blood cells, washed in RPMI, and then plated into methylcellulose (see Hematopoietic progenitor colony forming assays for further details) to allow for progenitor expansion. Cells were liberated from methylcellulose by solubilizing them with RPMI warmed to 37 °C. Cells were washed with RPMI before being re-suspended in Lonza electroporation solution from Kit C (Lonza, Basel, Switzerland) and electroporated using program W-001 using the Amaxa nucleofector (Lonza, Basel, Switzerland). For every one million cells transfected, 2 µg of plasmid constructs or 60 pg of siRNA was used. Electroporation efficiency was monitored by transfecting an eGFP construct as a control. After electroporation, cells were maintained in RPMI supplemented with StemSpan (StemCell Technologies #02691) for 72 h before experiments were conducted.

**Recombinant adenoviral transduction procedure**. We utilized eGFP Adenovirus (#1060) and *NRAS* viral oncogene homolog adenovirus (#1565), both from Vector Biolabs (Malvern, PA) for the transduction procedure. Briefly, we aliquoted 10 million PBMC cells from CMML patients into 1 mL of RPMI and added either control (GFP) or *NRAS*-expressing adenovirus at an MOI of 100. We spun the cells for 30 min at 1200 RPM and let the cells sit an additional 30 min at room temperature. We then collected the cells and plated them at the same concentration (using RPMI) into cytokine enriched methylcellulose media for colony formation (MethoCult™GF H4434, StemCell Technologies, Vancouver, CA). After 7 days at 37 °C, we enumerated and imaged the progenitor colony growth.

**In vitro drug studies on progenitor colonies with PLK1 inhibitor**. Patient-derived PB and/or BM MNC were plated at $4 × 10^4$ cells per well in 96-well plates in 200 µL of RPMI supplemented with 10% fetal bovine serum and 10X StemSpan (StemCell Technologies #02691)[47]. Media was also supplemented with either DMSO as a vehicle control. Volasertib (PLK1 inhibitor) was tested at concentrations 10 nM, 100 nM, 250 nM, 500 nM, and 1000 nM. On day 4 after plating the cells, they were centrifuged to aspirate and replace media. Drug assays were completed on day seven. At this time, cells in each well were counted using a hemocytometer. Subsequently, an MTS assay was performed following the manufacturer's protocol.

**Western blot**. PVDF membranes were blocked in Tris-buffered saline and 0.3% Tween-20 (TBST) with 5% skim milk overnight at 4 °C. Primary antibodies used in western blotting were prepared in a solution of TBST with 2% skim milk at a working concentration of 1:1000 and included antibodies against PLK1 (4535), KMT2A (14197), phospho-ERK1/2 (Thr202/Tyr204), JAK2 (3230), and STAT3 (4904) all from Cell Signaling Technology. Antibodies were also used against vinculin (Bethyl, A302-535A), total ERK1/2 (R&D, MAB1576), NRAS (Santa Cruz, Sc-31). HRP-conjugated goat anti-rabbit IgG (Millipore, 12-348) and goat anti-mouse IgG (Millipore, 12-349) secondary antibodies were used at a concentration of 1:5000 and detected using SuperSignal West Dura Extended Duration Substrate (Thermo Scientific, 34076).

**Genetically engineered mouse models**. The *NRAS*$^{G12D+}$ genetically engineered mouse model used for this manuscript was developed in collaboration with the Zhang laboratory at the University of Wisconsin in Madison. All mouse lines were maintained in a pure C57BL/6 genetic background (>N10). Mice bearing a conditional oncogenic *Nras* allele (*Nras$^{Lox-stop-Lox (LSL) G12D/+}$*) were crossed to *Vav-Cre*[63] mice to generate the experimental mice, including *Nras $^{LSL G12D/+}$; Vav-Cre,* and *Vav-Cre* mice. These mice were genotyped as previously described[64,65]. All animal experiments were conducted in accordance with the *Guide for the Care and Use of Laboratory Animals* and approved by an Animal Care and Use Committee. For lineage analysis of PB and BM, flow cytometric analyses were performed as described[66]. Stained cells were analyzed on a LSR II flow cytometer (BD Biosciences). Antibodies specific for surface antigens Mac-1 (M1/70) and Gr-1 (RB6-8C5) were purchased from eBioscience. Surface proteins were detected with FITC-conjugated antibodies (eBioscience unless specified) against B220 (RA3-6B2), Gr-1 (RB6-8C5), CD3 (17A2, Biolegend), CD4 (GK1.5), CD8 (53-6.7), and TER119 (TER-119, 14-5921-85), and PE-conjugated anti-CD117/c-Kit (2B8) antibody. To analyze phospho-ERK1/2, total BM cells were deprived of serum and cytokines and then stimulated with cytokines as previously described[66]. Phospho-ERK1/2 was analyzed in defined Lin$^{-/low}$ c-Kit$^+$ cells as described. Phospho-ERK1/2 was detected by a primary antibody against phospho-ERK (Thr202/Tyr204; Cell signaling Technology) followed by APC-conjugated donkey anti-rabbit F(ab')2 fragment (Jackson ImmunoResearch, AB-2340601). Mouse tissues were fixed in 10% neutral buffered formalin (Sigma-Aldrich) and further processed at the Experimental Pathology Laboratory at the University of Wisconsin-Madison and at the Mayo Clinic.

**Patient-derived xenograft models**. The NOD.Cg-*Prkdc$^{scid}$ Il2rg$^{tm1Wjl}$*/SzJ-SGM3 (NSG-S) mice were used for the PDX experiments[25]. The mice were 6–8 weeks old at the start of the experiments with a weight of 20–30 g. Mice were irradiated at 2.5 Gy with a cesium source irradiator and xenografted 24 h after irradiation, with ×10 (6) BM MNCs from *NRAS* mutant CMML patients. Informed consent was

obtained from all patients whose BM MNCs were used to generate patient PDX models (Mayo Clinic IRB# 15- 003786 and Mayo Clinic IACUC protocol A00003488-18). In this manuscript, we provide experimental details on two PDX models generated in triplicate. Both patients had provided written informed consent and had pCMML. Individual patient details are as follows:

1. Patient 1 - *NRAS, ASXL1, TET2* mutated pCMML with 6% bone marrow blasts and 0% circulating blasts. Treatment naïve at time of sample collection with a normal karyotype.
2. Patient 2 - *NRAS, TET2, SETBP1*, and *SRSF2* mutated pCMML with 3% bone marrow blasts and 0% circulating blasts. Treatment naïve at the time of sample collection with a normal karyotype.

Volasertib was dosed at 20 mg/kg body weight intraperitoneally once a week for 2 weeks[16,67,68]. Mice were monitored daily for signs of disease and euthanized by carbon dioxide at moribund appearance, or 2 weeks after the last treatment. Spleen, BM, and PB cells were collected at the time of necropsy and processed for flow cytometry. Cells from blood, BM and spleen were stained with the following antibodies: murine CD45.1 BUV737 (BD Biosciences-564574), human CD45 BV605 (BD Biosciences-564048), human CD3 APC (BD Biosciences-561810), and human CD33 PE (BD Biosciences-555450). All blood, BM, and spleen samples were analyzed on a LSRII flow cytometer (BD Biosciences). Femur, spleen, lung, and liver samples were fixed overnight in 10% neutral buffered formalin, dehydrated, paraffin embedded and 5 μm sections were prepared. After hematoxylin-eosin staining was performed on all of the samples, sections were examined by a pathologist. Complete blood counts were also analyzed on all PDX mice at endpoint (IDEXX Veterinary Diagnostics). Immunohistochemistry staining was performed at the Pathology Research Core (Mayo Clinic, Rochester, MN) using the Leica Bond RX stainer (Leica). Slides for CD3 stain were retrieved for 30 min using Epitope Retrieval 2 (EDTA; Leica) and incubated in protein block (Rodent Block M, Biocare) for 30 min. The CD3 primary antibody (Clone F7.2.38; Dako) was diluted to 1:150 in Bond Antibody Diluent (Leica). Slides for CD14 and CD163 stains were retrieved for 20 min using Epitope Retrieval 2 (EDTA; Leica) incubated in protein block (Rodent Block M, Biocare) for 30 min. The CD14 primary antibody (Rabbit Polyclonal – HPA001887; Sigma) was diluted to 1:300 in Bond Antibody Diluent (Leica). The CD163 primary antibody (Clone 10D6, Leica) was diluted to 1:400 Background Reducing Diluent (Dako). All primary antibodies were incubated for 15 min. The detection system used was Polymer Refine Detection System (Leica). This system includes the hydrogen peroxidase block, post primary and polymer reagent, DAB, and Hematoxylin. Immunostaining visualization was achieved by incubating slides 10 min in DAB and DAB buffer (1:19 mixture) from the Bond Polymer Refine Detection System. To this point, slides were rinsed between steps with 1X Bond Wash Buffer (Leica). Slides were counterstained for five minutes using Schmidt hematoxylin and molecular biology grade water (1:1 mixture), followed by several rinses in 1X Bond wash buffer and distilled water, this is not the hematoxylin provided with the Refine kit. Once the immunochemistry process was completed, slides were removed from the stainer and rinsed in tap water for five minutes. Slides were dehydrated in increasing concentrations of ethyl alcohol and cleared in three changes of xylene prior to permanent cover slipping in xylene-based medium. Mice assessments included impact of therapy on the neoplastic clone (assessed by flow cytometry), impact on the percentage of circulating blasts and monocytes, impact on liver and spleen sizes and weights, impact on BM changes such as hypercellularity and dysplasia. We did also assess potential adverse events, including additional cytopenias, cutaneous and gastrointestinal complications. Volasertib dosing at 40 mg/kg body weight intraperitoneally did result in significant gastrointestinal toxicity (necrotic bowel loops) along with death of the treated mice.

**Statistical analysis**. Distribution of continuous variables was statistically compared using Mann–Whitney or Kruskal–Wallis tests, while nominal or categorical variables were compared using the $\chi^2$ or Fischer's exact test. Time to event analyses used the method of Kaplan–Meier for univariate comparisons using the log-rank test. OS was calculated from the date of diagnosis to date of death or last follow-up, while AML-free survival (LFS) was calculated from date of diagnosis to date of AML transformation or death. All statistical calculations were carried out using JMP statistical software, version 16.0, SAS Institute Inc. Cary, NC, 1989-2019.

**Reporting summary**. Further information on research design is available in the Nature Research Reporting Summary linked to this article.

## Data availability

We do not have patient consent to release raw NGS data. Any inquiries for accessing these data should be directed to Mrinal Patnaik (patnaik.mrinal@mayo.edu) and we will grant access to the de-identified datasets for research purposes. WES data that support the findings in Fig. 2 have been deposited in EGA under ID EGAD00001007026. RNA-seq data that support the findings in Figs. 4 and 5 have been deposited in GEO under series GSE156209. The ChIP-seq data that support the findings of this study have been deposited in GEO under series GSE156377. The UCSC session can be accessed through the genome browser [https://genome.ucsc.edu/s/mbinder/CMML_RAS_H3K4me1]. Source data are provided with this paper.

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

## Acknowledgements

Current publication is supported in part by grants from the "Gerstner Family Career Development Award", The "Center for Individualized Medicine- Mayo Clinic", and "The Henry J. Predolin Foundation for Research in Leukemia, Mayo Clinic, Rochester, MN, USA". This publication was also supported by CTSA Grant Number KL2 TR000136 from the National Center for Advancing Translational Science (NCATS). Its contents are solely the responsibility of the authors and do not necessarily represent the official views of the NIH. R.M.C. was supported by the 2020 Eagles 5th District Cancer Telethon Cancer Research Fund Fellowship Award. M.E.F.Z. was supported by CA136526. S.I.N. was supported by Foundation ARC 2017, Foundation Gustave Roussy and Swiss Cancer League (grant KFC-3985-08-2016) grants. ES group is labeled "Equipe labellisée de la Ligue Nationale Contre le Cancer and supported by the French National Cancer Institute. P.V. was supported by the Austrian National Science Fund (FWF) grants F4704-B20 and P30625-B28. A.A.M was supported by the Conquer Cancer Foundation of American Society of Clinical Oncology (ASCO) Young Investigator Award (YIA) and Grant. We would like to thank the University of Wisconsin Carbone Comprehensive Cancer Center (UWCCC) for use of its Shared Services (Flow Cytometry Laboratory, Genome Editing and Animal Models Shared Resource, and Experimental Pathology Laboratory) to complete this research. This work was supported by R01CA152108 and a Scholar Award from the Leukemia & Lymphoma Society to J.Z. This work was also supported in part by NIH/NCI P30 CA014520-UW-Comprehensive Cancer Center Support. The diagram in Supplementary Fig. 7A was created using BioRender.com.

## Author contributions

R.M.C., D.V., T.L., D.L.M., E.J.T., L.L.A., I.H., A.B., A.A.M., S.N., A.Y., I.P., G.C., S.L.S., J.H., and M.M.P. performed experiments, analyzed sequencing data, and helped write the manuscript. A.V., X.Y., M.B., E.P., and J.Z. helped perform mouse experiments with A.V., M.E.B., and E.P. conducting PDX experiments and X.Y. and J.Z. developing the GEMM. B.A., K.D.R., M.B., N.D., C.P., K.B., and M.E.F. helped with ChIP-seq. M.T.H. helped with histopathology and flow cytometry. T.W., S.H.K., G.S., P.V., E.S., T.G., K.G., and M.M.P. helped write and edit the manuscript.

## Competing interests

The authors declare no competing interests.
