## [Peer Review File · Nature Communications]

Reviewers' comments:

Reviewer #1 (Remarks to the Author); expert on CMML patient stratification and genomics/RNAseq:

In this manuscript, Carr and colleagues reported the results of genetic, functional and mouse models studies suggesting that RAS pathway mutations are associated with a unique gene expression profile enriched in mitotic kinases, and that this process is regulated by overexpressed KMT2A resulting in promoter H3K4 monomethylation. In addition, pharmacologic inhibition of PLK1 in vitro and in RAS mutant patient-derived xenografts suggested potential efficacy of such therapeutic strategy in proliferative CMML.

Major points

The Author summarize in the manuscript title that - RAS mutations drive proliferative chronic myelomonocytic leukemia via a novel KMT2A-PLK1 axis. However, the experimental plan is entirely based on NRAS mutation, and no evidence that the findings can be extended to other RAS pathway mutations is provided. As a general comment, extending the experimental plan to other RAS pathway mutations would increase the value of the findings here reported. Alternatively, the title should be changed and focused on NRAS.

The Authors provide evidence of a significant association of NRAS mutations with proliferative phenotype. However, this association does not seem exclusive, as NRAS mutations were detected in a fraction of CMML with apparent dysplastic phenotype (actually NRAS mutations are observed also in classical MDS). The Authors should report in detail clinical features in dysplastic CMML harbouring NRAS mutations, and extend the experiments to NRAS-mutated dCMML (see next point).

The Authors performed all the experiments of DNA immunoprecipitation and NGS, PLK1 and WEE1 expression, and of KMT2A expression in patients with NRAS-mutant pCMML and in patients with RAS pathway wildtype dCMML. However, in order to properly test the association of reported changes with proliferative phenotype and RAS mutation, they should include additional cohorts of RAS-wild type pCMML and RAS-mutated dCMML.

While RAS pathway mutations are significantly enriched in CMML with proliferative phenotype, a not negligible fraction of pCMML harbour different drivers. It would be of interest extending the experimental plan to other pCMML, in order to assess whether the mechanisms here reported are specific of NRAS mutations or not. They partly did this by analyzing a fraction of JAK2-mut, but these observations should be expanded (see also the point above).

The Authors should analyze in more detail differences in disease phenotype and outcome between CMML with clonal or subclonal RAS mutations.

Overexpression of PLK1 and WEE1 was analyzed based on mRNA. Expression levels should be also confirmed at the protein level.

In volasertib-treated mice, the Authors reported reduction in BM and splenic malignant monocytic/histiocytic tumor infiltrates, human leukocytes and spleen size/mass in volasertib treated mice compared to vehicle-treated controls. Were these changes significant?

The effect of NRAS and RAS pathway on survival and AML evolution should be tested in

multivariable models.

Minor points

Survival difference between proliferative and dysplastic subtypes is well known. The validation of this observation in the study cohort can be moved to supplemental material.

In the WES cohort, CMML patients without RAS pathway mutations had a better OS, in comparison to those with RAS pathway mutations ($p < 0.05$). Please include correct P value.

Discussion can be shortened and should be more focused on the study results.

Reviewer #2 (Remarks to the Author); expert on epigenetics, leukaemia and RAS:

In this article by Carr et al. the authors show that proliferative CMML is characterized by RAS pathway mutations. RAS pathway mutation is also associated with transformation to AML. These mutations are associated with increased expression of PLK1 and WEE1 kinases, kinases that are associated with cell proliferation. This over expression is secondary to overexpression of KMT2A, a H3K4 lysine methylase, secondary to Ras pathway activation. Through CHIP-seq experiments, the authors show KMT2A enriches H3K4 monomethylation at the promoters of PLK1 and WEE1. Inhibition of these kinases is shown to be a potential targeted therapy in RAS pathway mutated CMMLs using mouse models and xenografts.

The authors are commended for compiling a large data set of 1183 CMML patients to analyze mutational data and outcomes. They perform mutational analysis across this large sample set and can correlate with clinical outcomes. The authors then use primary samples to perform expression and functional experiments. These are validated with a mouse model. Overall, the data is clear as the authors describe. The work is important in providing support for clinical trials using these inhibitors.

My main concerns are regarding:

- 1) the novelty of the RAS clinical findings- which has been previously described already in CMML
- 2) the use of bulk MNCs from PB and BM for these studies
- 3) the distinction between whether the molecular pathway and drug sensitivity described is truly RAS specific vs a property of proliferative CMMLs.

In more specifics:

I have questions about the method of sequencing the separate cohorts, since they are analyzed with similar but not exactly the same genes and exons. Of the French cohort, what was the rationale for the exon selection? Would any loss of function mutations be missed since the exon selection did not cover the entire gene? How were the Austrian samples sequenced? It's unclear if the mutation profiles are complete enough for the conclusions or whether the 2 data sets are comparable.

The finding that RAS mutations are more common in proliferative CMML has been shown before. (see Ricci C, et al. Clin Cancer Res. 2010 Apr 15;16(8):2246-56. Patel BJ, et al.

Leukemia. 2017 Dec;31(12):2815-2823. Onida F. and Beran M. Curr Hematol Rep. 2004 May;3(3):218-26. Nösslinger T, et al. Leuk Res. 2001 Sep;25(9):741-7. Elena C, et al. Blood. 2016 Sep 8; 128(10): 1408–1417.) These studies show that NRAS mutations are also associated with disease progression and proliferative disease is associated with worse prognosis. This takes away from the novelty of the work

Is the effect on NRAS on survival stand up in multivariate mutation analysis using genes previously associated with prognosis (NRAS, ASXL1, RUNX1, SETBP1)?

Should NF1 be included in the Ras pathway mutations?

What do the authors mean that the “NRAS mutations were most significantly associated with aggressive disease features”? Is it more associated than other genes?

In Figure 3, I would like to see direct controls for effective NRAS transduction or transfection (also in Fig 4d, 5c for example - what is the NRAS level). How much is NRAS going up or down? The Megakaryocyte phenotype is unclear in 3e. The CST antibody for pErk is for p42,44 - there are usually 2 bands. If one looks up images of pErk Westerns the 2 bands is the convention on how pErk is shown. Can the authors show a larger portion of the blot? Are the right bands being selected? Is the tErk and pErk from the same membrane blot throughout the figures 3,4,5?

In Figure 4 and 5, it's not ideal to use total MNCs for RNA-seq and ChIP-seq. Although the authors present data in Supplementary Fig 4 to justify use of MNCs, I don't understand the argument with just the graph - were there no significantly variable genes between bulk and sorted? In the actual experiments - what were the % composition of the cell types in the analyzed samples - were the % Monocytes similar? Are the differences a result of differences in cell types? The authors should at least validate the RNA-seq results and targeted Chip-seq results in more purified populations.

In Figure 4 are the ingenuity results selective? What is the list of all pathways, how were these 4 pathways selected - were they the only ones?

The elevated PLK1 and WEE1 are derived from proliferative vs dysplastic CMML signatures. The authors then try to specify this is through NRAS. Is the data in 4e from NRAS-G12D into dCMML? They only specify CMML in the experiment design. What is the expression level of WEE1 and PLK1 in NRAS mutant dCMML? That WEE1 and PLK1 would be upregulated in proliferative cells is not unexpected. These are cell cycle related genes and would typically be increased in rapidly cycling cells. What is the expression level of WEE1 and PLK1 in pCMML that is JAK2VF mutant or pCMML without NRAS mutations compared to dCMML? Would the inhibitors volasertib and be active in pCMML without NRAS mutation or in NRAS mutant dCMML?

In Figure 5, the authors show that H3K4me1 is upregulated in pCMML with NRAS. Were input sequences used to call the PEAKS with MACS2? It's not clear from the methods. What is the consistency between samples- 18 Ras pathway and 12 dCMML? As these are MNC PB and BM samples, I would suspect there would be heterogeneity. How is this taken into account or dealt with? The authors are showing us only 1 track for each CHIP experiment and this may be quite selective. The track for Wee1 in 5B does not show the promoter only the gene body. There should be more summary data of which peaks were called and what was common between samples. Perhaps a heatmap could be included of the peaks or Venn

diagram. Can the authors add a track into their ChIP seq plots showing where peaks were called and where the differential peaks were called? Can the authors show in a correlation between the RNA-seq and the H3K4me1 promoter peaks for other genes? Is this a general trend in which H3K4me1 peaks are directing upregulation of multiple target genes?

Again, is the KMT2A over-expression just in NRAS mutant pCMML. What is the expression levels in other mutants or pCMML with NRAS pathway mutations. What is the expression in dCMML that has a RAS pathway mutation?

Reviewer #3 (Remarks to the Author); expert on PDX models, therapy and leukaemia:

The manuscript highlights the pathogenic role of RAS pathway mutations and their prognostic link with pCMML. The frequency, VAF of RAS pathway mutations are much higher in pCMML as compared to dCMML. pCMML remains an entity in need for new treatment options as existing options for treatment of MDS are not most effective in this disease, though these are used in clinical practice.

Authors also have developed a RAS mutant mouse model that mimicks human CMML. Their model differs from the MX1 CRe mRAS model and they instead use VAV Cre mRAS that more faithfully resembles CMML phenotype.

In search of therapeutic vulnerabilities they have identified PLK1 and WEE1 as targets that are over expressed in RAS mutant CMML due to upregulation of KMT2A and consequent increase in H3K4 Mono methylation at PLK1, WEE1 promoter sites.

The strength of the manuscript is importantly in that much of the work has been done using patient derived samples which makes it authentic.

There are few questions that need clarifications as well as few suggested experiments

1. RAS pathway mutations are prognostic in an univariate model. What factors are most important in a multivariate model of survival and leukemic transformation (LT) in CMML?
2. Does VAF of RAS pathway mutation as a continuous variable trump the prognostic impact of arbitrary difference between dCMML and pCMML based on WBC count
3. In the mouse model of CMML, did the mice acquire non RAS pathway mutations as disease develops?
4. In Fig 4 it may be useful to show the hierarchy of pathways that are upregulated. That will convince the reader about authors selecting to target PLK1/WEE1.
5. Did authors establish mechanistic link between mutant RAS and transcriptional upregulation of KMT2A
6. Does KMT2A knockdown phenocopy RAS knockdown in patient samples as regards colony formation etc?
7. While PLK1 inhibition reduces disease burden in the mouse model of RAS mutant CMML, authors need to show if that impacted survival in mice.

Reviewer #4 (Remarks to the Author); expert on ChIP-seq:

In this manuscript, Carr et al. elucidated the role of RAS mutations in the pathogenesis of proliferative chronic myelomonocytic leukemia (pCMML). This is a remarkable amount of work, including leukemia phenotype with the generation of a Vav-Cre-NrasG12D mice, clinical and genomic data with prognostic implications, and experimental data using primary samples with gene overexpression/knockdown. This nicely demonstrated the involvement of

NRAS pathway in the upregulation of several cell cycle regulators. I think most of the data shown in this manuscript (Figures 1-4B and 7) are very convincing.

However, given these excellent data, I have a concern about the selection of PLK1 and WEE1 genes (Figures 4C-E). I also have some concerns about their interpretation on the ChIP-seq data (Figure 5). Valid rationale and supporting evidences are needed.

Specific comments

1. Figure 4: selection of PLK1 and WEE1

It is not clear to me why the authors selected these 2 genes in Figure 4. I guess that fold-change of mRNA expression in pCMML as compared to dCMML were about x2.5-3.5 in Figure 4C. However, there are many other genes which showed higher fold-changes in Figure 4B. Also, although they mentioned several other genes (e.g., AURKA, CHEK1, CDC45, etc) on page 8, it is not clear what is the rationale focusing on PLK1 and WEE1 among these genes. I understand that PLK1 may be druggable as they showed in Figure 7. But, I strongly worry if the authors might be missing many other important genes regulated by RAS. Hence, current approach is somewhat biased. Please also see my comments 2.4 and 3

2. Figure 5: Interpretation of ChIP-seq results

The authors emphasized a global enrichment of H3K4me1 in pCMML samples as compared to dCMML cases. As examples, they showed baseline H3K4me1 signals at PLK1 and WEE1 gene loci. They also performed NRAS knockdown, which showed a significant downregulation of H3K4me1 enrichment by ChIP-PCR. I think knockdown data is convincing. However, I have some concerns about their interpretation of ChIP-seq.

1) First, they used MACS2 for the peak calling. In my understanding, peak is shown either "called" or "uncalled" from this analysis. Although the authors did not describe full details of this analysis, my guess is that they performed peak calling for each of CMML samples independently, evaluated the total number of called peaks, and they showed the average number in Figure 5A. Can they clarify this point? But, this approach is still slightly tricky. One potential pitfall is that peak calling is often affected by the level of baseline signal, which could be affected by experimental and sample conditions, especially when handling primary samples. Indeed, I see very broad peaks with high baseline signals at PLK1 and WEE1 gene loci for both pCMML and dCMML samples. I wonder which regions were actually detected by peak calling? Also, how many of dCMML and pCMML samples were called at these elements?

2) Alternatively, the authors may show metagene plots to evaluate the difference in average signal level between pCMML and dCMML samples, as they actually did for DNA methylation analysis. However, this data is currently not included. Can they perform metagene analysis for H3K4me1, H3K4me3 or H3K27me3 ChIP-seq data? Do they actually observe significant differences in H3K4me1?

3) Third, what is the definition of "promoter"? In the first paragraph on page 10, they described "enrichment of H3K4me1 at promoters of PLK1 and WEE1". Which region is the promoter they defined? A confusing is that H3K4me1 signal level at the transcriptional start site and its immediate upstream of the WEE1 gene (which I think the promoter) is even low,

while higher signals is observed above exons 2-4 of this gene in pCMML samples. I do not think this intragenic element is usually called promoter.

4) Forth, are PLK1 and WEE1 one of most significant examples that show difference in H3K4me1 by ChIP-seq? Please also see my comment 3 below.

3. Integration of RNA-seq and ChIP-seq

Although the authors have excellent RNA-seq and ChIP-seq datasets, ChIP-seq data was currently used only to highlight the WEE1 and PLK1 gene loci. These two datasets were not integrated in an unbiased way. For example, can they first select genomic loci which show significant difference in H3K4me1 signal based on ChIP-seq, and then filter the genes which are significantly upregulated in pCMML samples based on RNA-seq? If they rank the selected genes by fold-change values for RNA expression, can PLK1 and WEE1 still be selected as one of top hits?

Also, can they include supplemental tables showing fold-change and p-values for all genes selected by RNA-seq and the list of genes called by ChIP-seq? I think these data are needed as supporting evidences.

4. Figure 6: Selection of KMT2A

1) Similarly, was KMT2A one of top hits selected by RNA-seq analysis? Is this gene element also associated with higher level of H3K3me1 signal in pCMML samples?

2) The difference between KMT2A ChIP and IgG control is only about 2-fold enrichment. Can they perform ChIP for other KMT proteins to show the specificity?

POINT-BY-POINT RESPONSE

Reviewer #1

1. The Author summarize in the manuscript title that - RAS mutations drive proliferative chronic myelomonocytic leukemia via a novel KMT2A-PLK1 axis. However, the experimental plan is entirely based on NRAS mutation, and no evidence that the findings can be extended to other RAS pathway mutations is provided. As a general comment, extending the experimental plan to other RAS pathway mutations would increase the value of the findings here reported. Alternatively, the title should be changed and focused on NRAS.

Reply: We thank the reviewer for this comment. We have extended our findings to *KRAS* mutant proliferative CMML (pCMML). We demonstrate increased cell counts after transfecting *RAS* wild-type dysplastic CMML (dCMML) samples with mutant *KRAS*^{G12D} (**Supplementary Figure 3D**) and also demonstrate decreased cell count after knocking down mutant *KRAS* in pCMML samples (**Supplementary Figure 3C**). We also show decreased expression in mitotic checkpoint kinases *PLK1* and *WEE1* with *KRAS* knockdown in *KRAS* mutant pCMML samples, followed by increased expression with transfection of mutant *KRAS* in *RAS* wildtype dCMML samples (**Figure 4F and G**). Furthermore, knockdown of mutant *KRAS* in pCMML samples decreased enrichment of H3K4me1 and occupancy of KMT2A at the *PLK1* and *WEE1* gene promoters, whereas transfecting *RAS* wild-type dCMML samples with mutant *KRAS*, increased the levels of H3K4me1 and occupancy KMT2A at the gene promoters of *PLK1* and *WEE1* (**Figures 5G-I and 6E-G**). These experiments support the fact that RAS mutations drive the pCMML phenotype via a novel KMT2A-PLK1 axis and that this axis is not specific to NRAS mutations alone. *RAS* pathway mutations also include negative regulators of the *RAS* pathway, such as *CBL*, *PTPN11* and *NF1*. These genes have complex regulatory mechanisms, often involving more than one signaling pathway (*CBL* is also a negative regulator of *JAK2*), making interpretation of experiments manipulating these gene very challenging. They, however, are considered critical components of *RAS* pathway signaling and confer a similar phenotype in CMML as do direct mutations involving the *RAS* GTPases¹⁻⁶.

2. The Authors provide evidence of a significant association of NRAS mutations with proliferative phenotype. However, this association does not seem exclusive; as NRAS mutations were detected in a fraction of CMML with apparent dysplastic phenotype (actually NRAS mutations are observed also in classical MDS). The Authors should report in detail clinical features in dysplastic CMML harboring NRAS mutations, and extend the experiments to NRAS-mutated dCMML (see next point).

Reply: As suggested, we have carried out a detailed phenotypic correlation between *NRAS* mutant pCMML and *NRAS* mutant dCMML, detailed in our response to point # 5. With regards to experimental work, we first demonstrate a statistically significant lower level of expression of the mitotic checkpoint kinases, *PLK1* and *WEE1*, in *NRAS* mutant dCMML in comparison to *NRAS* mutant pCMML (**Figure 4C**). We then show that there is no impact on cell count numbers after knockdown of *NRAS* in *NRAS* mutant dCMML (**Supplementary Figure 3F**). Similarly, knockdown of *NRAS* in *NRAS* mutant dCMML had no impact on the expression of the mitotic checkpoint kinases *PLK1* and *WEE1* (**Supplementary Figure 4G**) nor did it significantly change enrichment and occupancy of H3K4Me1 or KMT2A at the gene promoters of *PLK1* and *WEE1* (**Supplementary Figures 5I and 6C**). These experiments demonstrate that oncogenic RAS pathway

mutations do not have the same phenotypic impact in dCMML as they do in pCMML. This may in part be due to the consequence of relatively lower variant allele frequency burdens of RAS pathway mutations in dCMML cases relative to pCMML⁶. In the samples used to carry out progenitor colony forming assays the mean VAF for *NRAS* mutant dCMML was 15%, while that of *NRAS* mutant pCMML was 38%.

3. The Authors performed all the experiments of DNA immunoprecipitation and NGS, *PLK1* and *WEE1* expression, and of *KMT2A* expression in patients with *NRAS*-mutant pCMML and in patients with RAS pathway wildtype dCMML. However, in order to properly test the association of reported changes with proliferative phenotype and RAS mutation, they should include additional cohorts of RAS-wild type pCMML and RAS-mutated dCMML.

Reply: We thank the reviewer for this comment. In order to prove the strict association between RAS mutant pCMML, *KMT2A* and the mitotic checkpoint kinases, we first demonstrate by qPCR that, in several patients, there is a statistically significant increase in expression of the mitotic kinases (*PLK1* and *WEE1*) in *NRAS* mutant pCMML relative to *NRAS* mutant dCMML, *JAK2* mutant pCMML and *NRAS* wild type dCMML (**Figure 4C**). We also show that knocking down *JAK2* in *JAK2* mutant (*NRAS* wild-type) pCMML has no impact on the expression of *PLK1* and *WEE1* (**Supplementary Figure 4F**), nor does it impact the occupancy and enrichment of H3K4me1 and *KMT2A* at the gene promoters of *PLK1* and *WEE1* (**Supplementary Figures 5H and 6D**). These findings are similar to experiments carried out on *NRAS* mutant dCMML samples in response to point # 2, where knockdown of *NRAS* in *NRAS* mutant dCMML had no impact on the expression of *PLK1* and *WEE1* (**Supplementary Figure 4G**), nor did it significantly change enrichment and occupancy of H3K4Me1 or *KMT2A* at the gene promoters of *PLK1* and *WEE1* (**Supplementary Figures 5I and 6C**). In summary, with the addition of these experiments we demonstrate that regulation of *PLK1* and *WEE1* expression is largely driven by clonal RAS pathway activity, as opposed to non-specific proliferative signaling from alternative drivers such as *JAK2*.

4. While RAS pathway mutations are significantly enriched in CMML with proliferative phenotype, a not negligible fraction of pCMML harbours different drivers. It would be of interest extending the experimental plan to other pCMML, in order to assess whether the mechanisms here reported are specific of *NRAS* mutations or not. They partly did this by analyzing a fraction of *JAK2*-mut, but these observations should be expanded (see also the point above).

Reply: We thank the reviewer for this comment. Approximately 10% of CMML cases have *JAK2* mutations⁷. In RAS pathway wild type pCMML, *JAK2* is the most common driver seen. We first demonstrate that expression of the mitotic checkpoint kinases, *PLK1* and *WEE1*, are lower in *JAK2* mutant pCMML in comparison to *NRAS* mutant pCMML (**Figure 4C**). We then knocked down *JAK2* in *JAK2* mutant pCMML and demonstrate no impact on the expression of the mitotic checkpoint kinases (**Supplementary Figure 4F**). Similarly, knocking down *JAK2* had no impact on the enrichment of H3K4me1 and occupancy *KMT2A* in *JAK2* mutant pCMML samples (**Supplementary Figures 5H & 6D**). Of note, we carefully selected the *JAK2* mutant pCMML samples, ensuring that they did not have either clonal or sub clonal RAS pathway mutations. In addition, as noted above, we have increased our studies of *KRAS* mutant pCMML patient samples, showing similar results as those seen with *NRAS* mutant pCMML cases. Together, these experiments conclusively demonstrate that the RAS-*KMT2A*-*PLK1* axis is specific to RAS pathway mutant pCMML and is not

seen in *JAK2* mutant pCMML.

5. The Authors should analyze in more detail differences in disease phenotype and outcome between CMML with clonal or subclonal RAS mutations.

Reply: Following the reviewer's suggestion, we have carried out a detailed analysis of disease phenotype and outcomes between dCMML with RAS pathway mutations and pCMML with RAS pathway mutations. Salient features are highlighted below.

In **Supplementary Figure 1E**, we first demonstrate that the Acute Myeloid Leukemia free survival of *NRAS* mutant pCMML is significantly worse in comparison to *NRAS* mutant dCMML ($p=0.01$). We have also performed a detailed assessment of four key clinical parameters including white blood cell count (WBC), absolute monocyte count (AMC), circulating immature myeloid cells (IMC), and peripheral blood (PB) blasts in *NRAS* mutant dCMML and compared it with *NRAS* mutant pCMML (**Supplementary Figure 1B**). With regards to all four parameters, *NRAS* mutant dCMML exhibited significantly less aggressive features compared to *NRAS* mutant pCMML. At the same time, when *NRAS* mutant dCMML was compared with *NRAS* wild-type dCMML, these findings did not reach statistical significance.

6. Overexpression of PLK1 and WEE1 was analyzed based on mRNA. Expression levels should be also confirmed at the protein level.

Reply: Protein lysates from patient-derived samples have now been analyzed by Western blotting for PLK1 and WEE1 in knockdown experiments with siKRAS (**Figure 4F**), siJAK2 and siNRAS (**Supplementary Figures 4F-G**).

7. In volasertib-treated mice, the Authors reported reduction in BM and splenic malignant monocytic/histiocytic tumor infiltrates, human leukocytes and spleen size/mass in volasertib treated mice compared to vehicle-treated controls. Were these changes significant?

Reply: Thank you for this question. The reduction in human leukocytes in the peripheral blood, spleens and bone marrows of the Volasertib-treated NSGS mice were statistically significant compared to the vehicle treated controls (**Figure 7F**). Also, in **Figure 7H**, we demonstrate a statistically significant reduction in spleen sizes in Volasertib treated NSG-S mice, in comparison to vehicle treated controls. The reductions in bone marrow and splenic malignant monocytes/histiocytes were more subjective and based on expert pathological interpretation. We were not able to apply statistical rules to this. However, there were clear differences based on morphology.

8. The effect of *NRAS* and RAS pathway on survival and AML evolution should be tested in multivariable models.

Reply: We have performed an extensive multivariate analysis for all the driver genes and clinical parameters seen in CMML patients, so as to assess for their impact on the risk for leukemic transformation and overall survival in the combined data set. This is included in **Figure 1 H-I** and **Supplementary figures 1G-J**. We have added the statistical methods in detail to the method section. This includes the following statement; Multivariate analysis was executed using logistic regression model, LASSO-selected features (Regression Shrinkage and Selection via the Lasso. Robert Tibshirani) and applied to measure enrichment / depletion between two subgroups. We then applied a multi variate cox model for prediction of overall survival based on identified leukemic

driver mutations and then combined mutational data with clinical parameters. Analysis was first carried out for individual RAS pathway genes and then was also assessed for combined RAS pathway genes delegated as the oncogenic RAS pathway category. Importantly, mutant *NRAS* alone did not reach statistical significance as an individual predictive marker impacting overall survival and AML free survival. While the combined oncogenic RAS pathway category was statistically significant in a model that only included genetic factors (HR=1.2, CI: 1.08-1.33), this association was reduced to a trend when assessed in a model that included genetic and clinical parameters (HR=1.11, CI:0.99-1.24). We have included all of these findings in **Supplementary Figure 1**.

9. Survival difference between proliferative and dysplastic subtypes is well known. The validation of this observation in the study cohort can be moved to supplemental material.

Reply: Following the reviewer's suggestion, we have moved part of these results to **Figures 1 and Supplementary Figure 1**.

11. In the WES cohort, CMML patients without RAS pathway mutations had a better OS, in comparison to those with RAS pathway mutations ($p < 0.05$). Please include correct P value.

Reply: The exact P value is 0.045. This has been added to the manuscript.

12. Discussion can be shortened and should be more focused on the study results.

Reply: As suggested, the discussion has been shortened and the results section expanded to better describe the study's findings.

Reviewer #2

1. I have questions about the method of sequencing the separate cohorts, since they are analyzed with similar but not exactly the same genes and exons. Of the French cohort, what was the rationale for the exon selection? Would any loss of function mutations be missed since the exon selection did not cover the entire gene? How were the Austrian samples sequenced? It's unclear if the mutation profiles are complete enough for the conclusions or whether the 2 data sets are comparable.

Reply: We have now provided details on the sequencing data of all 3 cohorts. Of note, all 3 centers are large clinical centers of repute and have designed their NGS probes for clinical practice. We now provide exon coverage for all three NGS panels used in the study in **Supplementary Figure 1A**. The exon coordinates (in hg38) of the driver genes were taken from:

https://ftp.ncbi.nlm.nih.gov/pub/CCDS/current_human/CCDS.current.txt.

The target coordinates for all three panels were converted from hg19 to hg38 using an online Liftover. The coordinates of the COSMIC mutations were taken from the COSMIC website in the form of the CosmicMutantExportCensus.tsv file.

Further, mutations were selected based on the following criteria:

- 1) Whether the origin of the cancer is from hematopoietic or lymphoid tissue
- 2) Mutations in coding DNA sequences (CDS)
- 2) Recurrent (2 or more times in blood cancers) missense mutations
- 3) Plus, any nonsense and frameshift mutations

We then calculated what percentages of mutations (as defined by the aforementioned criteria) were covered by targets in all 3 panels. This has been demonstrated in **Supplementary Figure 1A**. This figure also represents the percentage of length of coding DNA sequences covered by the targets. As evident from the figure, the critical recurrently mutated genes seen in CMML, including *ASXL1*, *SRSF2*, *TET2*, and RAS pathway genes had comprehensive coverage on all 3 panels. While it is certainly possible that a loss of function mutation might have been missed since entire genes have not been covered, the current exons selected for coverage have been based on prior extensive data from WGS, WES and NGS studies carried out in CMML, giving us confidence that most relevant mutations have been identified by these sequencing assays⁸⁻¹¹.

2. The finding that RAS mutations are more common in proliferative CMML has been shown before. (see Ricci C, et al. Clin Cancer Res. 2010 Apr 15;16(8):2246-56. Patel BJ, et al. Leukemia. 2017 Dec;31(12):2815-2823. Onida F. and Beran M. Curr Hematol Rep. 2004 May;3(3):218-26. Nösslinger T, et al. Leuk Res. 2001 Sep;25(9):741-7. Elena C, et al. Blood. 2016 Sep 8; 128(10): 1408–1417.) These studies show that NRAS mutations are also associated with disease progression and proliferative disease is associated with worse prognosis. This takes away from the novelty of the work.

Reply: While we agree that prior studies have demonstrated that RAS mutations are more common in pCMML and that their presence is associated with disease progression and poor outcomes, we have taken this finding to the next level by demonstrating the following:

1. Using our large data set (>1000 patients), we have defined variant allele frequency cut offs associated with clonal and subclonal RAS pathway mutations and correlated this to disease phenotype and outcomes.
2. Clonal RAS mutations are more likely to give rise to a proliferative phenotype and be associated with adverse outcomes in comparison to subclonal mutations.
3. Clonal RAS pathway mutations have a unique gene expression profile with upregulation of mitotic checkpoint kinases, such as *PLK1* and *WEE1*.
4. Mechanistically this regulation is uniquely mediated by the lysine methyltransferase KMT2A.
5. We present a unique *NRAS*^{G12D} driven genetically engineered mouse model that replicates the pCMML phenotype.
6. We present one of the largest epigenetic assessments in CMML patient samples, including techniques such as RNAseq, ChIP-seq and DIP-seq.
7. Finally and most importantly, we demonstrate therapeutic vulnerabilities by defining the RAS-KMT2A-PLK1 pathway and provide exciting preliminary data for a phase 1/2 study using PLK1 inhibition in RAS mutant pCMML.

2. Is the effect on NRAS on survival stands up in multivariate mutation analysis using genes previously associated with prognosis (NRAS, ASXL1, RUNX1, SETBP1)?

Reply: In the revised version, we have provided a very detailed multivariable analysis including genetic and clinical variables, estimating their impact on OS and AML-FS. While *NRAS* mutations did not have an independent prognostic impact in comparison to the aforementioned mutations, when it was analyzed in the context of cumulative RAS pathway mutations, statistical significance was achieved. This however was lost once clinical variables were included in the model. This is reflected in **Figures 1H-I**,

Supplementary Figures 1G-J and in the main manuscript.

3. Should NF1 be included in the Ras pathway mutations?

Reply: We completely agree that *NF1* should be included as a RAS pathway mutation. We have included it in our analysis using whole exome sequencing (**Figures 2A-D**). While it was present in the Austrian NGS panels, it was not included in the French or the Mayo clinic panels. This was largely due to the fact that it has the lowest frequency amongst all the oncogenic RAS pathway mutations⁹. We acknowledge this limitation in the method section.

4. What do the authors mean that the “NRAS mutations were most significantly associated with aggressive disease features”? Is it more associated than other genes?

Reply: We thank the reviewer for this comment. As there were several genes associated with aggressive features we modified the statement as follows: *NRAS* mutations were significantly associated with aggressive disease features.

5. In Figure 3, I would like to see direct controls for effective NRAS transduction or transfection (also in Fig 4d, 5c for example - what is the NRAS level). How much is NRAS going up or down? The Megakaryocyte phenotype is unclear in 3e. The CST antibody for pErk is for p42,44 - there are usually 2 bands. If one looks up images of pErk Westerns the 2 bands is the convention on how pErk is shown. Can the authors show a larger portion of the blot? Are the right bands being selected? Is the tErk and pErk from the same membrane blot throughout the figures 3,4,5?

Reply: Thank you. Regarding direct controls of NRAS/KRAS transduction and transfection, as you are well-aware, this can be quite challenging especially in the context of primary samples. We have addressed this suggestion to the best of our ability:

- Due to the poor quality of available antibodies, direct quantification of RAS protein levels by Western blot was challenging. However, two representative blots have now been provided demonstrating efficiency of NRAS knockdown in patient samples (**Supplementary figure 3B**). Unfortunately, we were unable to successfully demonstrate KRAS knockdown by Western blot.
- An example of electroporation efficiency of a GFP-expressing construct into CMML patient-derived monocytes has been provided (**Supplementary figure 3A**).
- NRAS and KRAS knockdown/overexpression was confirmed by qPCR (**Figure 4D and 4F**). Levels of RAS expression by qPCR with overexpression have now been included in the modified **Figures 4E and G**.

We have provided a better image for the megakaryocyte morphology of the NRAS^{G12D} mouse in **Figure 3I**. This image shows a large mono-lobate megakaryocyte with open chromatin and atypia. The image has been blown up here:

As far as the pERK bands are concerned, we are convinced of the accuracy of the results presented. Because experiments were conducted by two independent lab members, different methodologies were employed. In some membranes, total ERK and phosphorylated ERK were probed for on two separate membranes in parallel. In others, membranes were probed for phosphorylated ERK followed by stripping and probing for total ERK. In both situations, bands were identified at the expected molecular weight. In those membranes that were stripped before re-probing, total and phosphorylated ERK bands overlapped. We have included examples of these blots with molecular weight markers to more clearly demonstrate this:

With regards to the two vs single bands in pERK, our findings are likely due to the use of 10 or 12% gels for electrophoresis causing merging of bands. If observed closely, there are likely two bands present in our blots such as in the blown up image displayed below:

We have also submitted our original Western blots along with review.

6. In Figure 4 and 5, it's not ideal to use total MNCs for RNA-seq and ChIP-seq. Although the authors present data in Supplementary Fig 4 to justify use of MNCs, I don't understand the argument with just the graph - were there no significantly variable genes between bulk and sorted? In the actual experiments - what were the % composition of the cell types in the analyzed samples - were the % Monocytes similar? Are the differences a result of differences in cell types? The authors should at least validate the RNA-seq results and targeted Chip-seq results in more purified populations.

Reply: Thank you very much for the comment. While we completely understand that it is not ideal to use total mononuclear cells for RNAseq and ChIP-seq studies, when we tried to sort bone marrow and blood for CD34+/CD38- progenitor cells, the attrition was extensive. From a starting volume of 50 million cells, we would often get down to values of 2 million or less, given that this is a chronic leukemia and not always enriched in CD34+ cells. This degree of attrition did not allow us to carry out ChIP-seq for the various marks that we have used in this manuscript, given that we need a minimum of 1 million cells per mark. In addition, additional studies such as DIP-seq and hematopoietic progenitor colony-forming assays were also hampered. We did not want to sort for CD14+ given that that is a terminal/mature monocyte marker, is largely present in the peripheral blood and not the bone marrow, and does not truly represent the progenitor population well. Hence, we used mononuclear cells. In our hands, based on assessing Ficolling techniques, greater than 95% of mononuclear cells obtained in patients with chronic myelomonocytic leukemia are composed of monocytes and monocyte precursors. The fraction of B,T lymphocytes and bone marrow stromal cells is very small (<5%).

Supplementary Figure 4A shows the detected transcripts by RNA-seq using unsorted MNCs versus CD34-sorted (left scatterplot) and unsorted MNCs versus CD14-sorted (right scatterplot). For both comparisons Spearman's rank correlation coefficient is above 0.90. This means there is a very tight positive correlation between each transcript measured in unsorted MNCs and the respective sorted cell population (pairwise correlation). These pairwise correlations are not driven by the large number of lowly- and non-expressed genes (correlation coefficients remain above 0.90 after excluding lowly- and non-expressed genes). Another way of looking at this is differential gene expression (as brought up by the reviewer). There is a small number of differentially expressed genes between CD34-sorted and unsorted MNCs (15 genes up-, 21 genes down-regulated, FDR < 0.05, volcano plot below). In our mind a certain degree of differential gene expression (i.e. a few dozen to hundred genes) is to be expected due to technical variation even when applying rigorous quality control standards and conservative correction for multiple hypothesis testing. We do not interpret these 36 differentially expressed genes as evidence of a systematic difference between unsorted and sorted cells. The heatmap in **Supplementary figure 4B** shows the expression of housekeeping

genes and select genes relevant to myeloid biology in CD34-sorted and unsorted samples. Again, there are no appreciable systematic differences between the two cell populations for these genes. Based on these results we decided not to employ further cell-sorting strategies, since the same answers in terms of gene expression can be obtained from unsorted MNCs.

In addition, we were partly able to validate our RNAseq results in CD14 sorted monocyte populations from the peripheral blood. **Supplementary figure 4E** demonstrates the statistically significant overexpression of *PLK1* in RAS mutant CD14-sorted pCMML vs RAS wild type dCMML and healthy controls. While there was a trend for *WEE1*, this did not reach statistical significance. We were not able to carry out ChIP-seq on these sorted cells due to the aforementioned attrition in the number of cells and the unavailability of sufficient chromatin to assess individual marks.

7. In Figure 4 are the ingenuity results selective? What is the list of all pathways, how were these 4 pathways selected - were they the only ones? The elevated *PLK1* and *WEE1* are derived from proliferative vs dysplastic CMML signatures. The authors then try to specify this is through *NRAS*. Is the data in 4e from *NRAS*-G12D into dCMML? They only specify CMML in the experiment design. What is the expression level of *WEE1* and *PLK1* in *NRAS* mutant dCMML? That *WEE1* and *PLK1* would be upregulated in proliferative cells is not unexpected. These are cell cycle related genes and would typically be increased in rapidly cycling cells. What is the expression level of *WEE1* and *PLK1* in pCMML that is *JAK2*VF mutant or pCMML without *NRAS* mutations compared to dCMML? Would the inhibitors volasertib and be active in pCMML without *NRAS* mutation or in *NRAS* mutant dCMML?

Reply: We apologize for the confusion. In **Figure 4E**, mutant *NRAS* is transfected into RAS pathway wild type dCMML patient samples. We have corrected this in the figure legends. We also demonstrate that the expression levels for the mitotic checkpoint kinases *PLK1* and *WEE1* is significantly lower in *NRAS* mutant dCMML, in comparison to *NRAS* mutant pCMML. This data has been added to **Figure 4C**. We have also carried out work demonstrating that these mitotic checkpoint kinases are not elevated in *JAK2* mutant pCMML, in comparison to *NRAS* mutant pCMML (**Figure 4C**). We have also carried out knock down experiments using *JAK2* siRNA in *JAK2* mutant pCMML and show that it does not significantly impact the expression of the mitotic checkpoint kinases, nor does it affect the occupancy/enrichment of H3K4me1 and KMT2A at the gene promoters of these mitotic checkpoint kinases. We have also provided a list of all pathways significantly upregulated in pCMML in **Supplementary Figure 4D** (ingenuity pathway analysis). Prominent pathways include mitotic cell cycle processing, mitotic cell cycle, cell division and chromosome segregation. All of these pathways did converge on mitotic check point kinases namely *PLK1* and *WEE1*.

Given that all our data strongly supports a clonal role for RAS mutations in the RAS-KMT2A-*PLK1* axis, we do not think that *PLK1* inhibitors would be effective in pCMML without RAS pathway mutations (**Figure 7B**) or in dCMML with RAS pathway mutations.

8. In Figure 5, the authors show that H3K4me1 is upregulated in pCMML with *NRAS*. Were input sequences used to call the PEAKS with MACS2? It's not clear from the methods. What is the consistency between samples- 18 Ras pathway and 12 dCMML? As these are MNC PB and BM samples, I would suspect there would be heterogeneity. How is this taken into account or dealt with? The authors are showing us only 1 track for each ChIP experiment and this may be quite selective. The track for *Wee1* in 5B does not show the promoter only the gene body. There

should be more summary data of which peaks were called and what was common between samples. Perhaps a heatmap could be included of the peaks or Venn diagram. Can the authors add a track into their ChIP seq plots showing where peaks were called and where the differential peaks were called? Can the authors show in a correlation between the RNA-seq and the H3K4me1 promoter peaks for other genes? Is this a general trend in which H3K4me1 peaks are directing upregulation of multiple target genes?

Reply: Thank you very much. Input sequences were used to call the peaks with MACS2. We have added this to the methodology section. The 18 RAS pathway mutant pCMML samples and the 12 RAS pathway wild type dCMML samples were all represented by mononuclear cells obtained from bone marrow biopsies (BM MNC). While the samples do show variability amongst themselves (**Figures 5C**), we do provide data for those peaks that appear in the majority of samples, including *PLK1* and *WEE1* in pCMML (excel spreadsheet attached).

With regards to the question of one track for each ChIP experiment, please note that the track demonstrated in **Figure 5** represents an average of 8 RAS mutant pCMML BM MNC and 8 RAS wild type dCMML MNC and is not a single tract. We have explained this in the figure legend. We now also show the called peaks along each gene and a list of genes that are called in the majority of samples (which include *PLK1* and *WEE1*). Unlike in *PLK1* which exhibited dominant H3K4me1 peaks ~1 kb upstream of the TSS; for *WEE1*, multiple peaks were observed up to 5 kb 5' to the TSS. However, combined enrichment for all samples indicates increased enrichment of H3K4me1 in RAS mutant pCMML samples, in comparison to RAS wild type dCMML sample (**Figure 5**). We have modified the track for *WEE1* to demonstrate the promoter region (**Figure 5E**).

We have now provided a Venn diagram that shows the peaks that were unique to dCMML and pCMML patients, as well as a Venn diagram that shows peaks found in the majority of samples (**Supplementary Figure 5G**). We have added a track into the ChIP-seq plots, demonstrating where peaks were called (**Figure 5E**). In addition, we have correlated the RNAseq with occupancy of H3K4me1 at the gene promoters. We provide this data as an excel spreadsheet in the supplementary material, as well as a Venn diagram and a volcano plot (**Figure 5B**). Importantly, *PLK1* and *WEE1* were among the genes enriched in H3K4me1. We do confirm that in general, H3K4me1 peaks are directing upregulation of multiple target genes.

9. Again, is the *KMT2A* over-expression just in *NRAS* mutant pCMML. What are the expression levels in other mutants or pCMML with *NRAS* pathway mutations? What is the expression in dCMML that has a RAS pathway mutation?

Reply: *KMT2A* overexpression was only seen in the RAS pathway mutant pCMML in comparison to controls. This was not specific to *NRAS* and applied to all RAS pathway genes. We did not see a significant difference between *NRAS* mutant pCMML and *NRAS* mutant dCMML. We have modified our manuscript and have deleted the word overexpressed from *KMT2A*. We primarily chose to study *KMT2A* given that we saw a global and sequence specific increase in H3K4Me1 in RAS mutant pCMML and the fact that by ChIP-PCR we saw an increased enrichment of *KMT2A* and *MEN1* at the promoters of *PLK1* and *WEE1*.

Reviewer # 3:

1. RAS pathway mutations are prognostic in an univariate model. What factors are most important in a multivariate model of survival and leukemic transformation (LT) in CMML?

Reply: Multivariate analysis using genetic and clinical variables has now been performed using the L1-regularized logistic regression model. Please see response to reviewer 1 for details.

2. Does VAF of RAS pathway mutation as a continuous variable trump the prognostic impact of arbitrary difference between dCMML and pCMML based on WBC count

Reply: We thank the reviewer for this suggestion. We obtained VAF_{NRAS} information for 124 patients (14 Mayo + 59 French + 51 Austrian) and VAF_{RAS} information for 336 patients (35 Mayo + 172 IGR + 129 Austrian). To assess the impact of VAF in NRAS or grouped RAS genes on clinical variables and outcomes we performed two analyses. Firstly, we correlated VAF_{RAS} as a continuous measure of subclonality in relationship to clinical parameters. We found that AMC (absolute monocyte count) was significantly associated with VAF_{NRAS} (Spearman correlation test $p=1.8E-02$; $r = 0.22$). We then assessed the same considering cumulative RAS pathway mutations. Principle component 1 (PC1), WBC, AMC and PLT were significantly associated with VAF_{RAS} .

We then performed ROC analysis, which demonstrated that $VAF_{NRAS} = 0.3$ was the most informative for predicting pCMML (≥ 0.3) versus dCMML (< 0.3). We were also able to apply a similar analysis to cumulative RAS pathway mutations and demonstrate that a VAF_{RAS} of > 0.19 was the most informative for predicting pCMML v dCMML. We have added these statements and figures to the manuscript (Figures 3C-F).

We then used these VAF thresholds to separate mutations into clonal and subclonal to assess associated clinical parameters. For the *NRAS* mutations this separation resulted in 87 clonal and 37 subclonal mutations. Lower platelet counts were significantly associated with clonal mutations (Mann-Whitney test, $p = 2.5E-02$; mean clonal – 113.48, mean subclonal 186.89; **Supplementary figure 3G**). There was a trend for other aggressive clinical/phenotypic features with clonal *NRAS* mutations; however these did not reach statistical significance.

A similar picture was observed for all cumulative RAS pathway mutations (230 clonal and 106 subclonal). Lower platelet counts were significantly associated with clonal mutations (Mann-Whitney test, $pval = 3.4E-03$; mean clonal: 118.46 mean subclonal: 159.99; **Supplementary figure 3I**). Additionally, a higher percentage of peripheral blood blasts were seen with clonal mutations (Mann-Whitney test, $pval = 2.1E-02$; mean clonal: 1.41 mean subclonal: 1.15; **Supplementary figure 3H**). While other aggressive clinical/phenotypic features were seen in the context of clonal mutations, these did not reach statistical significance.

We next tried to assess the usefulness of VAF_{RAS} compared to discrete RAS mutational status so as to improve the ability to predict proliferative versus dysplastic CMML phenotypes. When we used the continuous VAF_{RAS} value as a predictor, in comparison to a binary RAS mutation status, we found that the AUC characteristic of the ROC curve improved slightly (0.671 vs 0.641), which may indicate the usefulness of continuous VAF_{RAS} (**Figures 3E-F**).

3. In the mouse model of CMML, did the mice acquire non RAS pathway mutations as disease develops?

Reply: While we agree this is an excellent and intriguing question, due to limitations in resources we were not able to assess mouse exomes to accurately answer this question. We apologize.

4. In Fig 4 it may be useful to show the hierarchy of pathways that are upregulated. That will convince the reader about authors selecting to target PLK1/WEE1.

Reply: Thank you we have now added the hierarchy of pathways up regulated in **Supplementary figure 4D**.

5. Did authors establish mechanistic link between mutant RAS and transcriptional upregulation of KMT2A?

Reply: We have not established a mechanistic link between mutant RAS and the transcriptional up regulation of *KMT2A*. We have applied for additional funding mechanisms to carry out this aim. In future studies, we would like to look at various transcription factors (primarily identified by motif and expression analysis) that we have preliminarily identified as candidate regulators of KMT2A including FOS, JUN and GATA2.

6. Does KMT2A knockdown phenocopy RAS knockdown in patient samples as regards colony formation etc?

Reply: KMT2A knock down phenocopies RAS knockdown in patient samples with regards to colony formation. We have added these results to the **Supplementary Figure 6E**.

7. While PLK1 inhibition reduces disease burden in the mouse model of RAS mutant CMML, authors need to show if that impacted survival in mice.

Reply: We agree that this is a very valid point. However, given the SARS-CoV-2 pandemic, our vivariums and their staffing have been greatly impacted. We will not be able to carry out survival studies for the foreseeable future and request an exemption to this point.

Reviewer # 4:

1. Figure 4: selection of PLK1 and WEE1. It is not clear to me why the authors selected these 2 genes in Figure 4. I guess that fold-change of mRNA expression in pCMML as compared to dCMML were about x2.5-3.5 in Figure 4C. However, there are many other genes which showed higher fold-changes in Figure 4B. Also, although they mentioned several other genes (e.g., AURKA, CHEK1, CDC45, etc) on page 8, it is not clear what is the rationale focusing on PLK1 and WEE1 among these genes. I understand that PLK1 may be druggable as they showed in Figure 7. But, I strongly worry if the authors might be missing many other important genes regulated by RAS. Hence, current approach is somewhat biased. Please also see my comments 2.4 and 3.

Reply: We agree and are well aware that other genes with higher levels of expression, relative to *PLK1* and *WEE1*, may also be important and are currently pursuing different funding mechanisms to explore their role in chronic myelomonocytic leukemia. The reasons for selecting *PLK1* and *WEE1* are as follows:

- a. These genes were clearly up regulated in pCMML with RAS pathway mutations, in comparison to dCMML or pCMML forms without RAS pathway mutations, including JAK2 mutant pCMML (**Figure 4C**).
- b. *PLK1* and *WEE1* are genes directly downstream of RAS, thus giving us optimism with regards to their inhibition, decreasing survival of mutant cells due to an inherent dependency. In a seminal paper by Luo J et al., it was clearly shown that RAS mutant cell lines (DLD-1 and HCT116: Colorectal cancer cells lines) were hypersensitive to inhibition of PLK1 function¹². BI-2536, a small molecule PLK1 inhibitor used in the study showed dramatic effects in RAS mutant cell lines. In addition, RAF1/CRAF has been shown to have a direct interaction with PLK1. Milego et al demonstrated that RAF1 gets phosphorylated on serine 338 and localizes to the mitotic spindle of proliferating tumor cells in vitro and in murine tumor models or biopsies from cancer patients. Treatment of tumors with allosteric, but not ATP-competitive RAF inhibitors prevented RAF1/CRAF phosphorylation on serine 338, localization to the mitotic spindle and caused cell cycle arrest at pro-metaphase. Mechanistically, RAF1 was found to associate with PLK1 at the centrosomes and spindle poles during G2/Mitosis. Indeed, allosteric or genetic inhibition of phospho-S338 RAF1 impaired PLK1 activation and accumulation at kinetochores causing pro-metaphase arrest, while a phospho-mimetic S338D /RAF1/CRAF mutant potentiated PLK1 activation, mitosis and tumor progression in mice¹³.
- c. Very importantly, in the field of myeloid leukemia, Volasertib the PLK1 inhibitor had already completed extensive pre-clinical and clinical trial testing (not selected for RAS mutations and not in CMML). In fact, at the time of writing this paper the phase 3 trial of Volasertib had been completed in acute myeloid leukemia, providing us with significant data with regards to safety, efficacy and tolerability. This provided us with strong rationale to assess PLK1 as a target, given that a lot of safety and dosing data already existed for this drug¹⁴⁻¹⁸.
- d. The same applies for *WEE1*, as at the time of writing this paper, at Mayo Clinic there was a clinical trial assessing the safety and efficacy of cytarabine with a *WEE1* inhibitor (AZD-1775) in myelodysplastic syndromes and acute myeloid leukemia. Since the dosing regimen and safety data were already available, we felt that this would be an important target to explore in CMML.

2. Figure 5: Interpretation of ChIP-seq results. The authors emphasized a global enrichment of H3K4me1 in pCMML samples as compared to dCMML cases. As examples, they showed baseline H3K4me1 signals at *PLK1* and *WEE1* gene loci. They also performed NRAS knockdown, which showed a significant downregulation of H3K4me1 enrichment by ChIP-PCR. I think knockdown data is convincing. However, I have some concerns about their interpretation of ChIP-seq. First, they used MACS2 for the peak calling. In my understanding, peak is shown either “called” or “uncalled” from this analysis. Although the authors did not describe full details of this analysis, my guess is that they performed peak calling for each of CMML samples independently, evaluated the total number of called peaks, and they showed the average number in Figure 5A. Can they clarify this point? But, this approach is still slightly tricky. One potential pitfall is that peak calling is often affected by the level of baseline signal, which could be affected by experimental and sample conditions, especially when handling primary samples. Indeed, I see very broad peaks with high baseline signals at *PLK1* and *WEE1* gene loci for both

pCMML and dCMML samples. I wonder which regions were actually detected by peak calling? Also, how many of dCMML and pCMML samples were called at these elements?

Reply: We apologize for not being very clear with our methods of peak calling. As mentioned above and in response to prior reviewer questions, we have now outlined in detail how we have used MACS2 for peak calling. We did not assess epigenetic enrichment differences that may be significant where genes were identified in both populations. However, we do provide data and lists for all genes called in at least one sample (comprehensive) or in the majority of samples (consensus), as well as a breakdown of where these peaks appeared (**Figure 5**).

With regards to H3K4me1 we have called this as a narrow peak. We have now modified **Figure 5E** to demonstrate peaks called in majority of samples by a track demarcated underneath the peaks. *WEE1* and *PLK1* were called in the majority of pCMML samples (80%) and in a minority of dCMML samples.

3. Alternatively, the authors may show metagene plots to evaluate the difference in average signal level between pCMML and dCMML samples, as they actually did for DNA methylation analysis. However, this data is currently not included. Can they perform metagene analysis for H3K4me1, H3K4me3 or H3K27me3 ChIP-seq data? Do they actually observe significant differences in H3K4me1?

Reply: Thank you. We have now provided metagene plots evaluating the difference in average signal levels between pCMML and dCMML. This has been included in **Supplementary Figure 5A-F**. We show that the number of reads is equivalent between proliferative and dysplastic CMML patients, that this translates to fewer peaks called for H3K4me1 enrichment in dCMML samples and that the peaks are much less consistent (and reduced at promoters) in dCMML samples. This data has been included in **Figure 5** and **Supplementary Figure 5**.

4. What is the definition of “promoter”? In the first paragraph on page 10, they described “enrichment of H3K4me1 at promoters of *PLK1* and *WEE1*”. Which region is the promoter they defined? A confusing is that H3K4me1 signal level at the transcriptional start site and its immediate upstream of the *WEE1* gene (which I think the promoter) is even low, while higher signals is observed above exons 2-4 of this gene in pCMML samples. I do not think this intragenic element is usually called promoter.

Reply: We have provided evidence now for conservative (+/- 1 kb from the transcriptional start site [TSS]) and liberal (+/- 5 kb from the TSS) definitions of promoters as these two distances have been used extensively in the past. In addition, our data shows a dispersion of enrichment away from the TSS in most dCMML samples. One interesting finding is that the enrichment of H3K4me1 is much less consistent in dCMML samples (**Figures 5C-D**).

5. Are *PLK1* and *WEE1* one of most significant examples that show difference in H3K4me1 by ChIP-seq? Please also see my comment 3 below.

Reply: We have provided of a volcano plot demonstrating the cross-section between the RNAseq and the ChIP-seq data sets (**Figure 5B**). In the up regulated genes that have enrichment of H3K4me1, *PLK1* and *WEE1* are prominent, but not the most significant examples in the data set. Regardless, given the aforementioned reasons

(Response to Comment 1), we selected *PLK1* and *WEE1* for for the study, given that they were attractive targets for inhibition in the clinical management of proliferative, RAS pathway mutant pCMML.

6. Integration of RNA-seq and ChIP-seq. Although the authors have excellent RNA-seq and ChIP-seq datasets, ChIP-seq data was currently used only to highlight the *WEE1* and *PLK1* gene loci. These two datasets were not integrated in an unbiased way. For example, can they first select genomic loci which show significant difference in H3K4me1 signal based on ChIP-seq, and then filter the genes which are significantly upregulated in pCMML samples based on RNA-seq? If they rank the selected genes by fold-change values for RNA expression, can *PLK1* and *WEE1* still be selected as one of top hits?

Reply: We have integrated the RNAseq and ChIPseq data and do show that *PLK1* and *WEE1* are indeed in the top list of druggable targets in myeloid neoplasms. We provide a newly prepared volcano plot demonstrating this (**Figure 5B**) and also provide the list of genes in the **Supplementary Data**.

7. Also, can they include supplemental tables showing fold-change and p-values for all genes selected by RNA-seq and the list of genes called by ChIP-seq? I think these data are needed as supporting evidences.

Reply: We have now provided this information as requested.

8. Figure 6: Selection of *KMT2A*. Was *KMT2A* one of top hits selected by RNA-seq analysis? Is this gene element also associated with higher level of H3K3me1 signal in pCMML samples?

Reply: Thank you. *KMT2A* was overexpressed in pCMML samples when compared to normal controls (**Figure 6A**). This however was not a top hit among overexpressed genes. Regardless, we chose to assess *KMT2A* given that it is a major lysine methyltransferase responsible for methylation of the histone 3 lysine 4 mark. Since we saw significant global increases in H3K4me1 (**Figure 5A**), we felt that *KMT2A* would be an important mediator, warranting further investigations.

9. The difference between *KMT2A* ChIP and IgG control is only about 2-fold enrichment. Can they perform ChIP for other *KMT* proteins to show the specificity?

Reply: As suggested, we have carried out ChIP-qPCR studies for *KMT2C* and *KMT2D* and demonstrate that the findings described in this paper are specific to *KMT2A* (**Supplementary Figure 6H**). On knocking down *KMT2C* and *2D* there were no changes in occupancy of H3K4me1 at the promoters of *PLK1* and *WEE1*. Hence, although these lysine methyltransferases are known to monomethylate H3K4, in RAS mutant pCMML, this is largely mediated by *KMT2A*. We also agree that while the *KMT2A* ChIP and the IgG controls only showed 2 fold enrichment, this however was statistically significant and importantly when we knocked down *NRAS/KRAS* in RAS mutant pCMML samples we were able to see statistically significant changes in the occupancy of *KMT2A* at the gene promoters. Additionally significant changes in *KMT2A* enrichment were also seen on transfecting RAS wild type samples with mutant *NRAS/KRAS*.

REFERENCES:

1. Kratz CP, Schubbert S, Bollag G, Niemeyer CM, Shannon KM, Zenker M. Germline mutations in components of the Ras signaling pathway in Noonan syndrome and related disorders. *Cell Cycle*. 2006;5(15):1607-1611.
2. Lipka DB, Witte T, Toth R, et al. RAS-pathway mutation patterns define epigenetic subclasses in juvenile myelomonocytic leukemia. *Nat Commun*. 2017;8(1):2126.
3. Niemeyer CM. RAS diseases in children. *Haematologica*. 2014;99(11):1653-1662.
4. Pylayeva-Gupta Y, Grabocka E, Bar-Sagi D. RAS oncogenes: weaving a tumorigenic web. *Nat Rev Cancer*. 2011;11(11):761-774.
5. Rauen KA. The RASopathies. *Annu Rev Genomics Hum Genet*. 2013;14:355-369.
6. Buradkar A, Bezerra E, Coltro G, et al. Landscape of RAS pathway mutations in patients with myelodysplastic syndrome/myeloproliferative neoplasm overlap syndromes: a study of 461 molecularly annotated patients. *Leukemia*. 2020.
7. Patnaik MM, Pophali PA, Lasho TL, et al. Clinical correlates, prognostic impact and survival outcomes in chronic myelomonocytic leukemia patients with the JAK2V617F mutation. *Haematologica*. 2019;104(6):e236-e239.
8. Itzykson R, Kosmider O, Renneville A, et al. Prognostic score including gene mutations in chronic myelomonocytic leukemia. *Journal of clinical oncology : official journal of the American Society of Clinical Oncology*. 2013;31(19):2428-2436.
9. Merlevede J, Droin N, Qin T, et al. Mutation allele burden remains unchanged in chronic myelomonocytic leukaemia responding to hypomethylating agents. *Nature communications*. 2016;7:10767.
10. Patnaik MM, Itzykson R, Lasho TL, et al. ASXL1 and SETBP1 mutations and their prognostic contribution in chronic myelomonocytic leukemia: a two-center study of 466 patients. *Leukemia*. 2014;28(11):2206-2212.
11. Patnaik MM, Pierola AA, Vallapureddy R, et al. Blast phase chronic myelomonocytic leukemia: Mayo-MDACC collaborative study of 171 cases. *Leukemia*. 2018.
12. Luo J, Emanuele MJ, Li D, et al. A genome-wide RNAi screen identifies multiple synthetic lethal interactions with the Ras oncogene. *Cell*. 2009;137(5):835-848.
13. Mielgo A, Seguin L, Huang M, et al. A MEK-independent role for CRAF in mitosis and tumor progression. *Nat Med*. 2011;17(12):1641-1645.
14. Dohner H, Lubbert M, Fiedler W, et al. Randomized, phase 2 trial of low-dose cytarabine with or without volasertib in AML patients not suitable for induction therapy. *Blood*. 2014;124(9):1426-1433.
15. Gjertsen BT, Schoffski P. Discovery and development of the Polo-like kinase inhibitor volasertib in cancer therapy. *Leukemia*. 2015;29(1):11-19.
16. Janning M, Fiedler W. Volasertib for the treatment of acute myeloid leukemia: a review of preclinical and clinical development. *Future Oncol*. 2014;10(7):1157-1165.
17. Ottmann OG, Muller-Tidow C, Kramer A, et al. Phase I dose-escalation trial investigating volasertib as monotherapy or in combination with cytarabine in patients with relapsed/refractory acute myeloid leukaemia. *Br J Haematol*. 2018.
18. Rudolph D, Impagnatiello MA, Blaukopf C, et al. Efficacy and mechanism of action of volasertib, a potent and selective inhibitor of Polo-like kinases, in preclinical models of acute myeloid leukemia. *J Pharmacol Exp Ther*. 2015;352(3):579-589.

Reviewers' comments:

Reviewer #1 (Remarks to the Author):

In this revised manuscript, the Authors have addressed most of the concerns raised by the previous version. However, they did not address the issue of the effect of clonal (dominant) versus subclonal RAS mutations on disease phenotype and outcome, as well as on the expression of the mitotic kinases (PLK1 and WEE1), which is relevant to confirm the specific association between RAS mutations and activation of the KMT2A-PLK1 axis, and to get insight on variables that may affect signalling and sustain phenotype variability.

Reviewer #2 (Remarks to the Author):

In this revision, the authors have addressed many of the important issues raised to their initial submission. They have provided details validating the comparability of the different cohorts. They provide multivariate analysis of Hazard Ratios. This is a large data cohort to validate and determine mutational risk in patients. They have added experiments with NRASwt pCMML (specifically with JAK2 mt pCMML) to demonstrate specificity with PLK1 and WEE1 expression. They provide better context for the selection of these genes. The model provides pre-clinical validation for volasertib.

My main remaining concern is the ChIP-seq data. The authors do provide better description in Fig5C and D. But based on the pile-up plots in 5E, there appears to be a similar pattern of peaks and troughs between dCMML and pCMML. Where the authors are calling peaks in the pCMML PLK1 track, there appears to be more relative reads also in dCMML. Were there peaks called in dCMML track for PLK1 and WEE1? Did the delta plots in 5E normalize for read counts? The authors should be explicit on what peaks were called in individual samples or on summary in the dCMML track for PLK1 and WEE1. If there was no consensus peak for WEE1, how are they able to do the site-specific ChIP-qPCR experiments in Fig5F-I and Fig6?

The authors state they ran DiffBind in the methods, how did the dCMML and pCMML cluster or display on a PCA plot based on p/d and RAS mut/wt?

The Metagene analysis in S5D supports the differences and is a main finding that should be included in main figures rather than supplement.

Minor:

The authors do make new contributions regarding the role of RAS mutations to pCMML, but the introduction gives the impression this is completely new. The authors should acknowledge / cite the prior work regarding RAS mutations in pCMML in the introduction. This would provide more appropriate context for their work.

In figure S4G is the STAT3 blot for total STAT3 or pSTAT3, would not expect tSTAT3 to change so much.

Figure legends are missing for S3J, S4F. The legend for S4F is about qPCR results and is for S4G. There is a typo "including" in the figure legend S5D-F.

I do not understand the description of S5G ii, iii - if they are only in pCMML or dCMML why is there an overlap in the Venn diagram?

The statement regarding "most significantly associated" is still in the text in Figure 7.

I appreciate the authors including the statement regarding NF1 in the commentary. However, in Figure S1B, NF1 appears to not to be so rare, and also appears more common in the dysplastic type than the proliferative. Perhaps this is due to the lower sample numbers?

In 3I picture given, the Megakaryocyte circled in the top left does not seem mono-lobated, the center one does. It would be more clear if enlarged pictures of the Megas are included for both WT and Mutant in the main figures.

I'm unsure what the metagene analysis in S5A-C is trying to show. What is the plotted curve showing? I gather the plots are based on gene-RNA seq and colors are for the ChIP-seq "epigenetic enrichment" data. Is the logFC calculated by dCMML / pCMML? Is it surprising that the H3K27me3 plot (a general marker of gene repression), should have the same pattern as the H3K4me1/3 plots (a general marker of gene activation)? A more detailed figure legend would be helpful.

Reviewer #3 (Remarks to the Author):

The authors have responded effectively to my comments

I have two minor comments

1. Suppl. Fig 4A: why did they decide on 5 samples each from different patients for pairwise correlation of expression. Would it not be more convincing using same sample source to compare among defines subpopulations

2. Suppl Fig 5J: Distribution of 5mc.5hmc at TSS is not enough evidence that RAS and not TET2 mutation is responsible for the epigenetic phenotype in pCMML. For on 5mc/5hmc distribution in TET2 mutant context may not be all related to upstream occupancy but also to genebody occupancy.

Also PLK1 and Wee1 are not the only and top hits for differential expression in pCMML vs dCMML

Reviewer #4 (Remarks to the Author):

The authors tried to address the concerns previously raised by myself and other reviewers. I appreciate their efforts. Indeed, some new data such as metagene analysis have addressed one of my comments and improved the quality of this manuscript. However, their answers to many of my comments were indirect or raised more concerns, in particular, about their interpretation on ChIP-seq data. Please see my specific comments to the authors.

I think that the first part (analysis of clinical samples, and mouse models) was very nicely done in an unbiased manner, although the second part (PLK1, WEE1, KMT2A) was

somewhat biased. Hence, inclusion of the second part would give a negative impression, which might reduce overall quality of this paper.

Critically, I requested the authors to provide objective evidences to show how they selected PLK1 and WEE1 genes. The authors provided additional evidences (Figure 5B, a table “intersection”) after I requested and the editor reminded, which demonstrated that PLK1 and WEE1 were not one of the highest confidence targets. I have additional concern about ChIP-seq data and their representation, which is a critical evidence as one of selection criteria. Although they stated the rationale for choosing PLK1 and WEE1 by wording, I still feel that there is lack of objective evidences to support their justification. I worry if this paper may receive claims or correspondences about ChIP-seq data after being published as the current form.

Specific comments:

1. Selection of PLK1 and WEE1

The authors described the rationale for the selection of PLK1 and WEE1 genes by wording. I understand these points. I would agree with it, if it is supported by RNA-seq and ChIP-seq data. However, I still have many concerns.

The authors provided a table (“intersection”) via email, though I originally requested them to include in Supplemental Information as a direct evidence to support their justification. Please officially include it as a Supplemental Table, if they are confident about these data. Basically, these data indicate that PLK1 and WEE1 were selected by RNA-seq and ChIP-seq but were not the highest confidence targets in terms of fold-change or ChIP-seq profile.

Thus, I tried to investigate the quality and validity of their ChIP-seq analysis. They explained that they did a peak-calling by MACS2, which is a standard method. They replied that they have indicated peaks in red boxes along the gene track images in Figure 5E. There are two red boxes below pCMML track at PLK1. Otherwise, they did not further provide the peak calling result or images for each individual sample or did not comment how they evaluate statistical differences between dCMML and pCMML samples. Also, they mentioned that “WEE1 and PLK1 were called in the majority of pCMML samples (80%) and in a minority of dCMML samples”, but I could not find this data or such description in the manuscript. “Minority” is a very non-scientific and ambiguous term.

The most confusing is that there were no red boxes at the WEE1 gene, which may mean that peaks were even not called or not significant for this gene? If yes, why this gene was selected by ChIP-seq? Although they have performed knockdown experiment to show the difference in H3K4me1 peak, it is a different setting and thus indirect evidence. Thus, from their responses, I understood that the authors failed to answer to my question in a straightforward manner.

Additionally, it is not clear how they performed ChIP-seq experiments. According to the entry in the GEO dataset they deposited, they described “Pooled all sample in 1:1:2 ratios for H3K4Me1:H3K4me3:H3K27Me3”. But, these information were not described in the method. I guess that they performed ChIP-seq using different barcodes? How can other researchers reproduce their result? I have a strong concern in terms of the transparency.

2. Definition of promoters

The authors explained about their definition of promoters (“conservative” and “liberal”).

However, they did not answer to my original question. Which region is the promoter they defined for PLK1 and WEE1 genes? I am still not clear which element they are mentioning. As I commented above, there is no peak shown in red boxes at the WEE1 gene element. Because this is one of criteria by which they selected WEE1 gene (in Figure 5B, and a table), more detailed data are needed. This is a critical evidence that they have to show to convince reviewers and readers.

3. KMT2A ChIP

My original comment was on the quality of KMT2A ChIP. I suggested to do ChIP-PCR for KMT2B and KMT2C at this element but not H3K4me1. But, the authors tried to address this comment by performing a H3K4me1 ChIP-PCR after knockdown of KMT2C and KMT2D. This is an indirect evidence (it could be ignoratio elenchi). If they cannot perform ChIP for KMT2B or KMT2C, can they use their KMT5A ChIP samples for ChIP-seq to find if WEE1 and PLK1 gene loci are actually called?

I am very cautious to conclude based on the statistical significance, because the difference between KMT2A ChIP-PCR and IgG can be significant as long as the deviation among experimental replicates is tight. They also did not describe whether they did experiments in technical replicates (by PCR) or biological replicates (by different experiments). Thus, it is difficult to evaluate the validity of their experiments.

POINT-BY-POINT RESPONSE

Reviewer 1

1. In this revised manuscript, the Authors have addressed most of the concerns raised by the previous version. However, they did not address the issue of the effect of clonal (dominant) versus subclonal RAS mutations on disease phenotype and outcome, as well as on the expression of the mitotic kinases (PLK1 and WEE1), which is relevant to confirm the specific association between RAS mutations and activation of the KMT2A-PLK1 axis, and to get insight on variables that may affect signalling and sustain phenotype variability.

Response: We have provided a detailed analysis demonstrating the clinical impact of clonal versus sub-clonal RAS mutations on the disease phenotype and outcomes (**Figure 1H-I, Figure 3C-F, Supplementary 3 G-I**). Importantly, clonal RAS mutations were associated with higher peripheral blood blasts and lower platelet counts; both of which are known adverse prognosticators for this disease. We now provide **Figure 4E**, which demonstrates a significant increase in PLK1 expression with an increase in the variant allele frequencies of RAS pathway mutations. **Together, our data supports the fact that targeting PLK1 is meaningful in patients with proliferative disease, where the RAS pathway mutations are clonal/dominant events.**

Reviewer 2

In this revision, the authors have addressed many of the important issues raised to their initial submission. They have provided details validating the comparability of the different cohorts. They provide multivariate analysis of Hazard Ratios. This is a large data cohort to validate and determine mutational risk in patients. They have added experiments with NRASwt pCMML (specifically with JAK2 mt pCMML) to demonstrate specificity with PLK1 and WEE1 expression. They provide better context for the selection of these genes. The model provides pre-clinical validation for volasertib.

Response: We thank the reviewer for this comment. Based on this pre-clinical validation, we are in the process of launching a phase I/II clinical trial, testing the safety and efficacy of PLK1 inhibition in RAS mutant proliferative chronic myelomonocytic leukemia.

1. The authors do provide better description in Fig5C and D.

Response: The ChIP-seq data has been re-analyzed. Consequently, the previous C and D panels of Figure 5 have since been removed.

2. But based on the pile-up plots in 5E, there appears to be a similar pattern of peaks and troughs between dCMML and pCMML

Response: We thank the reviewer for suggesting a more clear analysis of this data. As indicated above, we have completely redone the ChIP-seq analysis and have strictly abided by ENCODE criteria for assessment of histone marks. We have addressed this query specifically by demonstrating individual tracks for patient samples and the consensus peak calling in pCMML and dCMML (**Figure 5D**). The presented patient samples were selected after restricting for those with good sequencing depth and adequate number of reads (per ENCODE criteria - <https://www.encodeproject.org/chip-seq/histone/>). Consensus peaks were called as broad peaks for H3K4me1 per ENCODE criteria. The indicated consensus peaks for both dCMML and pCMML samples are graphically indicated by colored bars between the individual tracks in **Figure 5D**. You will see that the consensus peaks called are different in size between dCMML and pCMML samples. Nonetheless, we agree that there is generally a similar peak and trough pattern between samples. However, we now demonstrate a statistically significant increase in the peak heights for H3K4me1 in pCMML relative to dCMML that is not accounted for by differences in relative reads between samples.

3. Where the authors are calling peaks in the pCMML PLK1 track, there appears to be more relative reads also in dCMML.

Response: Please see our response above.

4. Were there peaks called in dCMML track for PLK1 and WEE1? Did the delta plots in 5E normalize for read counts?

Response: Please see our response above. Given that we have now presented individual patient tracks, we no longer provide delta plots.

5. The authors should be explicit on what peaks were called in individual samples or on summary in the dCMML track for PLK1 and WEE1. If there was no consensus peak for WEE1, how are they able to do the site-specific ChIP-qPCR experiments in Fig5F-I and Fig6?

Response: Thank you very much. The peaks called at the promoters of PLK1 are now more explicitly explained in our response to point 2 above. We completely agree with the criticism of our WEE1 data, which has now been removed from this manuscript.

6. The authors state they ran DiffBind in the methods, how did the dCMML and pCMML cluster or display on a PCA plot based on p/d and RAS mut/wt?

Response: We have provided the DiffBind data showing a differential increase in H3K4me1 peaks in pCMML versus dCMML (**Figure 5B**). Due to sample heterogeneity, the PCA plot did not show significant clustering based on RAS pathway mutations, which could be due to the heterogeneous mutational make up of individual patients.

7. The Metagene analysis in S5D supports the differences and is a main finding that should be included in main figures rather than supplement.

Response: Thank you. Due to inclusion of the DiffBind analysis and limited space in the main figure, we request that this be allowed to remain in the supplementary figure.

8. I do not understand the description of S5G ii, iii - if they are only in pCMML or dCMML why is there an overlap in the Venn diagram?

Response: We apologize for the confusion. Given our extensive re-analysis, we have removed these figures.

9. I'm unsure what the metagene analysis in S5A-C is trying to show. What is the plotted curve showing? I gather the plots are based on gene-RNA seq and colors are for the ChIP-seq "epigenetic enrichment" data. Is the logFC calculated by dCMML / pCMML? Is it surprising that the H3K27me3 plot (a general marker of gene repression), should have the same pattern as the H3K4me1/3 plots (a general marker of gene activation)? A more detailed figure legend would be helpful.

Response: This is correct and the enrichment pattern is reflecting whether higher H3K4me3 or H3K27me3 are associated with increased or decreased gene expression, respectively. The patterns confirm this association. As you move from left to right, you would expect to have increased expression for H3K4me3 and decreased expression for H3K27me3, which is what we observe. In the case of H3K27me3, there is little change in gene expression patterns, hence few, if any, red dots appear. We have expanded the figure legend to reflect the same.

10. Is the logFC calculated by dCMML / pCMML?

Response: That is correct. We have added this to the method section

11. Is it surprising that the H3K27me3 plot (a general marker of gene repression), should have the same pattern as the H3K4me1/3 plots (a general marker of gene activation)? A more detailed figure legend would be helpful.

Response: It is unclear that H3K4me1 is only marking active genes, and H3K4me3 is not as consistent as H3K27Ac in showing differences between expressed and non-expressed genes. The pattern of dots (colour and localization) is not the same. The line is a LOESS curve which reflects a running means generalization for scatter plot data. If the marks cover similar mean genome coverage, then the line of best fit would be similar in both cases. Since these are cancer cells (and undifferentiated) it is unclear whether the lines should be the same. We have provided more details in the method section as well as in the figure legends.

12. The authors do make new contributions regarding the role of RAS mutations to pCMML, but the introduction gives the impression this is completely new. The authors should acknowledge / cite the prior work regarding RAS mutations in pCMML in the introduction. This would provide more appropriate context for their work.

Response: Thank you. We agree with you and have provided the necessary citations.

13. In figure S4G is the STAT3 blot for total STAT3 or pSTAT3, would not expect tSTAT3 to change so much.

Response: The Western blot was previously incorrectly labelled as "STAT3". This, in fact, represents phosphorylated STAT3 and not total STAT3. The Western blot label has been corrected to pSTAT3.

14. Figure legends are missing for S3J, S4F. The legend for S4F is about qPCR results and is for S4G. There is a typo "including" in the figure legend S5D-F.

Response: We apologize for this oversight. We have now accurately provided all the figure legends.

15. The statement regarding "most significantly associated" is still in the text in Figure 7.

Response: We apologize and have deleted this.

16. I appreciate the authors including the statement regarding NF1 in the commentary. However, in Figure S1B, NF1 appears to not to be so rare, and also appears more common in the dysplastic type than the proliferative. Perhaps this is due to the lower sample numbers?

Response: Yes, we agree with the reviewer. This is almost certainly skewed due to lower patient numbers. Looking at larger data sets the frequency of NF1 mutations in CMML is <5%.

17. In 3I picture given, the Megakaryocyte circled in the top left does not seem mono-lobated, the center one does. It would be clearer if enlarged pictures of the Megas are included for both WT and Mutant in the main figures.

Response: We apologize. We have now provided to enlarged pictures of the megakaryocytes from wild type and mutant mice in **Supplementary Figure 3J**.

Reviewer 3:

The authors have responded effectively to my comments. I have two minor comments

1. Suppl. Fig 4A: why did they decide on 5 samples each from different patients for pairwise correlation of expression. Would it not be more convincing using same sample source to compare among defines subpopulations.

Response: This is a very reasonable question. This issue has largely arisen due to significant attrition of sample size by sorting. Hence we had to use different sample sources to compare.

2. Suppl Fig 5J: Distribution of 5mc. 5hmc at TSS is not enough evidence that RAS and not TET2 mutation is responsible for the epigenetic phenotype in pCMML. For on 5mc/5hmc distribution in TET2 mutant context may not be all related to upstream occupancy but also to genebody occupancy. Also PLK1 and WEE1 are not the only and top hits for differential expression in pCMML vs dCMML.

Response: We agree with this comment. Given the superficial analysis that was provided with the DIP-seq data, we have actually removed this analysis from the manuscript.

Reviewer 4:

The authors tried to address the concerns previously raised by myself and other reviewers. I appreciate their efforts. Indeed, some new data such as metagene analysis have addressed one of my comments and improved the quality of this manuscript. However, their answers to many of my

comments were indirect or raised more concerns, in particular, about their interpretation on ChIP-seq data. Please see my specific comments to the authors.

I think that the first part (analysis of clinical samples, and mouse models) was very nicely done in an unbiased manner, although the second part (PLK1, WEE1, KMT2A) was somewhat biased. Hence, inclusion of the second part would give a negative impression, which might reduce overall quality of this paper.

Critically, I requested the authors to provide objective evidences to show how they selected PLK1 and WEE1 genes. The authors provided additional evidences (Figure 5B, a table “intersection”) after I requested and the editor reminded, which demonstrated that PLK1 and WEE1 were not one of the highest confidence targets. I have additional concern about ChIP-seq data and their representation, which is a critical evidence as one of selection criteria.

Response: This statement is incorrect and we apologize for any confusion. We have now made it very clear in the paper that the target (PLK1) was selected purely based on increased expression of protein coding genes in RAS mutant pCMML, in comparison to RAS wildtype dCMML. Nonetheless, the “intersection” data requested has now been included in **Supplementary Table 4**.

Although they stated the rationale for choosing PLK1 and WEE1 by wording, I still feel that there is lack of objective evidences to support their justification.

Response: We provide a list of top expressed protein coding and targetable genes (Figure 4C) and clarify that, while PLK1 was over expressed in pCMML, there were other genes that had higher expression values. However, amongst the top genes, given pre-existing rationale (thoroughly outlined in the paper) and the fact that there was a clinical grade small molecule inhibitor of PLK1 that had completed phase III clinical trials in myeloid neoplasms, we selected PLK1 for therapeutic targeting. **We did not use ChIP-seq data to identify targets. The ChIP-seq was carried out to identify mechanisms regulating the RAS-PLK1 axis.** We have done our best to highlight this in the text. We have also removed the WEE1 data given the lack of consensus peaks in the ChIP-seq experiments.

2. I worry if this paper may receive claims or correspondences about ChIP-seq data after being published as the current form.

Response: We sincerely hope that with all the changes and additional data, this is not the case.

3. The authors described the rationale for the selection of PLK1 and WEE1 genes by wording. I understand these points. I would agree with it, if it is supported by RNA-seq and ChIP-seq data. However, I still have many concerns.

Response: As mentioned above, **our selection criteria were purely based on the RNA seq data.** It certainly is a biased approach in the sense that, among the top over expressed protein coding and targetable genes, we selected PLK1 purely based on the fact that prior papers had demonstrated a connection between RAS and PLK1 (highlighted in the manuscript with references). This was further supported by the fact that Volasertib, a PLK1 inhibitor had completed phase 3 trials in acute myeloid leukemia, providing us with significant safety and efficacy data. We did not use the ChIP-seq data to select targets. The ChIP-seq studies were performed to delineate mechanisms regulating the RAS-PLK1 axis.

4. The authors provided a table (“intersection”) via email, though I originally requested them to include in Supplemental Information as a direct evidence to support their justification. Please officially include it as a Supplemental Table, if they are confident about these data. Basically, these data indicate that PLK1 and WEE1 were selected by RNA-seq and ChIP-seq but were not the highest confidence targets in terms of fold-change or ChIP-seq profile.

Response: Thank you. We are very confident about our data, especially given the robust responses that we see with the patient derived xenograft studies. As mentioned, we have removed WEE1 from the manuscript. Among the top overexpressed genes in RAS-mutant pCMML, PLK1 was the most easily druggable using a molecule validated in phase 3 trials, which motivated our choice. We now provide two data spread sheets; one providing all the genes that are differentially regulated between pCMML and dCMML and a 2nd spread sheet providing the intersection data between the RNAseq and the ChIP-seq.

5. They explained that they did a peak-calling by MACS2, which is a standard method. They replied that they have indicated peaks in red boxes along the gene track images in Figure 5E. There are two red boxes below pCMML track at PLK1. Otherwise, they did not further provide the peak calling result or images for each individual sample or did not comment how they evaluate statistical differences between dCMML and pCMML samples.

Response: We apologize for this. We now provide individual tracks with evidence for peak calling.

6. Also, they mentioned that “WEE1 and PLK1 were called in the majority of pCMML samples (80%) and in a minority of dCMML samples”, but I could not find this data or such description in the manuscript. “Minority” is a very non-scientific and ambiguous term.

Response: We once again apologize for this. We have completely done away with ambiguous terms. We have re-analyzed the data, strictly adhering to ENCODE criteria - <https://www.encodeproject.org/chip-seq/histone/>. We now provide individual tracks for 6 representative samples in both categories. We have converted the consensus bed files to bigwig files and present our UCSC genome browser session for review (https://genome.ucsc.edu/s/mbinder/CMML_RAS_H3K4me1).

7. The most confusing is that there were no red boxes at the WEE1 gene, which may mean that peaks were even not called or not significant for this gene? If yes, why this gene was selected by ChIP-seq? Although they have performed knockdown experiment to show the difference in H3K4me1 peak, it is a different setting and thus indirect evidence. Thus, from their responses, I understood that the authors failed to answer to my question in a straightforward manner.

Response: We apologize, WEE1 was **not** selected by ChIP-seq data, but based on its expression seen in the RNA-seq data. We however, agree with your concern that consensus peaks were not called for WEE1. Hence we have decided to remove this component from the manuscript.

8. Additionally, it is not clear how they performed ChIP-seq experiments. According to the entry in the GEO dataset they deposited, they described "Pooled all sample in 1:1:2 ratios for H3K4Me1 : H3K4me3 : H3K27Me3". But, these information were not described in the method. I guess that they performed ChIP-seq using different barcodes? How can other researchers reproduce their result? I have a strong concern in terms of the transparency.

Response: We apologize for this. We have improved on our submission to GEO in sincerely hope that the files are more easily accessible. Please note that this was not intentional and there was absolutely no intent to mask transparency. We completely vouch for our character and integrity. All ChIP-seq data used for downstream analyses have been deposited to the GEO under series GSE156377.

9. The authors explained about their definition of promoters (“conservative” and “liberal”). However, they did not answer to my original question. Which region is the promoter they defined for PLK1 and WEE1 genes? I am still not clear which element they are mentioning. As I commented above, there is no peak shown in red boxes at the WEE1 gene element. Because this is one of criteria by which they selected WEE1 gene (in Figure 5B, and a table), more detailed data are needed. This is a critical evidence that they have to show to convince reviewers and readers.

Response: Thank you. To reiterate, we did not select the targets based on ChIP-seq, but based on the over expression of genes from the RNAseq. We now use ENCODE criteria for promoter definition and define promoter regions as regions that regulate transcription of genes, located upstream and in close proximity to the transcription start sites; proximal promoter 2kb <https://www.encodeproject.org/chip-seq/histone/>

10. My original comment was on the quality of KMT2A ChIP. I suggested to do ChIP-PCR for KMT2B and KMT2C at this element but not H3K4me1. But, the authors tried to address this comment by performing a H3K4me1 ChIP-PCR after knockdown of KMT2C and KMT2D. This is an indirect evidence (it could be ignoratio elenchi). If they cannot perform ChIP for KMT2B or KMT2C, can they use their KMT2A ChIP samples for ChIP-seq to find if WEE1 and PLK1 gene loci are actually called?

Response: We sincerely apologize as we were not able to optimize antibodies for carrying out ChIP-PCR for KMT2B/C at PLK1 promoters. To do so would require extensive optimization experiments in cell lines and further validation with these additional antibodies in primary patient samples, which would further consume precious patient-derived samples and be time and cost-prohibitive at this point.

11. I am very cautious to conclude based on the statistical significance, because the difference between KMT2A CHIP-PCR and IgG can be significant as long as the deviation among experimental replicates is tight. They also did not describe whether they did experiments in technical replicates (by PCR) or biological replicates (by different experiments). Thus, it is difficult to evaluate the validity of their experiments.

Response: For each CHIP-PCR experiment, the indicated “n” represents the number of biological replicates. Each biological replicate consists of three technical replicates. This has now been made explicit in the updated figure legends. We agree that the differences in KMT2A and IgG are rather small, which, in our hands, is a consistent finding when performing CHIP-PCR experiments probing for binding factors (as opposed to histone post-translational modifications).

Reviewers' comments:

Reviewer #1 (Remarks to the Author):

No additional comments for the Authors.

Reviewer #2 (Remarks to the Author):

In this revision, the authors have done further analysis of their data to provide the link between RAS pathway mutations, proliferative CMML, and a proposed specific target PLK1 that is mediated through KMT2A activation. They also have data showing PLK1 inhibition with Volasertib is effective in NRAS mutant disease.

While the authors include a new citation regarding prior work on RAS mutations in pCMML, they do not write any new text giving this information- that RAS mutations have been previously linked to pCMML and its prognosis.

Were the selection of the ChIP seq samples the only ones that met criteria? Did there happened to be just 3 NRAS pathway mutant and 3 NRAS wt samples, or other criteria used to select them. Did none of the other fit the ENCODE criteria?

I don't understand why H3K4me1 is different in S5A but not different in 5A. "for all three histone marks, there were no statistically significant differences in these 6 samples (Fig. 5A). There however were numerically more H3K4me1 peaks in pCMML vs dCMML"?

The MA plot in 5B shows that the entire plot is shifted down. The fit line seems to go across log -1. Is the data properly normalized (for read counts for example) when comparing between dCMML and pCMML? If so, they should provide details on how it was done and other evidence that it is so dramatically changed globally.

The ChIP qPCR data is very nice and significant. However, the authors have removed WEE1 as a target that was proposed previously. Previously they showed WEE1 had similar convincing results. They did not answer my question on how their ChIP qPCR for PLK1 promoter using H3K4me1 and KMT2A could still show significant results without a specific defined target site. This would make me want to have greater assurances that these experiments were done properly for PLK1. Can they specify where the primer for qPCR is situated in terms of the promoter. The defined pCMML PLK1 H3K4me1 consensus in 5D is now different from last time. How was relative enrichment calculated? Was a primer site outside the region used to normalize?

Reviewer #3 (Remarks to the Author):

The authors have made very good faith effort to address questions raised by reviewers. The choice to focus on PLK1 is partly driven by druggability compared to other high expressing targets and that is understandable from a translational context. Patient derived biological material also does limit ability to perform all ideal experiments and

compared to kinase driven events, epigenetic alterations do remain more difficult to show very robust changes.

Given this constraints the authors have conducted large part of their work with patient derived material, which makes the data more translationally relevant than cell line/murine model derived data.

Reviewer #4 (Remarks to the Author):

The authors have reanalyzed CHIP-seq data and revised several figures including gene track at PLK1 locus. They have removed WEE1 data, as it was not logically supported by CHIP-seq. I think that these changes made the manuscript more solid and avoided confusion.

In method section, several sentences about WEE1 are still included. Please amend or remove these.

Response to Referees

Reviewer 1

No additional comments for the Authors.

Response: Thank you for your time and effort to review our manuscript and, in the process, helping strengthen it.

Reviewer 2

In this revision, the authors have done further analysis of their data to provide the link between RAS pathway mutations, proliferative CMML, and a proposed specific target PLK1 that is mediated through KMT2A activation. They also have data showing PLK1 inhibition with Volasertib is effective in NRAS mutant disease.

1. While the authors include a new citation regarding prior work on RAS mutations in pCMML, they do not write any new text giving this information- that RAS mutations have been previously linked to pCMML and its prognosis.

Response: Statements referencing previous work linking RAS mutations to pCMML and prognosis have now been included in the introduction and cited to provide better context.

2. Were the selection of the ChIP seq samples the only ones that met criteria? Did there happened to be just 3 NRAS pathway mutant and 3 NRAS wt samples, or other criteria used to select them. Did none of the other fit the ENCODE criteria?

I don't understand why H3K4me1 is different in S5A but not different in 5A. "for all three histone marks, there were no statistically significant differences in these 6 samples (Fig. 5A). There however were numerically more H3K4me1 peaks in pCMML vs dCMML"?

Response: We see why figures 5A and S5A may cause confusion. ChIP-seq was originally conducted probing for H3K4me1, H3K4me3 and H3K27me3 in 12 samples derived from patients with RAS wild type dCMML and 18 from those with RAS mutant pCMML. However, with re-analysis of our ChIP-seq data by strictly adhering to ENCODE criteria (sequencing depth and number of reads) for assessment of histone marks, the selected 6 samples were those that met such restrictions. Therefore, findings in S5A represent data from all 30 samples and data in 5A is the global assessment of the 6 samples that strictly met ENCODE criteria. Once again, we apologize for the confusion and, to avoid this, we have removed panel S5A.

3. The MA plot in 5B shows that the entire plot is shifted down. The fit line seems to go across log -1. Is the data properly normalized (for read counts for example) when comparing between dCMML and pCMML? If so, they should provide details on how it was done and other evidence that it is so dramatically changed globally.

Response: Thank you for the question. The MA plot comparing H3K4me1 peaks between pCMML and dCMML was generated by normalizing H3K4me1 peaks obtained by ChIP-seq, using `dba <- dba.analyze (dba,method = DBA_DESEQ2,bCorPlot = F,bParallel = 12)` and `dba.plotMA (dba,method = DBA_DESEQ2)`. The Bioconductor package `Diffbind v2.16.0` was used in R v4.0.3. Reads were counted in intervals using `dba.count` function and `mapQCth = 1, minOverlap = 3` parameters. Differential binding affinity analysis was performed using `dba.analyze` with `method = DBA_DESEQ2, bCorPlot = F` parameters. Differentially bound sites identified were then used to plot MA with `dba.plotMA` function. This global view demonstrates that H3K4me1 peak size is mostly down-regulated in dCMML,

suggesting decreased binding affinity in dCMML compared to pCMML samples. This explains why the fit line approaches log -1. We have also added this more detailed explanation to the methods section.

4. The ChIP qPCR data is very nice and significant. However, the authors have removed WEE1 as a target that was proposed previously. Previously they showed WEE1 had similar convincing results. They did not answer my question on how their ChIP qPCR for PLK1 promoter using H3K4me1 and KMT2A could still show significant results without a specific defined target site. This would make me want to have greater assurances that these experiments were done properly for PLK1. Can they specify where the primer for qPCR is situated in terms of the promoter.

Response: This is an excellent question and we apologize for overlooking it initially. A specific target site is necessary for ChIP qPCR experiments/analyses. Details regarding the sequence and relative location of the primer set used in ChIP qPCR experiments are detailed in the table below:

	Sequence	Position relative to first bp of exon one
sense	CTATGACCTGCCAGTTTGCTA	-588 bp
antisense	GCCTCCTCGCTACTGAATTT	-482 bp

These details have now been clarified in the methods section of the manuscript. This primer set was chosen for the following three reasons:

1. It is located within the promoter and just upstream of the transcriptional start site of PLK1.
2. Based on the ChIP-seq data, it lies approximately in the center of the H3K4me1 consensus peak.
3. Common optimal primer design factors were considered including parameters such as primer length, G/C content, melting temperature and differential melting between primer pairs.

5. The defined pCMML PLK1 H3K4me1 consensus in 5D is now different from last time. How was relative enrichment calculated? Was a primer site outside the region used to normalize?

Response: We appreciate this question. It is correct that consensus peaks for H3K4me1 now appear different relative to our original analysis. This is the consequence of re-analyzing our ChIP-seq data at the request of other reviewers. The consensus peak depicted in the original version of the manuscript was mistakenly analyzed with H3K4me1 as a narrow peak. Following ENCODE criteria, H3K4me1 signal should be analyzed as a broad as opposed to narrow peak, hence the broader consensus peak depicted. Narrow peaks are generally called for transcription factors, since the region bound is limited. However, broad peaks are better for H3K4me1 since such regions can be significantly wider. Of note, the two previously identified narrow consensus peaks are contained within the current broad peak. This should not affect our ChIP qPCR results given primer sets used are still within the consensus area.

The relative enrichment for ChIP qPCR was done by starting with the raw Ct values for each antibody used (H3K4me1, KMT2A, IgG control). The $\Delta\Delta Ct$ was then calculated by subtracting the IgG control Ct from the immunoprecipitation Ct. Fold enrichment was then calculated as $2^{-\Delta\Delta Ct}$. These details have now been clarified in the methods section of the manuscript.

Reviewer 3

The authors have made very good faith effort to address questions raised by reviewers. The choice to focus on PLK1 is partly driven by druggability compared to other high expressing targets and that is understandable from a translational context.

Patient derived biological material also does limit ability to perform all ideal experiments and compared to kinase driven events, epigenetic alterations do remain more difficult to show very robust changes.

Given this constraints the authors have conducted large part of their work with patient derived material, which makes the data more translationally relevant than cell line/murine model derived data.

Response: We are thankful for the time and effort you put into reviewing our work. In particular, we appreciate your insights into the technical difficulty of our presented experiments.

Reviewer 4

The authors have reanalyzed CHIP-seq data and revised several figures including gene track at PLK1 locus. They have removed WEE1 data, as it was not logically supported by CHIP-seq. I think that these changes made the manuscript more solid and avoided confusion.

In method section, several sentences about WEE1 are still included. Please amend or remove these.

Response: Thank you for pointing out these oversights in the methods section. References to WEE1 have now been completely removed from the manuscript.

Reviewers' comments:

Reviewer #2 (Remarks to the Author):

The authors have made important changes to improve the manuscript.

The authors have now placed language in the text regarding RAS mutations

They have provided details on their qPCR methods.

They have provided information about the PLK1 consensus site.

My only remaining question not fully answered:

They provide information about how the MA plot was analyzed. If it is normalized then, as I asked before, this mark H3K4me1 would be globally changed. I asked for evidence that would show this because it is not what they are showing in 5A where there is no difference in peaks. Can they better explain this difference in the text?

Response to Referees

Reviewer 2

My only remaining question not fully answered:

They provide information about how the MA plot was analyzed. If it is normalized then, as I asked before, this mark H3K4me1 would be globally changed. I asked for evidence that would show this because it is not what they are showing in 5A where there is no difference in peaks. Can they better explain this difference in the text?

Response: We apologize for not completely answering your initial question. Figure 5A shows numerically higher mean H3K4me1 peak counts in pCMML compared to dCMML. However, this difference is not statistically significant. If the number of samples was increased, this numerical difference would likely reach statistical significance at some point. The bars show mean H3K4me1 peak counts. A peak is either present (MACS2 FDR < 0.05 for a given peak) or not present (MACS2 $p \geq 0.05$ for a given peak). The confidence in the presence of a given peak (MACS2 FDR = 0.049 for one peak versus FDR = 0.001 for another) is not leveraged in this analysis. In contrast, the MA plot represents a DiffBind analysis for differential H3K4me1 enrichment. This analysis does leverage the information how confident a peak is being called across replicates. The use of this additional information makes it a more powerful analysis, yielding a large number of peaks being enriched in pCMML. Thus, both analyses agree on an increased/enriched number of H3K4me1 peaks in pCMML compared to dCMML. The simple mean number of H3K4me1 peaks analysis is simply not powerful enough to reach statistical significance with the number of available samples.